# Scalable Simulation-Based Model Inference with Test-Time Complexity Control

**Manuel Gloeckler** [1] **J.P. Manzano-Patrón** [2] **Stamatios N. Sotiropoulos** [2,3] **Cornelius Schröder** [1]
**Jakob H. Macke** [1,4]

## Abstract

Simulation plays a central role in scientific discovery. In many applications, the bottleneck is no longer running a simulator—it is choosing among large families of plausible simulators, each corresponding to different forward models/hypotheses consistent with observations. Over large model families, classical Bayesian workflows for model-selection are impractical. Furthermore, *amortized* model-selection methods typically hard-code a fixed model prior—or complexity penalty—at training time, requiring users to commit to a particular parsimony assumption before seeing the data. We introduce *PRISM*, a simulation-based encoder-decoder that infers a joint posterior over both discrete model structures and associated continuous parameters, while enabling test-time control of model complexity via a tunable model prior that the network is conditioned on. We show that PRISM scales to families with combinatorially many (up to billions of) model instantiations on a synthetic symbolic regression task. As a scientific application, we evaluate PRISM on biophysical modeling for diffusion MRI data, showing the ability to perform model selection across several multi-compartment models, on both synthetic and in-vivo neuroimaging data.

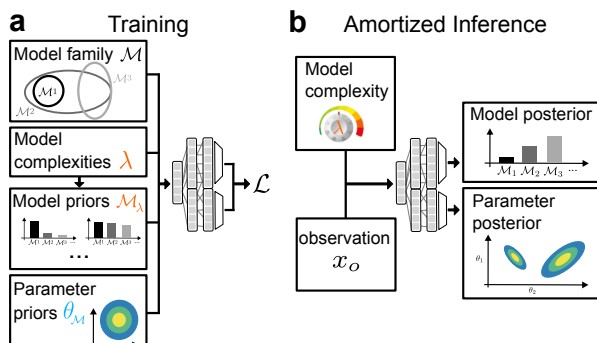

*Figure 1.* **PRISM overview.** **(a)** During training, we sample from a model family $\mathcal{M}$ with a hierarchical prior $p(\mathcal{M} \mid \lambda)$, where $\lambda$ controls a chosen notion of model complexity, and we jointly approximate the model posterior and the conditional parameter posterior $p(\mathcal{M}, \theta \mid \boldsymbol{x}, \lambda)$ by optimizing the loss $\mathcal{L} = \mathcal{L}(\mathcal{M}, p(\mathcal{M} \mid \boldsymbol{x}, \lambda)) + \mathcal{L}(\theta_{\mathcal{M}}, p(\theta \mid \mathcal{M}_{\lambda}, \boldsymbol{x}))$, for model evaluations $\boldsymbol{x} \sim p_{\mathcal{M}}(\boldsymbol{x} \mid \theta_{\mathcal{M}})$. **(b)** At inference time, we set $\lambda$ to tune parsimony, and select or explore models in the combinatorial space, infer model parameters, and deploy the resulting posteriors in downstream analyses.

## 1. Introduction

Simulation is a cornerstone of scientific inquiry and discovery (Winsberg, 2019; Skuse, 2019) and is increasingly relevant as researchers tackle more complex scientific systems (Lavin et al., 2022). In practice, researchers often face multiple plausible models for the same process, reflecting different mechanistic assumptions, parameterizations, and levels of abstraction. This proliferation creates a persistent challenge across modeling communities: deciding which model configurations are most supported by data in a given experimental regime, and how strongly.

For example, in computational neuroscience, Hodgkin–Huxley-type models (Hodgkin & Huxley, 1952) can be composed of many candidate ion-channel and morphological components (Schölzel et al., 2020) to represent neural activity and propagation. In epidemiology, the canonical SIR model (Kermack & McKendrick, 1927) has spawned a broad family of compartmental extensions (e.g., SEIR/SIRS, stratified or structured variants), with trade-offs between interpretability, identifiability, and predictive performance (Hethcote, 2000; Keeling & Rohani, 2008). In diffusion magnetic resonance imaging (dMRI), a range of

---

[*]Equal contribution [1]Machine Learning in Science, University of Tübingen and Tübingen AI Center, Tübingen, Germany [2]Sir Peter Mansfield Imaging Centre, School of Medicine, University of Nottingham, UK [3]National Institute for Health Research (NIHR) Nottingham Biomedical Research Centre, Queens Medical Centre, Nottingham, United Kingdom [4]Max Planck Institute for Intelligent Systems, Department Empirical Inference, Tübingen, Germany. Correspondence to: Manuel Gloeckler <manuel.gloeckler@uni-tuebingen.de>, Jakob H. Macke <jakob.macke@uni-tuebingen.de>.

*Proceedings of the $43^{rd}$ International Conference on Machine Learning*, Seoul, South Korea. PMLR 306, 2026. Copyright 2026 by the author(s).

multi-compartment biophysical models have been proposed as potential biomarkers to indirectly infer tissue microstructure from the scatter pattern of water molecules within the brain (Basser et al., 1994; Jelescu et al., 2020). Symbolic regression (Brunton et al., 2016; Biggio et al., 2021) can be seen through the same lens, where equations are constructed by selecting components from a large library. In all these cases, models formalize competing hypotheses, and selecting among them is central to scientific discovery. Yet, even modest libraries of components induce combinatorial model spaces that cannot be exhaustively enumerated or fit one by one. The central task is to infer which component combinations and parameters are supported by data, often prioritizing simple explanations over complex ones.

Bayesian model selection offers a principled framework to address this challenge, in particular by using criteria based on model-evidences (marginal likelihoods). However, there are two central challenges: First, Bayesian model selection methods must operate reliably over large model families. Conventional workflows (Gelman et al., 2020) are based on running Bayesian inference separately for each candidate model (e.g., via MCMC) and comparing models via marginal likelihoods, which scale poorly with the number of models and require tractable likelihoods. A flexible alternative is to *amortize* these costs by training neural networks to approximate key Bayesian computations once and reuse them across datasets. This connects to simulation-based inference (SBI) methods that learn posteriors, likelihood(-ratios) or model evidences from simulated data when likelihoods are unavailable (Cranmer et al., 2020a; Radev et al., 2023b; Deistler et al., 2025), and to "all-in-one" approaches that jointly learn posteriors and likelihood surrogates to support downstream model comparison (Radev et al., 2023a; Gloeckler et al., 2024; Chang et al., 2025). A second key challenge is controlling the trade-off between model complexity and goodness-of-fit. In principle, a prior over models can be parameterized (here, by $\lambda$) to interpolate between "narrow" priors favoring simple models and "broad" priors allowing more complex ones. Choosing $\lambda$ in advance is difficult, and practitioners often adjust this parsimony trade-off after seeing the data. However, most model-selection or discovery methods effectively lock in these assumptions at training time, making post-hoc adjustment difficult without retraining.

We propose *PRISM*, a *PRI*or-flexible *S*imulation-based *M*odel inference framework, based on a transformer encoder-decoder approach for simultaneous inference of models and model parameters (Fig. 1), which addresses both of these challenges. PRISM can identify posteriors over both model-components and associated parameters even in large model-spaces. In addition, it can amortize over a test-time tunable model prior. This allows practitioners to explore different complexities (e.g., in symbolic regression, expres-

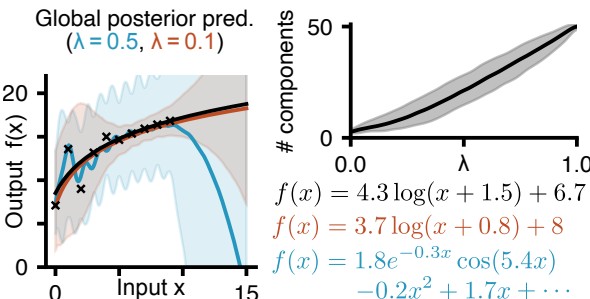

*Figure 2.* **Illustration on symbolic regression task.** *Left:* Ground-truth function and noisy observations (black, $x < 10$), compared to posterior predictive samples for two model complexities $\lambda$ (95% credible interval and one noiseless sample each, chosen as the simplest posterior draw). *Right:* Median number of model components as function of $\lambda$. *Below:* Equations of true function (black, 2 components), and two sampled equations for low and high complexity priors (2 and 9 components).

sions with different numbers of terms, Fig. 2) and select one without having to retrain the pipeline. Compared to previous model-inference approaches (Schröder & Macke, 2024), we remove several highly restrictive design choices, e.g., the reliance on Gaussian mixture density networks for the parameter posterior (and the associated need for analytic marginalization over inactive dimensions). Empirically, we show that PRISM amortizes effectively over large model families with billions of distinct model candidates on a synthetic symbolic regression example. We then showcase the application of PRISM to inference of tissue microstructure from dMRI data, by considering a range of biophysical multi-compartment models (e.g., Ball-and-Sticks (Behrens et al., 2007)). We demonstrate that PRISM can enable model comparison and selection in the context of fiber orientation estimation —on both synthetic and in-vivo data— and show that it outperforms previous SBI pipelines that consider each candidate model independently (Manzano-Patron et al., 2025).

## 2. Background and Notation

### 2.1. Bayesian parameter inference

Classically, we start with a probabilistic model $\mathcal{M}$ defined by model-specific parameters $\boldsymbol{\theta}_{\mathcal{M}}$ with prior $p(\boldsymbol{\theta}_{\mathcal{M}} \mid \mathcal{M})$ and a likelihood $p(\boldsymbol{x} \mid \boldsymbol{\theta}_{\mathcal{M}}, \mathcal{M})$. Given model $\mathcal{M}$ and data $\boldsymbol{x}_o$, Bayes' rule gives $p(\boldsymbol{\theta}_{\mathcal{M}} \mid \boldsymbol{x}_o, \mathcal{M}) \propto p(\boldsymbol{x}_o \mid \boldsymbol{\theta}_{\mathcal{M}}, \mathcal{M}) \, p(\boldsymbol{\theta}_{\mathcal{M}} \mid \mathcal{M})$. Classical methods (MCMC/VI) typically require model-specific implementations and repeated computation for inference per dataset $\boldsymbol{x}_o$ (Gilks et al., 1995; Beal, 2003).

Simulation-based (a.k.a. "likelihood-free") approaches operate purely on samples $\boldsymbol{x} \sim p(\boldsymbol{x} \mid \boldsymbol{\theta}_{\mathcal{M}}, \mathcal{M})$ without requiring likelihood evaluations. They also readily permit *amortized* inference, which trains a parameterized estimator (e.g., a

neural posterior, likelihood, or likelihood-ratio estimator) on simulated pairs $(\boldsymbol{\theta}_{\mathcal{M}}, \boldsymbol{x})$ to enable fast test-time inference (Papamakarios & Murray, 2016; Lueckmann et al., 2017; Radev et al., 2020; 2021; Jeffrey & Wandelt, 2024; Reuter et al., 2025). However, standard SBI approaches generally assume a fixed, well-specified model; under misspecification they may fail sharply out of distribution (Cannon et al., 2022; Kelly et al., 2025), making model construction and selection prerequisites for reliable amortization.

## 2.2. Bayesian model selection

To select an appropriate model, we consider a family of candidate simulators or probabilistic models $\boldsymbol{\mathcal{M}} = \{\mathcal{M}_1, \mathcal{M}_2, \dots\}$. Bayesian model comparison treats the model identity $\mathcal{M} \in \boldsymbol{\mathcal{M}}$ as a discrete latent variable with prior $p(\mathcal{M})$. Conditioning on data $\boldsymbol{x}_o$ gives $P(\mathcal{M} \mid \boldsymbol{x}_o) \propto p(\boldsymbol{x}_o \mid \mathcal{M}) \, P(\mathcal{M})$. Therefore, the *marginal likelihood* (Bayesian evidence)

$$ p(\boldsymbol{x}_o \mid \mathcal{M}) = \int p(\boldsymbol{x}_o \mid \boldsymbol{\theta}_{\mathcal{M}}, \mathcal{M}) \, p(\boldsymbol{\theta}_{\mathcal{M}} \mid \mathcal{M}) \, d\boldsymbol{\theta}_{\mathcal{M}} $$

is at the core of classical Bayesian model comparison (Jeffreys, 1939; Kass & Raftery, 1995; MacKay, 2003b, BMC). By averaging over the parameter prior, the evidence induces an Occam-type tradeoff: complex models that distribute probability mass broadly are penalized relative to simpler models that concentrate mass near the data $\boldsymbol{x}_o$ (Jeffreys, 1939; MacKay, 2003b). An alternative to model comparison or selection is the *Bayesian model average* over $P(\mathcal{M} \mid \boldsymbol{x}_o)$, which explicitly accounts for uncertainty over the model (Draper, 1995).

Computing the evidence requires high-dimensional integration and is generally intractable. Furthermore, for large model families $\boldsymbol{\mathcal{M}}$, workflows that run inference and evidence estimation *per model* become prohibitive. While Bayesian model comparison is sometimes seen as a nearly all-encompassing solution to model selection (MacKay, 2003a; Lotfi et al., 2022), the marginal likelihood penalizes a specific notion of *complexity* implicitly defined by the prior, which may be misaligned for a given task.

## 3. Methods

### 3.1. Problem setting

We aim to estimate the joint posterior over both models and parameters, $p(\mathcal{M}, \boldsymbol{\theta} \mid \boldsymbol{x}_o, \lambda) = P(\mathcal{M} \mid \lambda, \boldsymbol{x}_o) \, p(\boldsymbol{\theta} \mid \mathcal{M}, \boldsymbol{x}_o)$, for data $\boldsymbol{x} \in \mathbb{R}^{d_x}$ (Fig. 1), as in Schröder & Macke (2024). To control *model complexity* at inference time, we introduce a hierarchical prior on models $p(\mathcal{M} \mid \lambda)$, where $\lambda$ is a hyperparameter that controls model complexity. As the model space $\boldsymbol{\mathcal{M}}$ we consider all combinations of $C$ potential components, and index individual models as a binary

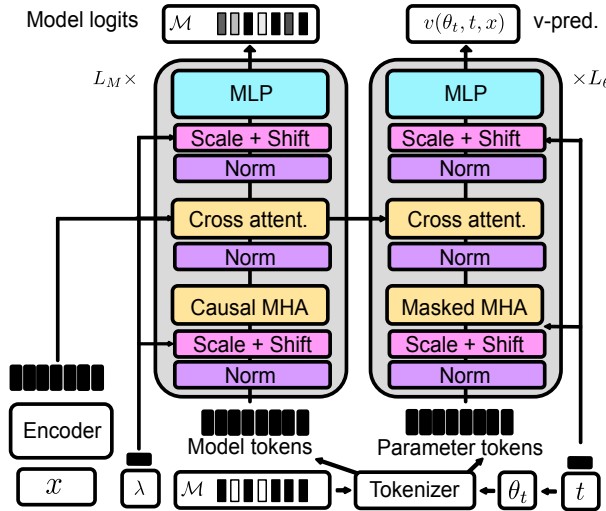

*Figure 3.* **Architecture overview.** PRISM is based on two parallel transformers to infer (i) the model posterior from tokenized model masks and (ii) the parameter posterior via a diffusion $v$-prediction network ($t$ denotes diffusion time). Observations $\boldsymbol{x}$ enter through cross-attention; $(\lambda, t)$ are injected via adaptive layer normalization (skip connections omitted).

mask $\mathcal{M} = (M_1, \dots, M_C)$ with $M_i \in \{0, 1\}$, representing the presence or absence of a specific component. Each component is associated with either independent or shared parameters $\boldsymbol{\theta}_{\mathcal{M}} \in \mathbb{R}^{d_{\mathcal{M}}}$ sampled from a model-specific parameter prior $p(\boldsymbol{\theta}_{\mathcal{M}} \mid \mathcal{M})$.

This setup yields two coupled approximation problems: (i) learning the *model posterior* $P(\mathcal{M} \mid \boldsymbol{x}_o, \lambda)$ (represented as a high-dimensional multivariate binary distribution) and (ii) learning the *parameter posterior* $p(\boldsymbol{\theta}_{\mathcal{M}} \mid \mathcal{M}_\lambda, \boldsymbol{x}_o)$, whose dimension varies with $\mathcal{M}$ (i.e., $\boldsymbol{\theta}_{\mathcal{M}} \subseteq \boldsymbol{\Theta} \in \mathbb{R}^{d_{\max}}, d_{\mathcal{M}} < d_{\max}$). Concretely, PRISM *amortizes Bayesian inference over a model space* which prior is parameterized by $\lambda$: a single trained network can be queried at any $\lambda$ at test time to obtain $p(\mathcal{M}, \boldsymbol{\theta} \mid \boldsymbol{x}_o, \lambda)$ without retraining.

### 3.2. Model architecture of PRISM

PRISM uses a transformer-based encoder-decoder architecture with two decoding streams: one for the discrete model structure, and a second that targets a posterior over continuous parameters given a model structure. These decoders are paired with a task-specific tokenizer, and both are conditioned on observations $\boldsymbol{x}$ and model complexity $\lambda$ (Fig. 3).

**Model posterior decoder.** We represent the model posterior as an autoregressive (AR) multivariate Bernoulli distribution over binary components $\mathcal{M} \in \{0, 1\}^C$: $q_\phi(\mathcal{M} \mid \boldsymbol{x}, \lambda) = \prod_{i=1}^{C} \text{Bern}\Big( M_i; p_\phi(M_{<i}, \boldsymbol{x}, \lambda) \Big)$. Simple factorized approximations fail systematically due to strong correlations, especially present on small model complexities $\lambda$ which enforces the posterior to concentrate on a few

low-complexity combinations (Fig. A.5, App. A.3). On structured discrete data such as natural language, autoregressive models still tend to outperform discrete-diffusion alternatives (Lou et al., 2024; Sahoo et al., 2024).

We implement conditioning on $x$ via cross-attention to the encoder tokens, and inject the model-complexity parameter $\lambda$ through adaptive layer normalization (AdaLN, Peebles & Xie 2023; Fig. 3, $\lambda$-dependent scale and shift). A causal attention mask with a padding token enables strict autoregressive decoding of the components.

**Parameter posterior decoder.**

To capture complex, potentially multimodal posteriors over continuous parameters—ubiquitous in our applications—we use a diffusion-based decoder, which is well-suited to flexible multimodal continuous densities and avoids the analytic-marginalization restrictions of mixture density networks (Schröder & Macke, 2024).

We adopt the EDM noise parameterization (Karras et al. (2022); i.e., $\boldsymbol{\theta}_t = \boldsymbol{\theta} + t\,\boldsymbol{\epsilon}$, $\boldsymbol{\epsilon} \sim \mathcal{N}(\mathbf{0}, \mathbb{I})$), and condition on the encoder context via cross-attention. Diffusion time $t$ is injected through AdaLN. The network is preconditioned for training on the $v$-prediction target (details in Sec. A.2),

$$\mathbf{v}_t = \alpha_t \boldsymbol{\epsilon} - \beta_t \boldsymbol{\theta}, \quad \mathbf{v}_\phi(\boldsymbol{\theta}_t, t, \boldsymbol{x}) = \alpha_t\,\hat{\boldsymbol{\epsilon}}_{\phi, \boldsymbol{\theta}_t, t, \boldsymbol{x}} - \beta_t\,\hat{\boldsymbol{\theta}}_{\phi, \boldsymbol{\theta}_t, t, \boldsymbol{x}}.$$

The diffusion decoder defines an amortized approximation $q_\phi(\boldsymbol{\theta} \mid \mathcal{M}, \boldsymbol{x}, \lambda)$ over a global parameter vector $\boldsymbol{\Theta} \in \mathbb{R}^{d_{\max}}$. We condition on $\mathcal{M}$ using an $\mathcal{M}$-dependent *block attention mask*: for any inactive component ($M_i = 0$), we mask (or remove) the corresponding parameter tokens from the diffusion-transformer input by masking all attention connections to and from that component's parameter block (equivalently, remove the associated rows/columns in the attention matrix; Fig. 3, Masked MHA). This effectively marginalizes unused parameters, reducing $\boldsymbol{\Theta}$ to $\boldsymbol{\theta}_{\mathcal{M}}$, while retaining a single shared parameter decoder across all models. When multiple components share global parameters, the associated tokens remain unmasked and attend to all active component tokens.

**Tokenizer.** The tokenizer is a task-dependent module and is built from established token-construction patterns (learnable identification embeddings, linear projections of continuous variables). It produces a token sequence for each decoder.

For model components, each component $i \in \{1, \ldots, C\}$ is assigned a learnable component identification token, shared by both decoders. The model decoder additionally requires the binary mask $\mathcal{M} = (M_1, \ldots, M_C)$; we encode this by adding a learned embedding of $M_i \in \{0, 1\}$ to the corresponding token. The parameter decoder requires the diffusion-state input $\boldsymbol{\theta}_t$ to be encoded as part of its token sequence. Rather than allocating a token per scalar parameter (as in autoregressive parameterizations), diffusion

permits structured grouping of parameters. We therefore use per-component parameter tokens: each component $i$ is associated with a parameter subvector $\boldsymbol{\theta}_i \in \mathbb{R}^{d_i}$, projected into the token space using a component-specific linear map. Components with $d_i = 0$ contribute no parameter tokens, and parameters shared by multiple components share the same token representation.

**Encoder.** The encoder maps the observation $x$ (and optional auxiliary variables) to a sequence of fixed-size vectors. The encoder can be chosen in a task-dependent manner. We use a transformer encoder, with positional embeddings for sequential data and without positional embeddings for exchangeable sets (e.g., unordered measurements).

### 3.3. Training

Training requires joint samples $(\lambda, \mathcal{M}, \boldsymbol{\theta}_{\mathcal{M}}, \boldsymbol{x})$ from the generative process. We optimize two losses jointly: (i) a Bernoulli negative log-likelihood for the model decoder (implemented as binary cross-entropy over $M_i$), and (ii) a diffusion $v$-prediction objective for the parameter decoder (Salimans & Ho, 2022). The overall training objective is the sum of these terms (Fig. 1, details in Appendix A.2). Since we target regimes with large $\mathcal{M}$-spaces, we use an *online* setup in which training data is generated on the fly. Jointly training both objectives in this online setup is empirically stable across our tasks; example loss curves and a detailed per-task compute breakdown (model sizes, training time, runtimes) are reported in App. A.2.4 (Fig. A.1, Tab. 1, Tab. 2).

### 3.4. Deployment

Having both a *model* and *parameter* posterior enables uncertainty quantification at two levels: *local* uncertainty conditional on a fixed model, and *global* uncertainty that marginalizes over model uncertainty (Draper, 1995; Werner et al., 2021). Furthermore, the model posterior enables Bayesian model comparison/selection via $q(\mathcal{M} \mid \boldsymbol{x}_o, \lambda)$. At deployment, model selection reduces to recovering the discrete MAP of $q_\phi(\mathcal{M} \mid \boldsymbol{x}, \lambda)$ (we compare best-of-$N$, tempered sampling, and beam search in App. A.4.4); inference costs are summarized in App. A.2.4.

## 4. Results

### 4.1. Evaluation metrics

We use four standard metrics to evaluate posterior approximation quality (formal definitions in Sec. A.1):

(i) *(r)RMSE* — posterior-predictive fit to observations; the relative form subtracts the simulation-noise floor, so values near 0 indicate near-optimal fit.

(ii) *(r)KSD* — likelihood-based discrepancy from the score;

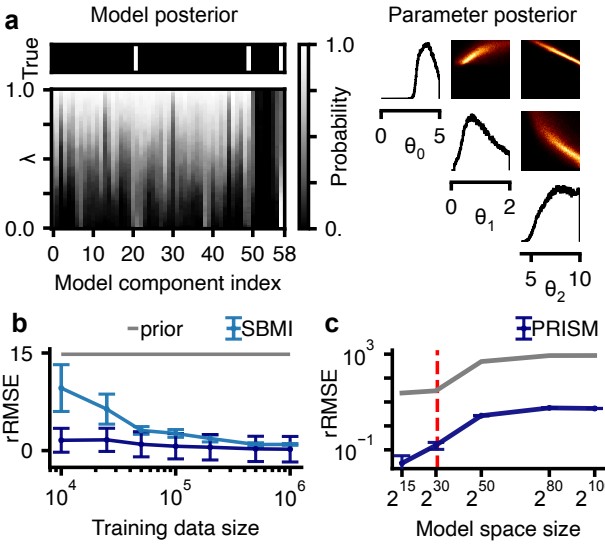

*Figure 4.* **PRISM on symbolic regression task.** **(a)** Model posterior across $\lambda$ and parameter posterior for the example in Fig. 2 (for $\lambda = 0.1$). **(b)** Comparison to (Schröder & Macke, 2024) for a fixed prior. **(c)** Scaling to large model spaces for a fixed computational training budget; red line indicates regimes beyond which not all models can be sampled during training.

   rKSD$< 1$ means the approximation is closer to the target than the prior.

(iii) *SBC / CE* — uses the fact that, under the simulator, the true parameter is itself a draw from the exact posterior. Thus, for calibrated $q$, it should rank like a typical draw among samples from $q$. Tail-heavy ranks indicate overconfidence; center-heavy ranks indicate underconfidence. CE summarizes the deviation from perfect calibration (CE $= 0$).

(iv) *ESS* — normalized effective sample size; quantifies how well approximate samples can be re-weighted toward the true posterior via importance sampling.

### 4.2. Toy Example: Noisy Symbolic Regression

To compare PRISM with SBMI (Schröder & Macke, 2024), we consider the additive symbolic regression task introduced in that work. This task is also readily extensible to substantially larger model spaces, which makes it well suited for studying scaling behavior in this regime.

The additive symbolic model family is defined as

$$f(x|\boldsymbol{\theta}, \mathcal{M}) = \sum_{k=0}^{K-1} M_k \cdot g_k(x|\boldsymbol{\theta}_k) + \sum_{m=K}^{K+M-1} M_m \cdot \epsilon_m(x|\boldsymbol{\theta}_m),$$

where $g_k$ are 'base functions' and $\epsilon_m$ represent different noise models, for a total of $C = K + M$ components. We follow the protocol of Schröder & Macke (2024): each function is evaluated on an equidistant grid over $[0, 10]$, but we extend the basis library to $K = 42$ unique base functions and $M = 8$ noise models (Tab. 10; per-$K$ basis

listing in App. B.1, Tab. 9). We allow repeated components, yielding up to $K = 100$ components. Repeated components can have different parameters and thus induce distinct probabilistic models and create posterior degeneracies that must be handled by the joint model–parameter inference networks. We define the complexity $\lambda$, and the corresponding family of model priors $\mathcal{M}_\lambda$ as $p(\mathcal{M} \mid \lambda) = \prod_k \mathrm{Ber}(M_k, \lambda) \prod_m \mathrm{Cat}(M_{K:M})$, with $\lambda \sim \mathrm{Unif}([0, 1])$. This definition gives rise to a simple yet intuitive complexity interpretation: $\lambda$ close to 1 results in complex models with many active model components ($M_i = 1$), whereas $\lambda \to 0$ results in sparse model samples. The noise models are unaffected and mutually exclusive. For all experiments we use as encoder a transformer with position encoding and 2 layers, and for model and parameter decoders 6 layers (details in Sec. A.2).

We first evaluate PRISM at $K = 50$ base functions. The model-averaged posterior predictive closely matches the observations in terms of RMSE (Fig. 2, A.2, A.3, Sec. A.1). Importantly, the posterior mass is spread across multiple distinct structures, indicating model uncertainty (Fig. 4a). Varying the complexity hyperparameter $\lambda$ reshapes the predicted posterior over $\mathcal{M}$, yielding structurally simpler explanations with a smaller number of components (Fig. 2a, 4a). At small $\lambda$, the posterior concentrates on constant-offset components ($i = 6, 7, 49, 50$) and logarithmic ($i = 20$) or logarithmic-like terms (e.g., $\sqrt{\cdot}$, $i = 21$), consistent with the data (Tab. 10). While the predictive fit is preserved within the training domain ($\boldsymbol{x} < 10$), deviations and uncertainty increase for more complex models when generalizing to $x > 10$ (Fig. 2b). This illustrates that explicit test-time control of the model prior $p(\mathcal{M} \mid \lambda)$ enables selecting a desired level of sparsity without retraining. For a direct comparison with prior work, we reproduce the largest setting in Schröder & Macke (2024) (SBMI, $K = 15$) with fixed model prior. PRISM clearly outperforms SBMI, even in the regime of small training data (Fig. 4b).

**Scaling with model-space.** A larger number of model components ($K \in \{30, 50, 80, 100\}$) results in model spaces from millions to $\mathcal{O}(10^{30})$ configurations with up to $d_{\max} = 223$ parameters, where per-model inference and evidence-based BMC are infeasible. In these settings, we train for a fixed time frame of 24 hours and then evaluate (irrespective of convergence). Up to model spaces of $2^{30}$, rRMSE is close to zero (Fig. 4c). Performance drops afterwards, which is expected as only a small subset of the model space is seen during training, but remains good relative to the task complexity (e.g., prior rRMSE, see Appendix A.3 for a discussion). Predictions remain well-calibrated even for large model spaces (Fig. A.3), though rKSD increases at the largest scales. We note that increasing network capacity mitigates this effect and consistently improves performance in both data-rich and data-limited regimes (Fig. A.3). We

further investigate model selection by constraining evaluation to a 200-model subspace and interpreting it as a 200-class classification problem in which data from many classes can be near-equivalent (i.e., redundant, functionally similar component combinations with substantial noise). We therefore report top-5 accuracy, measuring whether the true model lies among the five highest-probability candidates. Top-5 accuracy remains high across scales ($> 90\%$, Tab. 3), and we see a weak block structure in the confusion matrix (Fig. A.4), indicating that PRISM reliably concentrates posterior mass on a small set of plausible structures even when the full combinatorial space is underexplored.

### 4.3. Case study: fiber orientation reconstruction using diffusion MRI

We demonstrate applications of PRISM for biophysical model inference on *in-vivo* diffusion MRI (dMRI) neuroimaging data. dMRI is a non-invasive imaging technique that provides a powerful method to map microstructure and connectional architecture of the brain. This requires solving an inverse problem; biophysical models are used to link the diffusion profile of water molecules within brain tissue (as measured by the MR scanner) to the microstructural features of interest that hinder or restrict water diffusion. Over the past two decades, dMRI has accumulated a broad ecosystem of such signal models (see Panagiotaki et al. (2012) for a review). Most models are built as combinations of primitive compartments (e.g., isotropic 'ball' components to represent restricted diffusion within cell bodies, and anisotropic Gaussian compartments such as stick/tensor/zeppelin to define directionality). The task at hand, for example, is to select the number of 'stick' compartments in each voxel of the image that represent different axonal fiber orientations given the data (Fig. 5).

The structure of this problem, with hundreds of thousands of voxel-wise estimations per brain (i.e., voxel-wise fittings), is inherently suited for inference amortization. Each dMRI dataset comprises multiple acquisitions with given diffusion-encoding settings, conventionally summarized by a gradient direction $\vec{b}$ ('bvec') and a diffusion-weighting strength $b$ ('bval') (Fig. 6a). Recent simulation-based approaches have shown the potential of amortized frameworks for solving inverse problems in dMRI (Eggl & De Santis, 2024; 2025; Manzano-Patron et al., 2025). However, learning in these prior studies is restricted to a fixed model with a fixed acquisition protocol, often tied to a specific noise level and model, thus only providing inference over parameters (but not on multiple models). PRISM can address all these limitations, allowing for rapid joint inference on models and

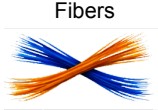
Fibers

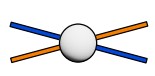
Ball & Sticks

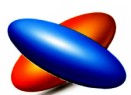
Tensors

*Figure 5.* dMRI models.

model parameters that is amortized across voxels, dMRI acquisition settings, and noise settings. PRISM is fully trained on simulations, but we also evaluate on real-world data, namely the UK Biobank (UKB) (Sudlow et al., 2015) and HCP (Van Essen et al., 2012) datasets.

#### 4.3.1. BALL-AND-STICKS MODEL

Next we demonstrate that PRISM jointly recovers fiber orientations and selects among Ball-and-Sticks compartment counts on in-vivo dMRI data, while remaining calibrated across acquisition protocols and matching established MCMC references in downstream tractography.

Following the specifications of Manzano-Patron et al. (2025) (App. B.2, with task configuration in App. B.2.4), we consider the multi-shell Ball-and-Sticks model (B3S) (Behrens et al., 2007; Jbabdi et al., 2012). Here, the normalized attenuation of the dMRI signal $S/S_0$ is explained as a combination of an isotropic *ball* compartment, $g_B(b, \vec{b}) = \exp(-bd)$, and a weighted sum of up to $N$ anisotropic *sticks*: $g_S(b, \vec{b}) = \exp\big(-bd\langle\vec{b}, \vec{\mu}\rangle^2\big)$. Model parameters include the diffusivity $d$, stick orientations $\vec{\mu}$, and per-compartment weights $f_k$, $k = 1, \ldots, K$.

The inference network is conditioned on diffusion-weighted measurements $\boldsymbol{S} = [s_1, \ldots, s_n]$ (Fig. 6a) and the corresponding acquisition settings $(b, \vec{b})$, enabling amortization across acquisition protocols. Since $\boldsymbol{S}$ is an exchangeable set indexed by acquisition settings (not an IID sequence), we encode it with a shallow (2-layer) transformer without position embeddings. We use the same model prior as in Sec. 4.2. We amortize over a broad range of randomly generated in-vivo-like protocols, varying b-values (up to $b = 6000$ s/mm$^2$), gradient directions $\vec{b}$, and number of acquisitions ($n$), enabling deployment across heterogeneous datasets. We consider models with $N \in \{0, 1, 2, 3\}$ stick compartments, amortizing across a range of fiber-complexity representations.

**Joint-posterior inference performance.** For each voxel $V_i$, we infer a posterior over models and, conditional on the model probabilities, a posterior over its parameters in an amortized manner (Fig. 6a). The parameter posterior admits a complex multimodal geometry due to mixture symmetries and the posterior predictives closely match the observations (Fig. 6a, right, Fig. A.8, A.12). The inferred model posterior aligns closely with a likelihood-based reference obtained from an unbiased Monte Carlo evidence estimator with PRISM's parameter posterior as importance proposal (Fig. 6a, down left; Fig. A.9, Appendix Sec. A.4.1 for the derivation). We compute SBC over $10^5$ simulations with ranks pooled across $\lambda$, $\mathcal{M}$ and $\boldsymbol{\theta}_{\mathcal{M}}$; per-model calibration errors remain close to the Monte Carlo floor for both posteriors and across acquisition schemes (Fig. 6b). To complement SBC with a discrepancy measure applicable

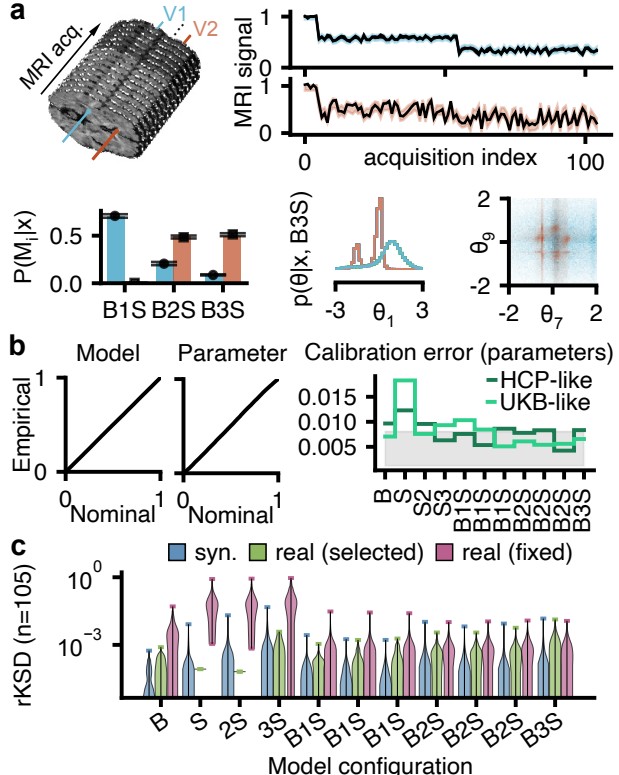

*Figure 6.* **PRISM on dMRI. (a)** Two voxels (V1, V2): dMRI signal (black), posterior predictives (color; 95% quantile), and multimodal parameter posteriors (bottom right; full marginals in Fig. A.8); model posteriors (bottom left) match unbiased MC evidence estimates (black). **(b)** *Left:* SBC over the full amortization scope. *Right:* per-model calibration error across the model space for two acquisition schemes (gray: expected error). **(c)** rKSD to the posterior on synthetic and experimental B3S-family data, for a 'fixed' vs. 'selected' model on UKB (HCP in Fig. A.7) → fixed-model SBI degrades under misspecification; joint selection restores accuracy. (B$X$S = Ball + $X$ Sticks; identically labeled models differ only in prior.)

to real experimental data, we apply rKSD to the parameter posterior (Fig. 6c). rKSD confirms excellent approximation on synthetic data, as well as on experimental data within the full PRISM pipeline. Importantly, if we restrict evaluation to a specific model class as a control experiment, it reveals larger deviations under overly simple (or inappropriate) models, consistent with expected performance degradations of SBI under model misspecification (Cannon et al., 2022). This analysis shows the importance of model selection for trustworthy predictions in this task.

Overall, these results show that the inferred model posterior closely matches evidence-based BMC ($R^2 = 0.97$; Fig. A.9), and that the parameter posterior is near the true posterior. Moreover, high effective sample sizes ($\approx 60\%$ on synthetic data; $\approx 40\%$ on real data, Tab. 4) suggest that remaining discrepancy can be corrected efficiently, e.g., via importance sampling (Dax et al., 2023). We further compare against a strong single-model (B3S) normalizing-flow

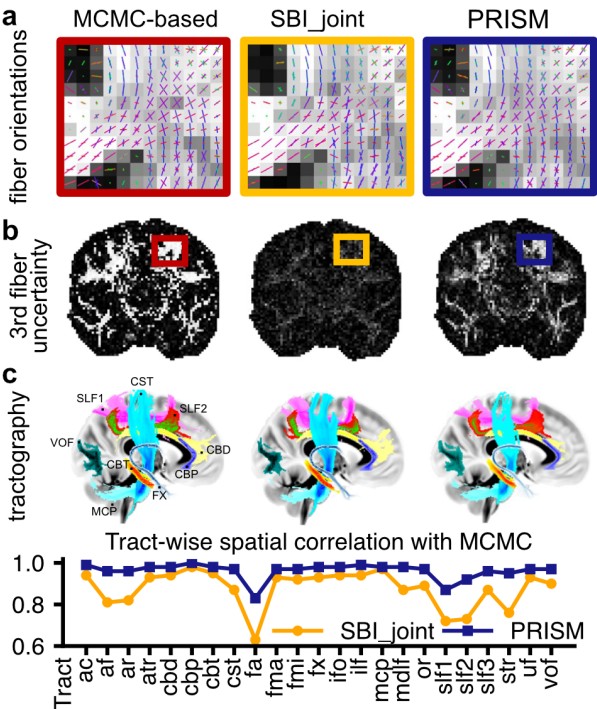

*Figure 7.* **Downstream Tractography. Comparisons against MCMC (BedpostX) and previous SBI approaches. (a)** 3-way fiber crossings from the Centrum Semiovale area. **(b)** Uncertainty of the third stick (fiber) orientation in terms of dispersion. High uncertainty (dark) indicates the third fibre is not supported by the data. *Last row:* Tractography of brain regions based on inferred posteriors. **(c)** Tractwise correlation with MCMC, compared to SBI$_{joint}$ (tract labels from a).

baseline (Sec. A.4.1, Tab. 5) under the same training budget. PRISM matches this baseline on synthetic data and improves on real data, while additionally estimating the model posterior and amortizing across the full B3S family. This supports that PRISM provides an accurate joint model–parameter posterior approximation.

**Probabilistic tractography.** The posterior distributions of fiber orientations can be used to identify probabilistic white-matter paths of anatomical connectivity. We use XTRACT (Warrington et al., 2022) to evaluate probabilistic tractography. We compare our reconstructed bundled tracts against the ones obtained using posterior estimates from BedpostX (Behrens et al., 2003), the domains gold-standard MCMC approach for this task (although not a ground-truth), and the SBI$_{joint}$ approach with restricted model choice proposed in (Manzano-Patron et al., 2025).

Compared to the SBI$_{joint}$ baseline, PRISM improves agreement with MCMC and increases coherence in crossing-fiber regions (Fig. 7a). In particular, the uncertainty of fiber orientations is better matched and yields sharper contrast between white and non-white matter (Fig. 7b). These improvements directly translate into higher spatial correlation of the reconstructed bundle tracts with MCMC across brain regions (Fig. 7c), with mean correlations of 0.96 versus 0.86 for the

SBI baseline. In addition, PRISM can be applied to a much larger set of datasets (i.e., arbitrary $b$, $\vec{b}$) without retraining. However, the computational cost of PRISM generally increases compared to SBI$_{joint}$ (based on normalizing flows) and benefits from GPU acceleration. Generating one thousand model posterior samples takes 5 ms, and 50 ms for the parameter posterior (on an Nvidia H100, Fig. A.6), allowing inference on, e.g., the UKB dataset within 4 minutes (in batches of 30k voxels).

### 4.3.2. EXTENDED MODEL CLASSES

Finally, to assess whether PRISM can scale to larger and heterogeneous model spaces, we extend the approach to a broader component library. PRISM can then either be queried for a specific model of interest, or to perform model selection over this much larger model (sub)space. This setup is intended as a proof of principle demonstrating scalability and flexibility of the method on dMRI-relevant model spaces.

We extend B3S with three Zeppelin (Z) and three Tensor (T) compartments—axially symmetric and full anisotropic Gaussian diffusion compartments, respectively—to form the BSZT space, and further add spherical-convolution components, e.g., Zhang et al. (2012); Sotiropoulos et al. (2012), to obtain BSZT+conv (Sec. B.2, Tab. 11). Throughout, we use a compact shorthand for compartment combinations: B, S, Z, T denote Ball/Stick/Zeppelin/Tensor; a prefix counts repetitions of the next compartment (e.g., B3S = one Ball + three Sticks, BZ = Ball + Zeppelin,...). In total, this library comprises 21 model components that can be grouped combinatorially, yielding a model family with up to 114 parameters (compared to up to 12 parameters in the B3S setting). Since candidates differ in parameterization, we penalize model complexity by parameter count via a dimension-penalizing prior (Sec. B.2.5).

We first verify on synthetic data that inference is well calibrated (Fig. 8a, with minor deviation for BSZT+conv, Fig. A.10) and achieve generally low predictive RMSE and KSD, but a drop in ESS (Tab. 6, 7). PRISM distinguishes models when they are sufficiently different, but, as expected, cannot reliably distinguish similar or equivalent models (Fig. A.11, Tab. 8). We ran model selection on UKB data (Sec. A.4.5), and investigated whether some fixed models or the full Bayesian model average had strong statistical support on empirical data: inferred models outperform DTI (Basser et al., 1994) and maximum-likelihood methods such as Rumba (Garyfallidis et al., 2014; Canales-Rodríguez et al., 2015) in signal reconstruction and leave-one-out cross-validation (Fig. 8b). Mean performance is comparable among inferred models, yet they often induce different uncertainty in fiber-orientation estimates (Fig. 8c; Fig. A.13): principal directions largely agree, but uncer-

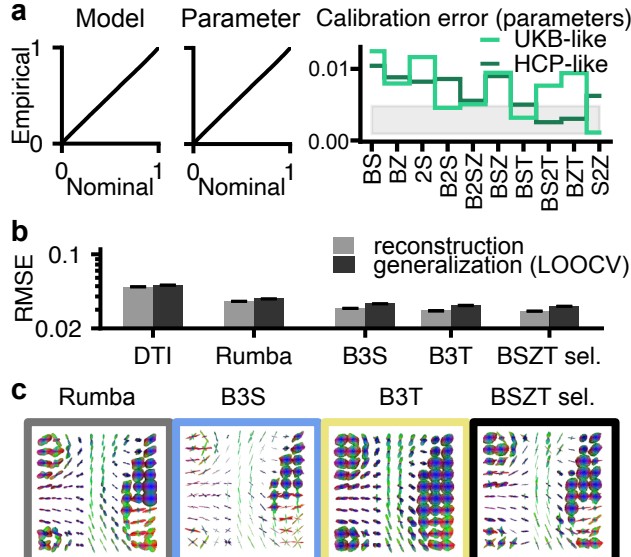

*Figure 8.* **PRISM on an extended model family for dMRI. (a)** Calibration error for the BSZT model on average (left) and per model class for the parameter posterior (right; x-tick labels use the compartment shorthand of Sec. 4.3) → low calibration error across the BSZT model space, with minor deviations on a subset of models. **(b)** Reconstruction and generalization RMSE for reference methods (DTI, Rumba) and for the inferred B3S, B3T, and BSZT-selected model average (LOOCV-based generalization) → inferred models match or outperform DTI/Rumba on both reconstruction and LOOCV. **(c)** Examples of predicted fiber orientation distributions (Sec. B.2.2) for a local patch → principal directions are robust across models, but the predicted uncertainty depends on the chosen model class.

tainty can vary substantially.

Overall, these results demonstrate that PRISM can amortize joint model-parameter inference across large, heterogeneous, and combinatorial model spaces that would be intractable with per-model approaches. It can be used to discover data-consistent models within large model spaces (Sec. A.4.5, Fig. A.15) and can be applied to different experimental setups (e.g., acquisition protocols). Even in the setting of noisy, real-world data, it yields model and parameter inferences that remain well calibrated, have competitive statistical performance on real data, and yield interpretable uncertainty estimates.

## 5. Related Work

In Approximate Bayesian Computation (ABC), likelihood-free model choice augments inference with a discrete model index and estimates posterior model probabilities via rejection sampling or SMC (Toni et al., 2009; Toni & Stumpf, 2010; Liepe et al., 2010). However, consistency requires sufficient summary statistics (Robert et al., 2011; Marin et al., 2014). ABC-RF instead casts model choice as supervised classification (Pudlo et al., 2016). Neural SBI amortizes inference and often uses classifier-like model selection to

infer posteriors over discrete model indices (Radev et al., 2021), with recent work improving accuracy and validation for moderately sized candidate sets (Kucharský et al., 2025; Elsemüller et al., 2024; Schumacher et al., 2025), but typically omitting parameter inference. Explicit enumeration during training or evaluation limits scalability to combinatorially large spaces. Model comparison can also use evidence estimates from NPE/NLE/NRE plug-in estimators (Spurio Mancini et al., 2023) or flow-based normalizer recovery from unnormalized densities (Srinivasan et al., 2024). "All-in-one" systems learn per-model posteriors and surrogate likelihoods for evidence computation (Gloeckler et al., 2024; Radev et al., 2023a), with COMPASS (Gunes et al., 2025) extending this direction. Although such workflows allow test-time changes to the model prior, they still require Monte Carlo integration over per-model parameter posteriors and, for COMPASS, per-model training, limiting scalability to combinatorial model spaces.

SBMI (Schröder & Macke, 2024) performs joint model–parameter inference over model collections, but relies on restrictive approximations such as analytic marginalization of inactive dimensions; its mixture-density-network posterior is less expressive than flows, diffusion models, or autoregressive decoders (Papamakarios et al., 2021; Karras et al., 2022). More broadly, *conditional amortization* generalizes amortized inference to context variables that can be set at test time (Elsemüller et al., 2023; Chang et al., 2025), of which our test-time prior $\lambda$ is one instance; PRISM additionally amortizes over the *discrete model identity* $\mathcal{M}$ itself—a high-dimensional structured latent that existing conditional-amortization frameworks do not address. Related are also mixed discrete–continuous SBI methods (Boelts et al., 2026; Ghiglino et al., 2026). Ghiglino et al. (2026) is particularly close in spirit, but uses discrete diffusion rather than autoregressive decoding over discrete aircraft topologies, which are analogous to model identities in our setting.

Relatable, symbolic regression seeks interpretable equations from data. Early methods relied on genetic programming (Schmidt & Lipson, 2009) or sparse regression over fixed bases (Brunton et al., 2016; Bakarji et al., 2023). Deep-learning approaches use graph neural networks (Cranmer et al., 2020b) or transformers (Biggio et al., 2021), but are typically deterministic and do not support systematic model comparison.

## 6. Discussion

We introduced PRISM, an amortized simulation-based inference framework for *joint inference* over discrete model structure and continuous parameters in large, combinatorial model families. By learning a joint posterior $p(\mathcal{M}, \theta \mid x, \lambda)$, PRISM supports not only model selection but also *model discovery* and efficient *marginalization over model uncer-*

*tainty* within a single amortized system. One advantage of amortization is the ability to perform rapid *Bayesian model averaging* across large sets of plausible models. Rather than conditioning conclusions on a single model, predictions and downstream quantities can be marginalized over model identity, weighted by posterior probabilities. This makes uncertainty due to competing mechanistic hypotheses explicit and is particularly valuable in settings requiring repeated or time-critical inference.

PRISM enables controlled exploration of model space via a tunable hierarchical prior $p(\mathcal{M} \mid \lambda)$ that trades off complexity and fit; this trade-off need not be fixed at training time. Choosing $\lambda$ *a priori* is often difficult and its effect on operational complexity is highly nonlinear—a familiar issue in nonparametric Bayesian methods (Escobar & West, 1995; Teh et al., 2006; Ghahramani, 2013), where complexity-controlling hyperparameters are routinely adjusted after inspecting inferred solutions. By conditioning on $\lambda$, PRISM lets users either fix it from prior knowledge or explore it post hoc (e.g., via cross-validation), much like selecting a regularization strength.

These benefits come with limitations. Like other large-scale amortized methods (Reuter et al., 2025), PRISM requires substantial simulation and training budgets, and in combinatorial model spaces only a small fraction of configurations is sampled during training; while PRISM generalizes well under such undersampling, posterior quality degrades when too large regions remain unseen, and our experiments illustrate promising trends rather than a controlled scaling-law study. The transformer backbone is quadratic in the number of model components $C$, which is unproblematic here ($C \leq 100$) but limits direct extension to libraries with thousands of components. Natural extensions include structured model priors (hierarchical, dependency-aware, or domain-conditioned), alternative parameter decoders (autoregressive or flow-matching), and sparse or linear-attention transformer variants to lift the quadratic-in-$C$ bottleneck.

Despite these challenges, PRISM shows that amortized inference can scale model discovery, selection, and averaging to regimes that are inaccessible to classical per-model Bayesian workflows while retaining calibrated uncertainty over both model identity and parameters. PRISM provides a practical foundation for simulation-based scientific inference in rich and heterogeneous model and data spaces, and we expect it to be useful across scientific disciplines.

## Software and Data

We used `jax` (Bradbury et al., 2018), `hydra` (Yadan, 2019) to track all configurations, and utilities from `sbi` (Boelts et al., 2025). Code to reproduce results at https://github.com/mackelab/prism.

## Impact Statement

This paper presents work whose goal is to advance the field of machine learning for science. There are many potential societal consequences of our work, none of which we feel must be specifically highlighted here.

## Acknowledgments

We thank Stefan Wahl for feedback on the manuscript. We thank all members of the Mackelab for discussions and feedback on the manuscript. This work was funded by the German Research Foundation (DFG) under Germany's Excellence Strategy – EXC number 2064/1 – 390727645 and SFB 1233 'Robust Vision' (276693517), the German Federal Ministry of Education (Tübingen AI Center), the European Union (ERC, DeepCoMechTome, 101089288), and the "Certification and Foundations of Safe Machine Learning Systems in Healthcare" project funded by the Carl Zeiss Foundation. MG is a member of the International Max Planck Research School for Intelligent Systems (IMPRS-IS). JP and SS are supported by an ERC Consolidator Grant (101000969), and JP is also supported by a Wellcome Trust, UK bioimaging technology award (313367/Z/24/Z).

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

# A. Experimental details

We first define the metrics we used to assess approximation quality across the amortization scope (Sec. A.1). We then provide additional training details (Sec. A.2).

## A.1. Evaluation metrics

Having both a *model* posterior and a *parameter* posterior enables uncertainty quantification at two levels: *local* uncertainty conditioned on a fixed model, and *global* uncertainty that marginalizes over model uncertainty (Werner et al., 2021). Concretely, we evaluate the two posterior predictive distributions:

$$
\begin{aligned}
\text{Local}: \ & p(\boldsymbol{x} \mid \boldsymbol{x}_o, \mathcal{M}) \ \approx \ \mathbb{E}_{q(\boldsymbol{\theta}\mid\boldsymbol{x}_o,\mathcal{M})}[p(\boldsymbol{x} \mid \boldsymbol{\theta}_{\mathcal{M}}, \mathcal{M})], \\
\text{Global}: \ & p(\boldsymbol{x} \mid \boldsymbol{x}_o) \ \approx \ \mathbb{E}_{q(\mathcal{M},\boldsymbol{\theta}\mid\boldsymbol{x}_o)}[p(\boldsymbol{x} \mid \boldsymbol{\theta}_{\mathcal{M}}, \mathcal{M})].
\end{aligned}
\tag{1}
$$

The model posterior further supports Bayesian model comparison (BMC) over any user-specified subset $\{\mathcal{M}_i\} \subset \boldsymbol{\mathcal{M}}$ by ranking models via $q(\mathcal{M}_i \mid \boldsymbol{x}_o, \lambda)$ (Radev et al., 2020; Elsemüller et al., 2024). Finally, it enables *model discovery* by exploring the combinatorial space through samples $\mathcal{M} \sim q(\mathcal{M} \mid \boldsymbol{x}_o, \lambda)$, surfacing plausible component combinations beyond hand-designed candidates.

However, to measure performance we report the following quantities since exact reference posteriors are infeasible across our amortization scopes:

(i) *Predictive RMSE*: We report *local* RMSE under a fixed model and *global* RMSE under the model-averaged posterior predictive (Gelman et al. 1996). We report relative RMSE (rRMSE) to the lower bound (noise floor) where appropriate.

(ii) *Calibration (SBC)*: Simulation-based calibration (SBC) assesses self-consistency of the inferred posterior by testing rank uniformity for parameters and models drawn from the generative process (Talts et al., 2018). Intuitively, SBC asks whether the true simulated value looks like a typical draw from the inferred posterior; deviations indicate that the posterior is too narrow (true value too often in the tails, "overconfident") or too wide (true value clustered near the center, "underconfident"). We report the absolute deviation of the empirical rank-CDF from the nominal $\mathrm{Uniform}(0,1)$ diagonal as the calibration error (CE). Note that calibration is a necessary, but not sufficient criterion for the correctness of the posterior distribution.

(iii) *Sample quality via (r)KSD*: When likelihood scores are available, we evaluate posterior sample quality using (kernelized) Stein discrepancy, which does not require the posterior normalizing constant (Liu et al., 2016; Gorham & Mackey, 2017). We report a normalized rKSD to mitigate scale effects across models and parameter dimensionalities.

**Predictive RMSE.** For an observation $\boldsymbol{x}_o$ and a fixed model $\mathcal{M}$, we form the *local* posterior predictive and for the full model family the *global* (model-averaged) posterior predictive following Equation 1. We compute MSE on the predictive samples $\boldsymbol{x}_i$ sampled from the local or global posterior predictive:

$$
\mathrm{MSE}(\boldsymbol{x}_i, \boldsymbol{x}_{o,i}) \ = \ \frac{1}{D} \sum_{j=1}^{D} \left( \hat{x}_i^j - x_{o,i}^j \right)^2,
$$

where $D$ is the dimensionality of $\boldsymbol{x}_o$. For *local* MSE (i.e., conditioned on a model) we use $\boldsymbol{x}_i \sim \mathbb{E}_{q(\boldsymbol{\theta}\mid\boldsymbol{x}_o,\mathcal{M})}[p(\boldsymbol{x} \mid \boldsymbol{\theta}, \mathcal{M})]$, and for *global* RMSE we use $\boldsymbol{x}_i \sim \mathbb{E}_{q(\mathcal{M},\boldsymbol{\theta}\mid\boldsymbol{x}_o)}[p(\boldsymbol{x} \mid \boldsymbol{\theta}, \mathcal{M})]$. For a dataset of $N$ observations we obtain the RMSE

$$
\mathrm{RMSE} = \sqrt{\frac{1}{N} \sum_{i=1}^{N} \mathrm{MSE}(\boldsymbol{x}_i, \boldsymbol{x}_{o,i})}
$$

As this includes irreducible error from noise within the simulator we normalize RMSE by a task-specific irreducible-error baseline. Concretely, for each ground-truth setting $(\mathcal{M}^\star, \boldsymbol{\theta}^\star)$ we generate $R$ replicate observations $\boldsymbol{x}_o^{(r)} \sim p(\boldsymbol{x} \mid \boldsymbol{\theta}^\star, \mathcal{M}^\star)$ with different simulator seeds. Computing the RMSE on such a dataset yields $\mathrm{RMSE}_{\min}$. We report $\mathrm{rRMSE} = \mathrm{RMSE} - \mathrm{RMSE}_{\min}$ so values close to $0$ indicate performance near the simulation-noise floor.

**Calibration (SBC).** Simulation-based calibration (SBC) checks self-consistency under the generative process: if $(\mathcal{M}, \boldsymbol{\theta}, \boldsymbol{x})$ are drawn from the simulator and we rerun inference on $\boldsymbol{x}$, then suitable probability integral transform (PIT) values, e.g., rank statistics, should be $\mathrm{Uniform}(0,1)$ (Talts et al., 2018). We apply SBC to both the *model posterior* $q(\mathcal{M} \mid \boldsymbol{x})$ (discrete) and the *parameter posterior* $q(\boldsymbol{\theta} \mid \mathcal{M}, \boldsymbol{x})$ (continuous, conditional on the generating mask). Notably, this only allows performance evaluation of the *uncertainty calibration* in the *well-specified* (i.e., synthetic) case.

More specifically, we sample for each trial $t = 1, \ldots, T$ model, parameters and data: $\mathcal{M}^{(t)} \sim p(\mathcal{M})$, $\boldsymbol{\theta}^{(t)} \sim p(\boldsymbol{\theta} \mid \mathcal{M}^{(t)})$, $\boldsymbol{x}^{(t)} \sim p(\boldsymbol{x} \mid \boldsymbol{\theta}^{(t)}, \mathcal{M}^{(t)})$, and then compute the probabilities $q(\mathcal{M} \mid \boldsymbol{x}^{(t)})$ and $q(\boldsymbol{\theta} \mid \mathcal{M}^{(t)}, \boldsymbol{x}^{(t)})$. We then compute SBC for the model masks and the parameters separately:

*Model SBC (discrete).* Let $\ell_{\mathrm{true}}^{(t)} = \log q(\mathcal{M}^{(t)} \mid \boldsymbol{x}^{(t)})$. Draw $S$ masks $\widetilde{\mathcal{M}}^{(t,s)} \sim q(\mathcal{M} \mid \boldsymbol{x}^{(t)})$ and set $\ell_s^{(t)} = \log q(\widetilde{\mathcal{M}}^{(t,s)} \mid \boldsymbol{x}^{(t)})$. We use a randomized rank to handle ties: define $n_<^{(t)} = \sum_s \mathbf{1}[\ell_s^{(t)} < \ell_{\mathrm{true}}^{(t)}]$ and $n_=^{(t)} = \sum_s \mathbf{1}[\ell_s^{(t)} = \ell_{\mathrm{true}}^{(t)}]$, draw $K^{(t)} \sim \mathrm{Unif}\{0, \ldots, n_=^{(t)}\}$ and $V^{(t)} \sim \mathrm{Unif}(0,1)$, and form $u_{\mathrm{model}}^{(t)} = (n_<^{(t)} + K^{(t)} + V^{(t)})/(S+1)$. This is important in discrete cases, since there is a non-zero probability of sampling the exact same mask with the exact same log probability.

*Parameter SBC (continuous, conditional on the generating model).* Let $\mathcal{I}(\mathcal{M}^{(t)})$ be the active parameter indices under $\mathcal{M}^{(t)}$, and draw posterior samples $\boldsymbol{\theta}^{(t,s)} \sim q(\boldsymbol{\theta} \mid \mathcal{M}^{(t)}, \boldsymbol{x}^{(t)})$. For each $j \in \mathcal{I}(\mathcal{M}^{(t)})$, compute the rank of the truth among samples, e.g. $r_j^{(t)} = 1 + \sum_s \mathbf{1}[\theta_j^{(t,s)} < \theta_j^{(t)}]$, and convert to a PIT value $u_j^{(t)} = (r_j^{(t)} - 0.5)/(S+1)$. We then assess uniformity by pooling all $\{u_j^{(t)}\}$ across trials and active indices.

*Calibration summary.* Given PIT values $\{u\}$ (either $\{u_{\mathrm{model}}^{(t)}\}$ or the pooled $\{u_j^{(t)}\}$), we compare the empirical CDF $\widehat{F}(u)$ to the nominal $F_0(u) = u$. As a scalar error we report $\mathrm{CE} = |\mathcal{G}|^{-1} \sum_{g \in \mathcal{G}} |\widehat{F}(g) - g|$ on a grid $\mathcal{G} \subset [0,1]$. We use a uniform grid with 100 vertices as default.

**(Relative) multi-scale KSD.** To quantify how well samples from the parameter posterior match the target posterior, we use a preconditioned, multi-scale Kernel Stein Discrepancy (KSD, Liu et al. (2016); Gorham & Mackey (2017)) computed *conditional on the generating model $\mathcal{M}$*. For each trial (or voxel) we draw samples $\{\boldsymbol{\theta}^{(s)}\}_{s=1}^S \sim q(\boldsymbol{\theta} \mid \mathcal{M}, \boldsymbol{x})$ and evaluate them against the unnormalized log posterior $\log p(\boldsymbol{\theta} \mid \mathcal{M}, \boldsymbol{x}) = \log p(\boldsymbol{x} \mid \boldsymbol{\theta}, \mathcal{M}) + \log p(\boldsymbol{\theta} \mid \mathcal{M})$, where the prior is applied only to the active dimensions indicated by $\boldsymbol{\theta}$-mask $\mathbf{m}_\theta(\mathcal{M})$ (effective dimension $d_{\mathrm{eff}} = \sum_j \mathbf{m}_{\theta,j}$). We obtain the score $\nabla_{\boldsymbol{\theta}} \log p(\boldsymbol{\theta} \mid \mathcal{M}, \boldsymbol{x})$ by automatic differentiation.

*Preconditioning.* To reduce sensitivity to anisotropy and scale, we whiten samples in the active subspace using an empirical covariance preconditioner: with $LL^\top \approx \widehat{\mathrm{Cov}}(\boldsymbol{\theta} \odot \mathbf{m}_\theta)$, we set $P = L^{-1}$, transform samples to $\boldsymbol{z} = P\boldsymbol{\theta}$, and transform scores accordingly $\boldsymbol{s}_{\boldsymbol{z}} = P^{-\top} \boldsymbol{s}_{\boldsymbol{\theta}}$ (again masking inactive dimensions).

*Multi-scale Stein kernel.* In whitened space we use an RBF base kernel with multiple bandwidths $k_h(\boldsymbol{z}, \boldsymbol{z}') = \exp(-\|\boldsymbol{z} - \boldsymbol{z}'\|^2/(2h^2))$ for $h \in \mathcal{H}$, and average the corresponding Stein kernels uniformly across scales. We use a broad range ($h \in \{0.1, 0.5, 1.0, 10.0\}$) to increase sensitivity to a broader range of "discrepancies". The (squared) KSD is computed as the usual U-statistic over the resulting Stein Gram matrix $H$:

$$\mathrm{KSD}^2 = \frac{1}{S(S-1)} \sum_{i \neq j} H_{ij}, \qquad \mathrm{KSD} = \sqrt{\max(\mathrm{KSD}^2, 0)}.$$

*Relative KSD (rKSD).* Because KSD magnitude depends on dimension and problem, we also report a standardized discrepancy. Let $\mathrm{KSD}_{\mathrm{prior}}$ be the KSD obtained by replacing posterior samples with prior samples $\boldsymbol{\theta}^{(s)} \sim p(\boldsymbol{\theta} \mid \mathcal{M})$ (using the same $S$, mask, preconditioning, and bandwidth set). We then define $\mathrm{rKSD} = \mathrm{KSD}/\mathrm{KSD}_{\mathrm{prior}}$, so $\mathrm{rKSD} < 1$ indicates the approximation is closer to the target than the prior baseline.

## A.2. Model architecture and training details

### A.2.1. MODEL DETAILS AND CONFIGURATIONS

The full model comprises (i) an *encoder* for the simulator output and global parameters (e.g., diffusion measurements and acquisition meta-data), (ii) a *tokenizer* that produces model component- and parameter-level token sequences together with structured attention masks, and (iii) two transformer *decoders*: an autoregressive (AR) decoder for the model posterior $q_\phi(\mathcal{M} \mid \mathbf{x}, \lambda)$ and a diffusion decoder for the parameter posterior $q_\phi(\boldsymbol{\theta} \mid \mathcal{M}, \mathbf{x}, \lambda)$.

We only vary `model_dim` and `num_layer` across experiments; all other hyperparameters are fixed or derived from these. We use 4 attention heads, an attention size of 16 (query/key/value projection dimension), and widening factor 4 (i.e., the feedforward MLP expands the hidden dimension by $4\times$ `model_dim`).

**Encoder.** Encoders across tasks are represented by a standard self-attention transformer encoder. For the symbolic regression task we group 10 datapoints into a single token and additionally add a positional embedding. For the dMRI task we embed the acquisition settings, i.e., $(b, \vec{\mathbf{b}})$, and diffusion signal $S$ as follows: we use a Random Fourier Embedding for the b-value $b$, and a linear mapping for the signal $S$ and the bvec $\vec{\mathbf{b}}$, which all get concatenated into a token of size `model_dim` that is then transformed by a transformer without positional embedding (to ensure exchangeability).

**Tokenizers.** The learnable tokenizers have a similar structure but task-specific variations. All used tokenizers have a learnable "identification" vector per component in which the necessary information is provided for the decoders.

The model decoder additionally requires the binary mask $\mathcal{M} = (M_1, \ldots, M_C)$; we encode this by adding a learned embedding of the mask values $M_i \in \{0, 1\}$ to the corresponding token.

(i) `SymbolicTokenizer`: We have several parameters per model component (Tab. 10). Each parameter of $\boldsymbol{\theta}_{\mathcal{M}} \subset \boldsymbol{\Theta}$ is linearly projected to a `model_dim`-dimensional vector and added to the component embedding. In addition to this embedding layer, we also need a component-specific linear decoding layer to reduce the token to the respective parameter dimension.

(ii) `DMRITokenizer` extends this to also allow for *global* or *shared* parameters per model component. For global parameters (i.e., shared diffusivity in B3S or the constraints on the fractions $\boldsymbol{f}$), we introduce a learnable identification token per dimension and add a linear projection of the value (e.g., $f_i$) to each token. In a similar style, *shared parameters* are grouped into an additional token and are removed from the corresponding compartment tokens.

We additionally construct a $\mathcal{M}$-dependent *block attention mask*: if $M_i = 0$, the corresponding parameter-token block is removed from the diffusion-transformer input and all attention connections to and from that block are masked. This implements attention-based marginalization of unused parameters and ensures $\boldsymbol{\theta}$ predictions only depend on active components (Gloeckler et al., 2024).

**Model-posterior decoder.** Discrete model selection is implemented with an *autoregressive* transformer decoder, representing the model posterior as a multivariate Bernoulli distribution over $\mathcal{M} \in \{0, 1\}^C$:

$$q_\phi(\mathcal{M} \mid \mathbf{x}, \lambda) = \prod_{i=1}^{C} \text{Bern}\Big( M_i \,\Big|\, p_\phi(M_{<i}, \mathbf{x}, \lambda) \Big). \tag{2}$$

Conditioning on $\mathbf{x}$ is implemented via cross-attention to the encoder tokens; $\lambda$ is injected through adaptive layer normalization (AdaLN) (Peebles & Xie, 2023). Specifically, we embed $\lambda$ by a Random Fourier Embedding. Each AdaLN block then uses an MLP to predict a "scale and shift" of size `model_dim` that then scales or shifts the transformer's hidden state. In contrast to Peebles & Xie (2023), we do not use additional gating. A causal attention mask (with the padding tokens) enforces strict autoregressive decoding. We can thus evaluate the probability within a single forward pass, but require $C$ forward passes for sampling it autoregressively. The evaluation cost is hence $\mathcal{O}(C^2)$ and the naive sampling cost is $\mathcal{O}(C \cdot C^2)$ ($\mathcal{O}(C^2)$ when using KV caching at the cost of memory).

**Parameter-posterior decoder.** Continuous parameter inference uses an EDM-style *diffusion* transformer (Karras et al., 2022). For a global parameter vector $\boldsymbol{\Theta} \in \mathbb{R}^{d_{\max}}$, we sample $t \in [10^{-4}, 80]$ and perturb parameters as

$$\boldsymbol{\theta}_t = \boldsymbol{\theta} + t\,\boldsymbol{\epsilon}, \qquad \boldsymbol{\epsilon} \sim \mathcal{N}(\mathbf{0}, \mathbf{I}), \tag{3}$$

conditioning on encoder tokens via cross-attention and injecting $t$ via Gaussian–Fourier time embeddings and AdaLN. We train with $v$-prediction,

$$\mathbf{v}_t = \alpha_t \boldsymbol{\epsilon} - \beta_t \boldsymbol{\theta}, \quad \mathbf{v}_\phi(\boldsymbol{\theta}_t, t, \boldsymbol{x}) = \alpha_t\, \hat{\boldsymbol{\epsilon}}_{\phi, \boldsymbol{\theta}_t, t, \boldsymbol{x}} - \beta_t\, \hat{\boldsymbol{\theta}}_{\phi, \boldsymbol{\theta}_t, t, \boldsymbol{x}}.$$

with $\alpha_t, \beta_t$ given by the normalized SNR $\alpha_t = \frac{1}{\sqrt{1+t^2}}, \beta_t = \frac{t}{\sqrt{1+t^2}}$. Note that $\mathbf{v}_\phi(\boldsymbol{\theta}_t, t, \boldsymbol{x})$ does not need two evaluations for $\hat{\boldsymbol{\epsilon}}, \hat{\boldsymbol{\theta}}$. From $\hat{\boldsymbol{\theta}}$ as given by Karras et al. (2022), we obtain $\hat{\boldsymbol{\epsilon}} = (\boldsymbol{\theta}_t - \hat{\boldsymbol{\theta}})/t$.

Sampling uses a 64-step EDM schedule on the probability flow ODE (Song & Ermon, 2019). If not specified otherwise, we use an ODE-based solver, e.g., the exponential Adams–Bashforth solver (Lu et al., 2025), which only requires a single forward pass per step. Since we group parameters compartmentwise, the number of tokens is $\mathcal{O}(C)$ and the sampling cost is $\mathcal{O}(T \cdot C^2)$, where $T$ is the number of sampling steps in the diffusion model.

To impose a consistent prior on $\boldsymbol{\theta}$ across models, we *reparameterize* each task-specific prior into a standard-normal latent space via a bijection. Concretely, we introduce latent variables $\boldsymbol{\theta} \sim \mathcal{N}(\mathbf{0}, \mathbf{I})$ and define task parameters as $\boldsymbol{\theta}_{\text{natural}} = T(\boldsymbol{\theta})$, where $T$ maps unconstrained Euclidean latents to the appropriate constrained domain (see below, e.g., intervals, manifolds). This ensures that samples always satisfy the task constraints (e.g., positivity, simplex constraints, or manifold membership), while allowing us to use a consistent normal prior across dimensions. Additionally, this allows us to reduce dimensionality to that of the domain and reduces degeneracy.

We use the following bijective transforms within our experiments:

- *Uniform on an interval* $[a, b]$ (componentwise): for $\boldsymbol{\theta} \sim \mathcal{N}(0, 1)$, set $u = \Phi(z) \in (0, 1)$ and

$$\theta = a + (b - a)\, u,$$

  where $\Phi$ is the standard-normal CDF.

- *Uniform on the upper half-sphere:* $\mathbb{S}_+^2 = \{\boldsymbol{n} \in \mathbb{R}^3 : \|\boldsymbol{n}\|_2 = 1,\ n_z \geq 0\}$: draw $z_1, z_2 \sim \mathcal{N}(0, 1)$, set $u_1 = \Phi(z_1)$, $u_2 = \Phi(z_2)$, and define

$$\varphi = 2\pi u_1, \qquad \theta = \arccos(u_2),$$

$$\boldsymbol{n} = \big(\sin\theta \cos\varphi,\ \sin\theta \sin\varphi,\ \cos\theta\big),$$

  which yields a uniform distribution over surface area on the hemisphere.

- *Simplex* $\Delta^{K-1}$ (Dirichlet stick-breaking): for a Dirichlet prior $\boldsymbol{\pi} \sim \text{Dir}(\boldsymbol{\alpha})$, we use the stick-breaking construction with $K-1$ latent variables $\boldsymbol{\epsilon} \in \mathbb{R}^{K-1}$. We map $\boldsymbol{\epsilon}$ to uniforms $u_i = \Phi(\epsilon_i)$ and then to Beta variables via the inverse regularized incomplete beta function:

$$v_i = I_{u_i}^{-1}(\alpha_i,\ b_i), \qquad b_i = \sum_{j>i} \alpha_j,$$

  followed by

$$\pi_i = v_i \prod_{j<i}(1 - v_j), \quad i = 1, \ldots, K-1, \qquad \pi_K = \prod_{j=1}^{K-1}(1 - v_j).$$

### A.2.2. ONLINE TRAINING

Data is generated on the fly via a streaming `SimulationDataset`. A simulator runs on CPU (if not otherwise specified) and fills a ring buffer of size $10^5$ in batch-size-sample chunks; a background thread refreshes consumed slots and host-to-device prefetching keeps the accelerator saturated. Training batches are fetched in a non-blocking manner indefinitely. Optimization uses RAdam (learning rate $5 \cdot 10^{-4}$), adaptive gradient clipping at 2.0, no scheduler, and an exponential moving average (EMA) tracking of parameters (decay 0.999). The objective combines an auto-regressive mask negative log-likelihood (implemented as binary-cross-entropy) and a diffusion $v$-prediction loss:

$$\mathcal{L}_{\mathcal{M}} = -\mathbb{E}\left[ \sum_{i=1}^{C} \log \text{Bern}\Big(M_i \,\Big|\, p_\phi(M_{<i}, \mathbf{x}, \lambda)\Big) \right], \qquad \mathcal{L}_{\boldsymbol{\theta}} = \mathbb{E}_{t, \boldsymbol{\epsilon}}\left[ \big\|\hat{\mathbf{v}}_\phi(\boldsymbol{\theta}_t, t, \mathbf{x}, \mathcal{M}) - \mathbf{v}_t\big\|_2^2 \right]. \tag{4}$$

Runs are executed on a single H100 GPU with 16 CPU cores, 32 GB host RAM. As final parameters we use the latest EMA parameter.

### A.2.3. ADDITIONAL SPECIFICATIONS

For the symbolic regression tasks, we varied `model_dim` $\in \{32, 64, 128\}$ while keeping the number of layers fixed. We used 2 layers for the encoder and 6 layers for the decoder, and trained with a batch size of 4096. For the comparison with SBMI, we used `model_dim`= 64, reduced the decoder to 4 layers, and reduced the batch size to 1024.

For the dMRI networks, we used `model_dim`= 128. The **B3S model** used layers $(2, 4, 6)$ for (encoder, model-selection, parameter inference), respectively, and the B3S simulator trained on a mixed acquisition scheme combining UKB and HCP-like data (Sec. B.2 for details). The **BSZT model** used layers $(2, 6, 8)$ for (encoder, model-selection, parameter inference), respectively, and trained on the same mixed acquisition scheme. The **BSZT+conv model** used `model_dim`= 128 and the same stack depths as the BSZT model, but with the spherical-convolution extension enabled in the simulator and training data generation. As the spherical convolution is much more expensive, we ran the simulation here also on GPU. All dMRI networks were trained for 72 hours.

### A.2.4. COMPUTE AND REPRODUCIBILITY

All networks were trained on a single NVIDIA H100 80GB HBM3 GPU with 8 CPU cores and 32 GB system RAM. The underlying server was equipped with $2\times$ AMD EPYC 9654 96-core processors (384 logical CPUs total), NVIDIA driver 580.105.08, and CUDA driver/runtime 13.0. Training uses batch size 2048; inference operations are reported at batch size 1. Symbolic networks were trained for 24 hours, dMRI networks for 72 hours.

We summarize parameter counts and wall-clock costs for sampling, log-prob evaluation, and forward/backward passes for the dMRI tasks alongside the NSF + Transformer baseline (Sec. A.4.1) in Tab. 1, and for the symbolic tasks across model sizes (`Tiny`: `model_dim`=32; `Full`: `model_dim`=128) in Tab. 2. Compared with the fixed-model NSF baseline, PRISM is moderately slower at parameter sampling and roughly $8\times$ slower at parameter log-prob evaluation (due to the diffusion ODE), while its backward pass is approximately $4\times$ faster, reflecting a lighter per-step training graph. We note that, unlike PRISM, the NSF baseline produces only the parameter posterior for a single fixed model and does not provide a model posterior or amortization across the model family.

Training loss curves (Fig. A.1) show that PRISM trains stably across all settings, with minor recoverable spikes that do not affect the final EMA parameters.

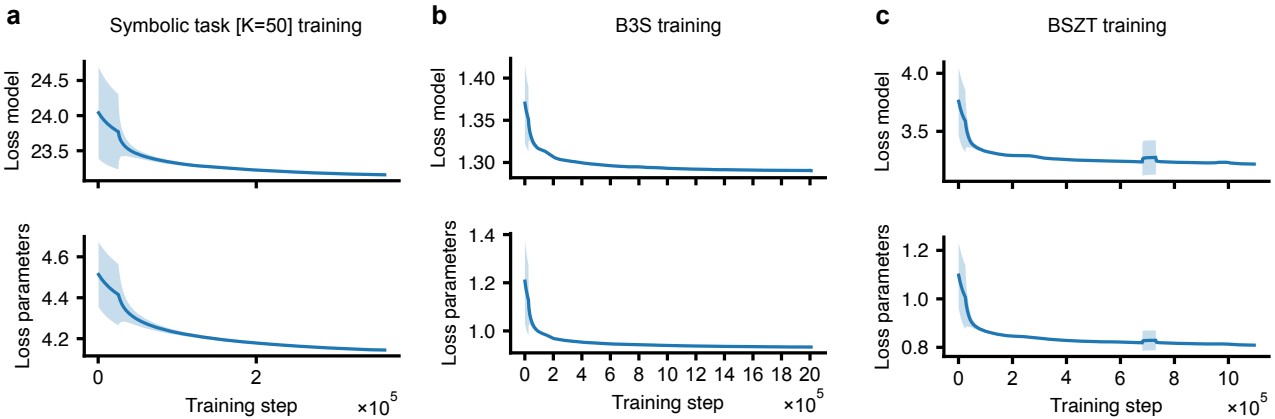

*Figure A.1.* **Training loss evolution** for **(a)** symbolic regression task with $K = 50$ components, **(b)** B3S-space model, and **(c)** BSZT-space model. Training is generally stable; occasional spikes are recovered by the next checkpoint and do not affect the final EMA parameters.

*Table 1.* **Model size and compute cost for the NSF + Transformer baseline and the PRISM dMRI models.** Wall-clock time is reported as mean ± standard deviation over 10 warmed-up runs. Inference operations use batch size 1; training operations use batch size 2048. Memory values are reported with the measurement used in each experiment: the NSF + Transformer baseline reports incremental peak CUDA allocated memory, whereas the PRISM dMRI variants report compiler-estimated temporary working memory (`temp_size_in_bytes`); the memory figures are therefore not directly comparable across columns. For the NSF + Transformer baseline, attempts to use `torch.compile` were unsuccessful due to `W0328` graph breaks; only `aot_eager_with_fallback` executed, with similar or worse runtime than eager execution.

| Category | Quantity | NSF + Transformer | B3S | BSZT | BSZT+ conv |
|---|---|---|---|---|---|
| **Parameter counts** | | | | | |
| | Total | 7,186,076 | 6,816,361 | 9,432,650 | 9,451,008 |
| | Encoder | 1,588,608 | 340,894 | 506,206 | 506,206 |
| | Parameter decoder | 5,597,468 | 4,547,648 | 6,063,424 | 6,063,424 |
| | Model decoder | – | 1,913,921 | 2,837,569 | 2,837,569 |
| | Tokenizer | – | 13,898 | 25,451 | 43,809 |
| **Inference** | | | | | |
| | Sample model | – | $1.05 \pm 0.01$ ms
0.42 MiB | $3.53 \pm 0.02$ ms
0.68 MiB | $6.10 \pm 0.01$ ms
0.80 MiB |
| | Log-prob model | – | $0.34 \pm 0.01$ ms
0.37 MiB | $0.47 \pm 0.02$ ms
0.63 MiB | $0.47 \pm 0.02$ ms
0.75 MiB |
| | Sample parameter | $19.32 \pm 1.37$ ms
0.69 MiB | $22.18 \pm 0.20$ ms
0.55 MiB | $30.86 \pm 0.18$ ms
1.37 MiB | $32.60 \pm 0.30$ ms
1.56 MiB |
| | Log-prob parameter | $17.31 \pm 1.28$ ms
0.69 MiB | $138.69 \pm 0.03$ ms
1.77 MiB | $247.63 \pm 0.04$ ms
8.94 MiB | $358.00 \pm 0.06$ ms
25.92 MiB |
| **Training (batch size 2048)** | | | | | |
| | Forward pass | $50.90 \pm 0.91$ ms
11,441.82 MiB | $10.59 \pm 0.12$ ms
668.03 MiB | $20.31 \pm 0.06$ ms
668.05 MiB | $27.50 \pm 0.08$ ms
668.04 MiB |
| | Backward pass | $142.36 \pm 0.90$ ms
12,026.89 MiB | $35.12 \pm 1.79$ ms
7.31 GiB | $69.75 \pm 0.41$ ms
15.45 GiB | $101.28 \pm 0.57$ ms
22.87 GiB |

*Table 2.* **Model size and compute cost for symbolic tasks.** Wall-clock time is reported as steady-state mean ± standard deviation over 10 warmed-up runs. Memory is reported as compiler-estimated temporary working memory (`temp_size_in_bytes`). Inference operations use batch size 1; training operations use batch size 2048.

| Category | Quantity | 15-task | | 50-task | | 100-task | |
|---|---|---|---|---|---|---|---|
| | | Tiny | Full | Tiny | Full | Tiny | Full |
| **Parameter counts** | | | | | | | |
| | Total | 1,313,037 | 5,770,461 | 1,321,124 | 5,802,548 | 1,331,344 | 5,843,104 |
| | Encoder | 34,272 | 332,544 | 34,272 | 332,544 | 34,272 | 332,544 |
| | Inference decoder | 821,120 | 2,972,480 | 821,120 | 2,972,480 | 821,120 | 2,972,480 |
| | Model decoder | 453,889 | 2,444,353 | 453,889 | 2,444,353 | 453,889 | 2,444,353 |
| | Tokenizer | 3,756 | 21,084 | 11,843 | 53,171 | 22,063 | 93,727 |
| **Inference** | | | | | | | |
| | Sample model | $12.16 \pm 0.10$ ms
0.22 MiB | $12.23 \pm 0.07$ ms
0.59 MiB | $88.24 \pm 0.01$ ms
0.32 MiB | $88.13 \pm 0.14$ ms
0.41 MiB | $296.76 \pm 0.01$ ms
0.73 MiB | $299.88 \pm 0.01$ ms
0.78 MiB |
| | Log-prob model | $0.76 \pm 0.02$ ms
0.22 MiB | $0.81 \pm 0.02$ ms
0.56 MiB | $1.77 \pm 0.01$ ms
0.34 MiB | $1.88 \pm 0.01$ ms
0.42 MiB | $3.14 \pm 0.01$ ms
0.81 MiB | $3.21 \pm 0.01$ ms
0.85 MiB |
| | Sample parameter | $56.46 \pm 0.18$ ms
0.42 MiB | $65.88 \pm 0.16$ ms
0.76 MiB | $73.51 \pm 0.12$ ms
0.89 MiB | $83.41 \pm 0.16$ ms
0.99 MiB | $109.18 \pm 0.02$ ms
1.53 MiB | $118.93 \pm 0.10$ ms
1.85 MiB |
| | Log-prob parameter | $148.35 \pm 0.05$ ms
2.27 MiB | $159.24 \pm 0.04$ ms
4.12 MiB | $247.92 \pm 0.29$ ms
22.21 MiB | $327.08 \pm 0.07$ ms
39.85 MiB | $576.13 \pm 0.13$ ms
130.18 MiB | $784.72 \pm 0.17$ ms
153.28 MiB |
| **Training (batch size 2048)** | | | | | | | |
| | Forward pass | $7.60 \pm 0.04$ ms
198.14 MiB | $12.84 \pm 0.11$ ms
419.74 MiB | $43.72 \pm 0.41$ ms
387.80 MiB | $54.79 \pm 0.12$ ms
589.05 MiB | $230.92 \pm 0.36$ ms
1.04 GiB | $251.14 \pm 0.66$ ms
1.14 GiB |
| | Backward pass | $24.24 \pm 0.11$ ms
3.71 GiB | $39.39 \pm 0.25$ ms
7.50 GiB | $78.02 \pm 0.47$ ms
9.88 GiB | $111.98 \pm 0.20$ ms
18.95 GiB | $302.63 \pm 0.35$ ms
18.34 GiB | $358.99 \pm 1.98$ ms
35.25 GiB |

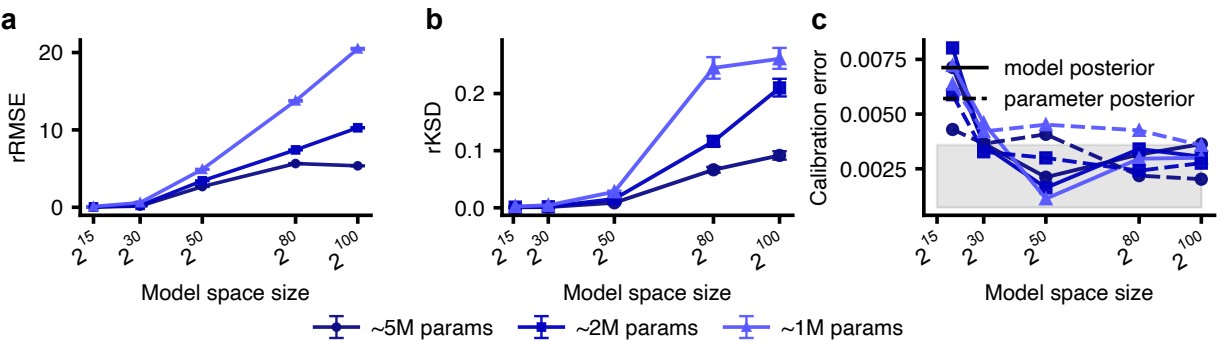

*Figure A.3.* **Evaluation metrics across symbolic regression tasks.** The rRMSE (**a**), rKSD (**b**) and calibration error (**c**) across all considered model spaces (x-axis) and model sizes (colors). The gray shaded area indicates the expected Monte Carlo error if perfectly calibrated. Model sizes refer to `model_dim` $\in \{32, 64, 128\}$.

### A.3. Additional symbolic regression results

For the same example function as in Fig. 2, we show in Fig. A.2 the results for the 1000 observations also used in our main evaluations. We can see that the model posterior is more certain about the number of components (Fig. A.2a right, Fig. A.2b left). Yet, due to large and 'peaked' noise for small $x$, the overall function is not identifiable and, e.g., a square-root approximation to the log is similarly well supported by the data as the true logarithmic component.

For each trained model with $K \in \{15, 30, 50, 80, 100\}$ symbolic components, we evaluated rRMSE, rKSD, and calibration error (Fig. A.3). Both rRMSE and rKSD indicate a very good approximation up to $2^{30}$, with only a minor deviation at $2^{50}$ and a clearer deviation afterwards. Across all $K$, calibration error remains very low and is typically close to the error expected at random. For $K = 15$ we observe larger (but still small) calibration deviations even though rRMSE and rKSD stay low. All in all, this does not imply that models with $K > 50$ are inherently poor: they still yield accurate and calibrated predictions, but they deviate from the Bayes-optimal predictor.

**Limited Training Data**   For the task with $K = 50$ model components, the smallest network completed 553k update steps, whereas the largest network completed roughly 363k. If we assume that each batch corresponds to a new model (i.e., no model is repeated; an upper bound), this amounts to approximately 1.4–2.2 billion simulations used for training. Even under the most generous assumption that each batch contains a distinct model configuration, the smallest network would have explored only about $10^{-4}\%$ of the full model space, with essentially one data point per visited model. Consequently, good performance necessarily relies on *extrapolation* across related models—this is the core benefit of *large-scale amortization*. At the same time, the results suggest that achieving near Bayes-optimal performance requires covering at least a non-trivial fraction of the model space (e.g., for $K = 30$ we observed roughly two simulations per model configuration).

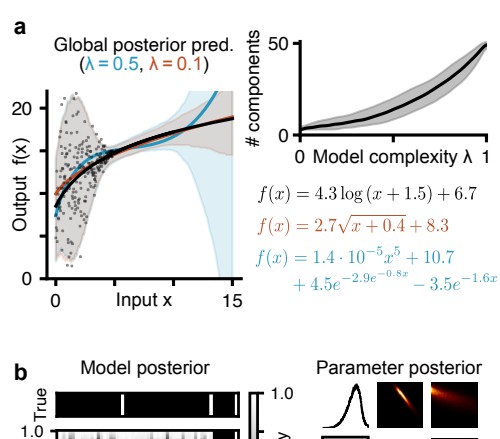

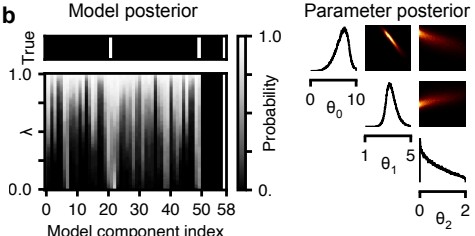

*Figure A.2.* **(a)** Illustration of symbolic regression on the same ground-truth function and noise model as in Fig. 2, but with 1000 observations. **(b)** Associated model and parameter posterior of the simple function (orange), which becomes more constrained compared to Fig. 2 with 10 observations.

**Classifier Perspective**   To assess the model-posterior network from a classification perspective, we treat each candidate model as a "class." Because the full model space is too large for standard multi-class evaluation, we restrict evaluation to a

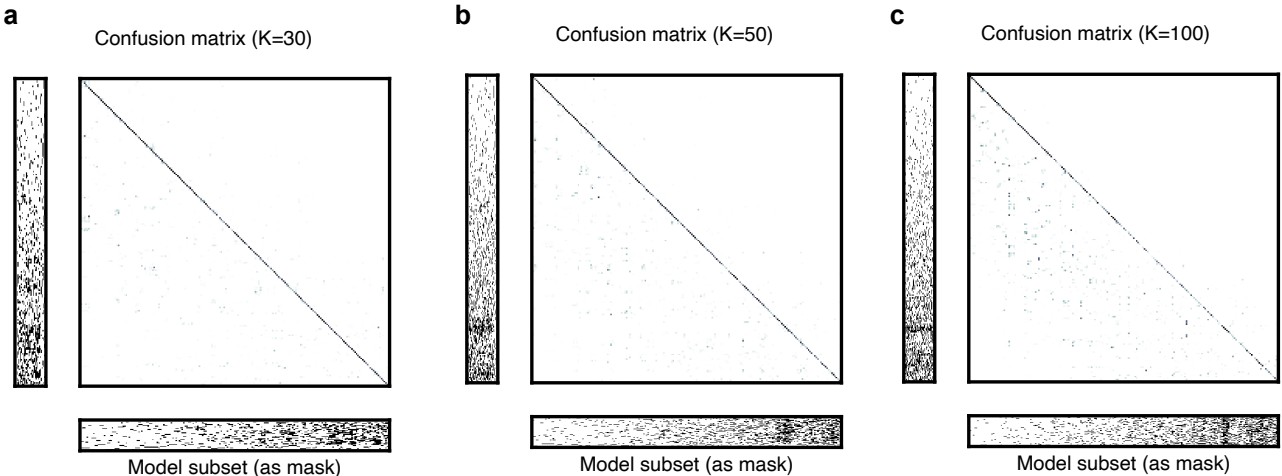

**a** Confusion matrix (K=30)   **b** Confusion matrix (K=50)   **c** Confusion matrix (K=100)

Model subset (as mask)   Model subset (as mask)   Model subset (as mask)

*Figure A.4.* **Confusion matrix for model posterior network.** For a subset of 200 randomly selected models (indicated by their binary mask, x/y axis, grouped by similarity) we show the confusion matrix averaged over random $\lambda$ for the symbolic regression tasks with $K = 30$, $K = 50$, $K = 100$

*Table 3.* Classification metrics for symbolic classification across different settings, as derived from the confusion matrices in Fig. A.4. Note that even a Bayes-optimal classifier will spread mass across multiple models.

| Metric | 15 | 30 | 50 | 80 | 100 |
|---|---|---|---|---|---|
| Accuracy (Top-1) | 0.796 | 0.730 | 0.690 | 0.560 | 0.503 |
| Accuracy (Top-5) | 0.981 | 0.995 | 0.995 | 0.940 | 0.905 |
| Macro precision | 0.813 | 0.776 | 0.741 | 0.647 | 0.597 |
| Macro recall | 0.797 | 0.742 | 0.688 | 0.548 | 0.503 |
| Macro F1 | 0.794 | 0.733 | 0.674 | 0.525 | 0.463 |

random subspace $\mathcal{M}_{200}$. Specifically, we sample 200 models (represented by their binary masks) and define the classifier

$$c(\mathcal{M} \mid \boldsymbol{x}) = \arg \max_{\mathcal{M} \in \mathcal{M}_{200}} q(\mathcal{M} \mid \boldsymbol{x}), \qquad \text{with } \boldsymbol{x} \sim p(\boldsymbol{x} \mid \mathcal{M}).$$

We then evaluate this classifier using standard metrics. First, we compute confusion matrices for $K = 30, 50, 100$ (Fig. A.4). We order models by similarity of their representative masks—i.e., by overlap in components rather than by direct functional equivalence. The resulting matrices are largely diagonal, indicating that the classifier typically recovers the true model, or a closely related one.

From these confusion matrices we derive standard classification metrics (Tab. 3). Top-1 accuracy decreases monotonically with increasing $K$, from 0.796 at $K = 15$ to 0.503 at $K = 100$, with corresponding declines in macro precision, recall, and F1. Yet, low accuracy is expected: as $K$ grows, we have larger estimation error but also the number of plausible model configurations increases and many configurations become observationally near-indistinguishable, rendering exact identification of a single "true" model class increasingly ambiguous.

In this setting, Top-5 accuracy is a more informative measure of posterior concentration than Top-1 accuracy (as, e.g., also employed for ImageNet (Russakovsky et al., 2015)). For $K \in \{15, 30, 50\}$, the true model is almost always contained among the five highest-probability candidates (Top-5 accuracy $\geq 0.981$), indicating that $q(\mathcal{M} \mid \boldsymbol{x})$ concentrates its mass on a very small set of alternatives even when the MAP choice is not always correct. For larger $K$, Top-5 accuracy remains high but declines (to 0.940 at $K = 80$ and 0.905 at $K = 100$). Importantly, this performance is achieved even though the training procedure cannot have covered all classes in $\mathcal{M}$ (and unlikely all the classes in $\mathcal{M}_{200}$), so the classifier must generalize to previously unseen model configurations. Overall, these results suggest that the model-posterior network retains substantial discriminative power at large $K$: while exact Top-1 accuracy is hard (or generally not possible) due to class degeneracy, the true model typically remains among a small set of highly plausible candidates.

**$\lambda$-dependent calibration and autoregressive vs. independent posteriors.** To assess how complexity control affects calibration and to validate the autoregressive (AR) factorization of the model posterior, we evaluate (i) the model-posterior calibration error as a function of $\lambda$ across $K \in \{15, 30, 50, 100\}$, and (ii) an ablation comparing the AR factorization to a fully independent (factorized Bernoulli) approximation (Fig. A.5). The AR posterior remains well calibrated across $\lambda$, whereas the independent approximation systematically fails SBC: it cannot capture the correlations induced by the data and by small-$\lambda$ priors, leading to clear deviations from the diagonal of the rank-CDF. The correlation matrices on the left illustrate that these prior- and data-induced dependencies are substantial, so modelling them via the AR factorization is necessary for calibrated joint model inference.

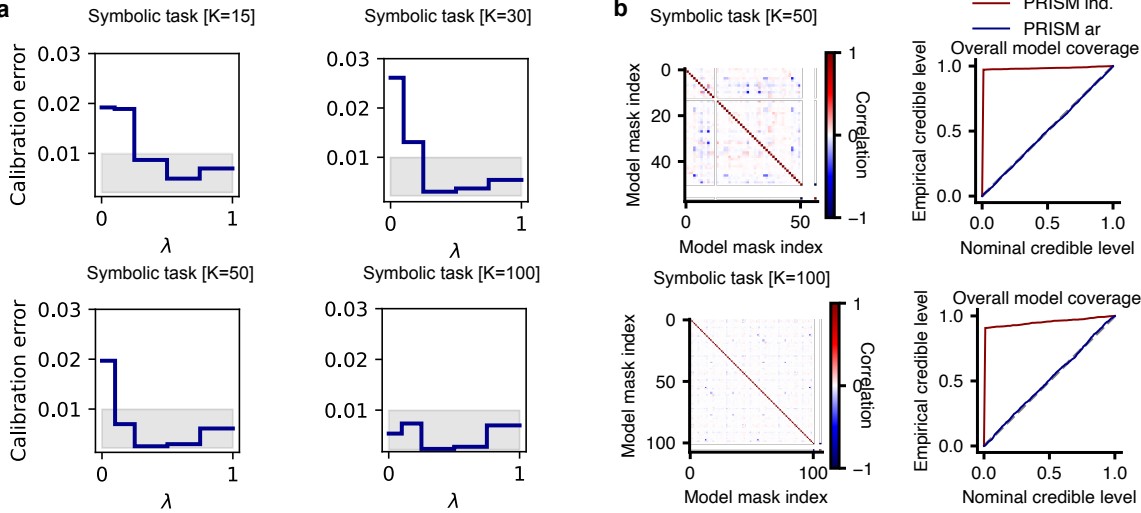

*Figure A.5.* **$\lambda$-dependent calibration and AR vs. independent ablation.** **(a)** Model-posterior calibration error across complexity $\lambda$ for the symbolic regression task with $K = 15, 30, 50, 100$ components. **(b)** *Left:* Exemplary correlation matrices of the model posterior, showing substantial component-component dependencies. *Right:* SBC calibration curves for the *independent* (ind.) vs. *autoregressive* (ar.) model-posterior approximations; the independent factorization fails to capture these correlations and is no longer well calibrated.

## A.4. Additional dMRI results

### A.4.1. EXTENDED B3S INFERENCE PERFORMANCE EVALUATION

We further verify inference performance by evaluating parameter inference on both synthetic datasets and a real dataset (Tab. 4). Notably, while our likelihoods are comparatively "simple", the induced posterior can still be highly complex—even within this constrained model family (Fig. A.8). This contrasts with other applications in simulation-based inference, where the likelihood (i.e. the simulator) is complicated but the associated posterior is often relatively simple (Dax et al., 2021).

In addition to KSD and RMSE, we report the effective sample size (ESS), which provides a proxy for how well approximate samples could be corrected toward the true posterior using, for example, importance sampling (cf. (Dax et al., 2021)). We compute the ESS as follows: Let $w_i = p(\boldsymbol{\theta}_i \mid \boldsymbol{x})/q(\boldsymbol{\theta}_i \mid \boldsymbol{x})$ denote importance weights for samples $\boldsymbol{\theta}_i \sim q(\boldsymbol{\theta} \mid \boldsymbol{x})$. We use the normalized ESS,

$$\text{ESS} = \frac{1}{N} \frac{\left(\sum_{i=1}^{N} w_i\right)^2}{\sum_{i=1}^{N} w_i^2}, \tag{5}$$

where $N$ is the number of samples. On synthetic data, we obtain normalized ESS values of approximately 60% on well-specified simulated data (Tab. 4). The ESS varies with diffusion steps of the sampling routine, but it quickly reaches its plateau with $T = 32$ network evaluations which takes around 100 ms for 1000 posterior samples (Fig. A.6).

We further evaluate the model posterior network under a stricter model-selection perspective. We therefore analyze the confusion matrix (Fig. A.9a). While it is straightforward to separate a Ball from a Stick, distinguishing among Stick-based models is substantially harder when they induce (near-) identical data distributions. This ambiguity also propagates to closely related composite models (e.g. BS vs. S). We attribute this primarily to three factors: First, we employ an informative fraction prior $\boldsymbol{f}$; *a priori*, the fraction associated with the third stick is already close to zero, so "removing" that component

*Table 4.* **Parameter inference performance metrics.** Median and 10th/90th quantiles of performance metrics on synthetic and real data for the effective sample size (ESS), the Kernel Stein discrepancy (KSD) and the predictive rooted mean square error (RMSE). Obtained via the tailored B3S model. Synthetic is over the whole model space, where real is restricted to specific models.

| | Synthetic | | Real (UKB) | | |
| --- | --- | --- | --- | --- | --- |
| Metric | UKB-like | HCP-like | B1S | B2S | B3S |
| ESS | 0.615 [0.098, 0.962] | 0.578 [0.084, 0.979] | 0.177 [0.021, 0.695] | 0.384 [0.053, 0.685] | 0.276 [0.070, 0.621] |
| KSD | 0.304 [0.000, 11.753] | 0.141 [0.000, 11.867] | 0.306 [0.000, 1.546] | 0.389 [0.000, 1.390] | 0.062 [0.000, 0.758] |
| RMSE | 0.020 [0.012, 0.076] | 0.020 [0.012, 0.076] | 0.030 [0.019, 0.054] | 0.023 [0.015, 0.034] | 0.020 [0.014, 0.032] |

has only a minor effect on $p(\boldsymbol{x} \mid \mathcal{M})$. Second, we use a single noise model across a wide range of noise scales (or SNRs). If B2S and B3S are already similar in the noiseless setting, they become virtually indistinguishable at most noise levels, except for very small noise. Third, there are multiple B2S and B1S variants that are not exactly but nearly equivalent in our configuration (differing primarily in the fraction prior). Consequently, the B2S variants must split probability mass among themselves, whereas the B3S model does not face the same internal ambiguity.

To further validate these findings on real data, we compare the approximate model posterior against a ground-truth reference computed via evidence estimates on a random subset of 1000 voxels from UKB (i.e., dMRI data using the UK Biobank acquisition protocol). We estimate the evidence $p(\boldsymbol{x}_o \mid \mathcal{M})$ using an importance sampling estimator based on $q(\boldsymbol{\theta} \mid \mathcal{M}, \boldsymbol{x}_o)$, which yields low-variance estimates in this setting. Concretely, we compute

$$p(\boldsymbol{x}_o \mid \mathcal{M}) \approx \frac{1}{N} \sum_{i=1}^{N} p(\boldsymbol{x}_o \mid \mathcal{M}, \boldsymbol{\theta}_{\mathcal{M}}^i) \frac{p(\boldsymbol{\theta}_{\mathcal{M}} \mid \mathcal{M})}{q(\boldsymbol{\theta}_{\mathcal{M}}^i \mid \mathcal{M}, \boldsymbol{x}_o)}, \qquad \boldsymbol{\theta}_{\mathcal{M}}^i \sim q(\boldsymbol{\theta}_{\mathcal{M}}^i \mid \mathcal{M}, \boldsymbol{x}_o), \qquad (6)$$

which is an unbiased Monte Carlo estimator of the evidence. Evidence estimates using the prior showed high variance, but this estimator with $N = 1024$ samples showed a low enough variance for two independent estimators to closely agree ($R^2 = 0.99$, Fig. A.9b).

As a complementary check, we also consider a biased evidence proxy derived from the model posterior network,

$$p(\boldsymbol{x}_o \mid \mathcal{M}) \propto \frac{q_\phi(\mathcal{M} \mid \boldsymbol{x}_o, \lambda)}{p(\mathcal{M} \mid \lambda)}, \qquad (7)$$

which is likewise well aligned with the unbiased importance-sampling estimates ($R^2 = 0.97$, Fig. A.9c). This demonstrates that models selected via the model-posterior network are well aligned with models selected through classical evidence-based model selection.

Given these evidence estimates, we can assess how well the approximate model posterior probabilities match the corresponding ground-truth model posterior,

$$p(\mathcal{M} \mid \boldsymbol{x}_o) \propto p(\boldsymbol{x}_o \mid \mathcal{M}) p(\mathcal{M} \mid \lambda), \qquad (8)$$

across all $\lambda$. We observe that the total variation distance (equivalently, the $\ell_1$ distance up to a factor $1/2$) remains low overall, but increases for small $\lambda$. This is consistent with model misspecification: for small $\lambda$, the network is biased toward "simple" observations, whereas our earlier results suggest that real data is more "complex" in the sense that more complex models have higher evidence.

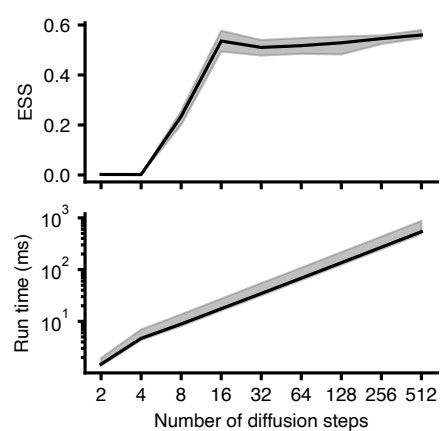

*Figure A.6.* ESS and run times as a function of the number of diffusion steps for 1000 posterior samples for V2 B3S posterior in Fig. 6. Evaluated on an Nvidia H100.

In conclusion, these experiments indicate that our model posterior approximation is accurate, and that the remaining unidentifiability is largely driven by the task configuration (Sec. B.2.4). A straightforward modification to mitigate this issue is to replace the single noise model with multiple noise regimes (e.g. small, medium, and large noise), since identifiability improves at lower noise levels where model-induced distributions are more separable. This is consistent with observations by Manzano-Patron et al. (2025), who report that identifiability is strongest at small noise scales and degrades as noise increases.

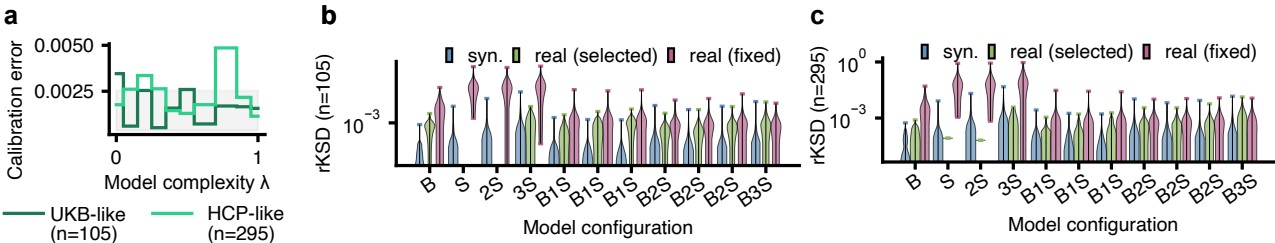

*Figure A.7.* **Calibration for dMRI data. (a)** Model posterior calibration in terms of SBC across $\lambda$ for UKB- and HCP-like data $\rightarrow$ calibration remains close to nominal across complexity levels. **(b)** rKSD on the UKB dataset for synthetic data (true model) and real data, comparing a *fixed* model to the model *selected* by the model-posterior network (same as Fig. 6c) $\rightarrow$ joint selection consistently outperforms fixed-model inference on real data. **(c)** Same as (b) for the HCP dataset $\rightarrow$ HCP shows the same fixed-vs.-selected gap as UKB, with overall similar accuracy.

**Comparison to a fixed-model NSF baseline.** To contextualize PRISM's parameter inference performance against a strong fixed-model SBI estimator, we additionally train a Neural Spline Flow (NSF) baseline with a transformer embedding network, restricted to the B3S model. The total parameter count is approximately 7M: 8 NSF transforms with hidden dimension 256 (as implemented in the `sbi` toolbox (Boelts et al., 2025)) together with a transformer encoder using 4 CLS tokens to summarize the input into a fixed-dimensional summary statistic for the flow (4 layers, model dimension 128, 512-dimensional summary; PyTorch defaults otherwise). The baseline was trained for 72h, matching the dMRI training budget. Detailed compute statistics are reported in Tab. 1 (Sec. A.2.4).

We compare PRISM (conditioned on the B3S model) against this NSF baseline on synthetic UKB-/HCP-like data and real UKB/HCP data (Tab. 5). On synthetic data PRISM is on par with the NSF; on real data PRISM consistently improves over the NSF, in particular in KSD and on the HCP-real subset. We emphasize that PRISM achieves this while *simultaneously* estimating the model posterior and amortizing across the full B3S family (and additional acquisition settings), whereas the NSF baseline is restricted to a single fixed model and acquisition. Standard flow- or diffusion-based posterior estimators are inherently fixed-model methods in this sense and do not scale directly to exponentially many model structures.

*Table 5.* **Parameter inference: PRISM (conditioned on B3S) vs. a fixed-model NSF + Transformer baseline.** Mean and 10th/90th quantiles of the effective sample size (ESS), the Kernel Stein discrepancy (KSD), and the predictive root mean square error (RMSE) on synthetic and real UKB/HCP data. The NSF baseline uses 8 NSF transforms with hidden dimension 256, a transformer encoder with 4 CLS tokens, 4 layers, model dimension 128, and a 512-dimensional summary statistic (PyTorch defaults otherwise; total parameter count $\approx$7M); see Tab. 1 for detailed compute comparison.

| | Synthetic | | Real | |
|---|---|---|---|---|
| Metric | UKB-like [B3S] | HCP-like [B3S] | UKB-real [B3S] | HCP-real [B3S] |
| **PRISM [conditioned on B3S]** | | | | |
| ESS | 0.095 [0.027, 0.179] | 0.098 [0.028, 0.191] | 0.105 [0.028, 0.208] | 0.082 [0.024, 0.173] |
| KSD | 2.256 [0.000, 7.641] | 2.711 [0.000, 11.657] | 2.613 [0.000, 9.648] | 5.728 [0.000, 19.207] |
| RMSE | 0.060 [0.020, 0.132] | 0.060 [0.020, 0.128] | 0.039 [0.022, 0.059] | 0.045 [0.028, 0.064] |
| **NSF + Transformer Embedding [B3S only]** | | | | |
| ESS | 0.093 [0.028, 0.185] | 0.090 [0.028, 0.180] | 0.061 [0.022, 0.123] | 0.041 [0.020, 0.070] |
| KSD | 2.732 [0.000, 10.540] | 4.885 [0.000, 21.037] | 5.198 [0.000, 16.654] | 15.016 [0.000, 66.854] |
| RMSE | 0.061 [0.020, 0.131] | 0.060 [0.020, 0.128] | 0.043 [0.024, 0.064] | 0.048 [0.029, 0.070] |

### A.4.2. EXTENDED MODEL SPACE INFERENCE

While the Ball-3-Stick model space is relatively small, we can extend it by adding alternative compartments (see Sec. B.2, Tab. 11 for a table of considered compartments). Specifically, a 'Stick' (S) can be considered a degenerate (rank-1) special case of a 'Tensor' (T) (Basser et al., 1994) (or of a 'Zeppelin' (Z), which is itself a constrained 'Tensor'). We consider the Ball-Stick-Zeppelin-Tensor (BSZT) space, which contains three copies of each of the S, Z, and T components (i.e.,

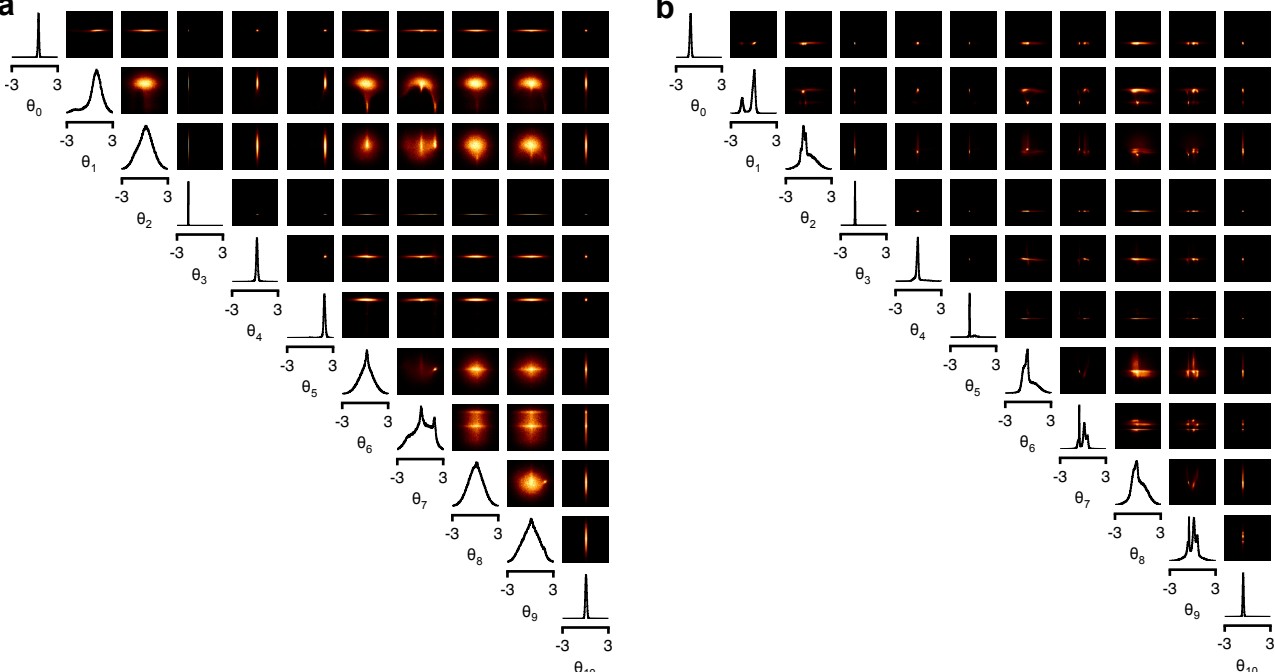

*Figure A.8.* **Parameter posterior for Ball-and-Stick model.** Full parameter posterior distribution conditioned on the B3S model **(a)** for V1 **(b)** for V2 as shown in Fig. 6.

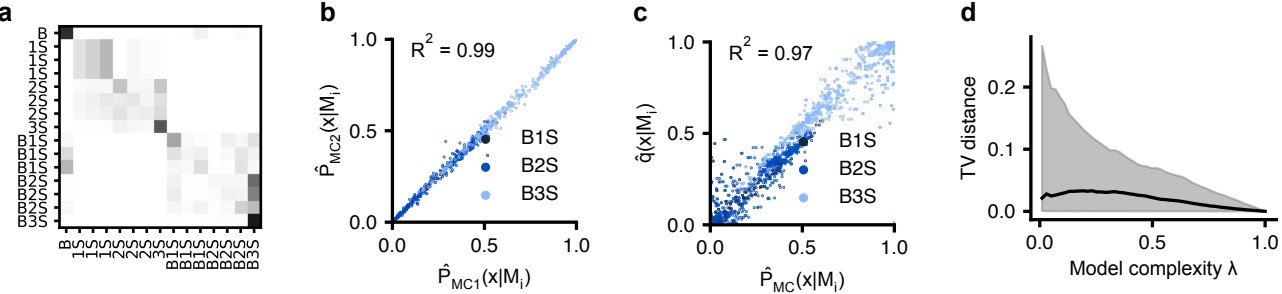

*Figure A.9.* **B3S model selection evaluation.** **(a)** Confusion matrix for all models. **(b)** Alignment between two Monte Carlo (MC) approximations of the evidence. **(c)** Alignment between the approximated evidence obtained from the PRISM model posterior probabilities compared to MC approximations. **(d)** Total variation (TV) distance across different model complexities $\lambda$.

analogous to B3S). We further extend this space to BSZT+conv, which includes additional components based on spherical convolution (Canales-Rodríguez et al., 2015). See Fick et al. (2019) for an overview.

We investigate the performance of models trained on the BSZT and BSZT+conv spaces (see Sec. B.2.5 for details). In contrast to B3S, which can be evaluated using relatively standard approaches (including classification-based analyses (Manzano-Patron et al., 2025)), these model spaces are substantially larger, which makes evaluation more challenging. Additionally, while smaller models can be accurately estimated with the investigated data (i.e., are relatively well-constrained up to symmetries), larger models such as B3T are under-constrained. This is also visible in the associated posteriors, which show a high degree of parameter degeneracy (Fig. A.12).

We first consider simulation-based calibration. Overall, both the model and parameter predictions remain well-calibrated (Fig. A.10a,b, left), although the calibration error is now slightly higher than the level expected under perfect calibration. For the BSZT space, deviations from the expected behavior are largely confined to specific settings (notably HCP acquisitions and a subset of models), whereas the BSZT+conv space exhibits more systematic deviations. We attribute this loss in performance also to computational constraints: simulating dMRI models based on spherical convolution is substantially

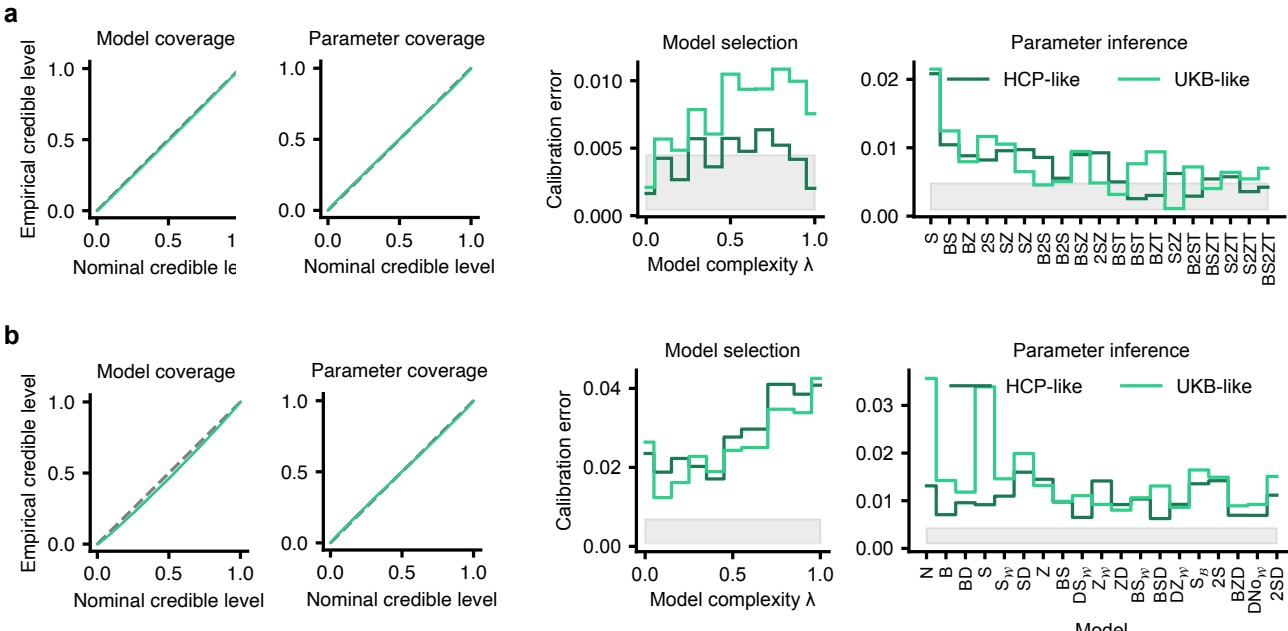

*Figure A.10.* **SBC evaluation for extended dMRI model spaces.** Calibration plot and error (over $\lambda$ and models) for the model trained on **(a)** the BSZT space $\rightarrow$ well calibrated overall, with elevated error confined to HCP-like settings and a subset of models, and **(b)** the BSZT + conv space $\rightarrow$ minor systematic deviations attributable to the higher simulation cost of spherical-convolution components, which limits coverage of the enlarged model space during training.

more expensive (even with an additional GPU), which limits the effective coverage of the enlarged model space during training. We further corroborate these trends using RMSE, ESS, and KSD (Tab. 6, 7).

We additionally evaluate the model posterior network from a classification perspective on a randomly sampled model subspace (Fig. A.11). This is inherently a difficult task for two reasons. First, many "complex" models can represent "simpler" ones by setting the corresponding fraction parameters to (near) zero. Second, in this configuration we include genuinely equivalent models (since we removed informative fraction priors), and, more broadly, Stick, Zeppelin, and Ball components can be viewed as constrained parameterizations of the Tensor model.

Despite these challenges, the confusion matrices remain largely concentrated along the diagonal, with off-diagonal mass primarily distributed among equivalent or closely related models, as expected. The top-5 accuracy is $94\%$ for the BSZT space and $98\%$ for the BSZT+conv space (Tab. 8; the latter increase can be primarily attributed to increased class diversity within the subspace). We emphasize that this improvement should not be interpreted as a strict dominance of BSZT+conv; rather, it is at least partly explained by differences in the evaluated subset, which is more diverse in the BSZT+conv case and therefore easier to separate in a top-$k$ sense.

*Table 6.* **Parameter inference performance metrics.** Median and 10th/90th quantiles of performance metrics on synthetic and real data for the effective sample size (ESS), the Kernel Stein discrepancy (KSD) and the predictive rooted mean square error (RMSE) for the BSZT space model. Synthetic is over the whole model space, where real is restricted to specific models.

| | Synthetic | | Real (UKB) | | |
|---|---|---|---|---|---|
| Metric | UKB-like | HCP-like | B3S | B3Z | B3T |
| ESS | 0.109 [0.026, 0.962] | 0.081 [0.028, 0.986] | 0.133 [0.045, 0.302] | 0.131 [0.040, 0.398] | 0.165 [0.061, 0.366] |
| KSD | 1.144 [0.000, 7.086] | 1.776 [0.000, 10.411] | 0.004 [0.000, 0.107] | 0.000 [0.000, 0.014] | 0.000 [0.000, 0.000] |
| RMSE | 0.019 [0.011, 0.072] | 0.019 [0.011, 0.072] | 0.020 [0.012, 0.032] | 0.020 [0.012, 0.032] | 0.018 [0.012, 0.030] |

### A.4.3. DMRI FOD RESULTS

Joint model–parameter inference assigns, for each voxel, both a model $\mathcal{M}$ and a corresponding parameter vector $\boldsymbol{\theta}_{\mathcal{M}}$. This flexibility is a major strength, but it also complicates standard per-parameter analyses: parameters are model-specific, and their interpretation can change across models. For example, in B3S one can extract inferred diffusion "orientations" by

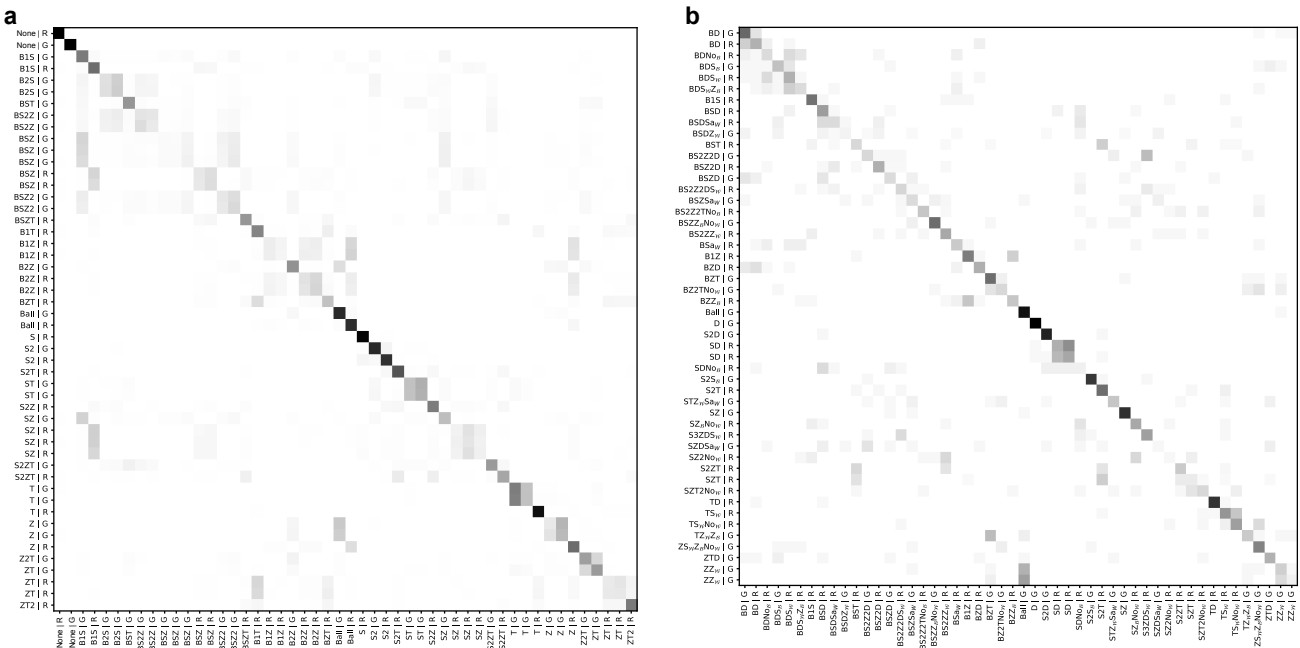

*Figure A.11.* **Confusion matrices for PRISM on dMRI data with full model family.** **(a)** on a 50 model subspace for the BSZT space model **(b)** on a 50 model subspace for the BSZT+conv space model

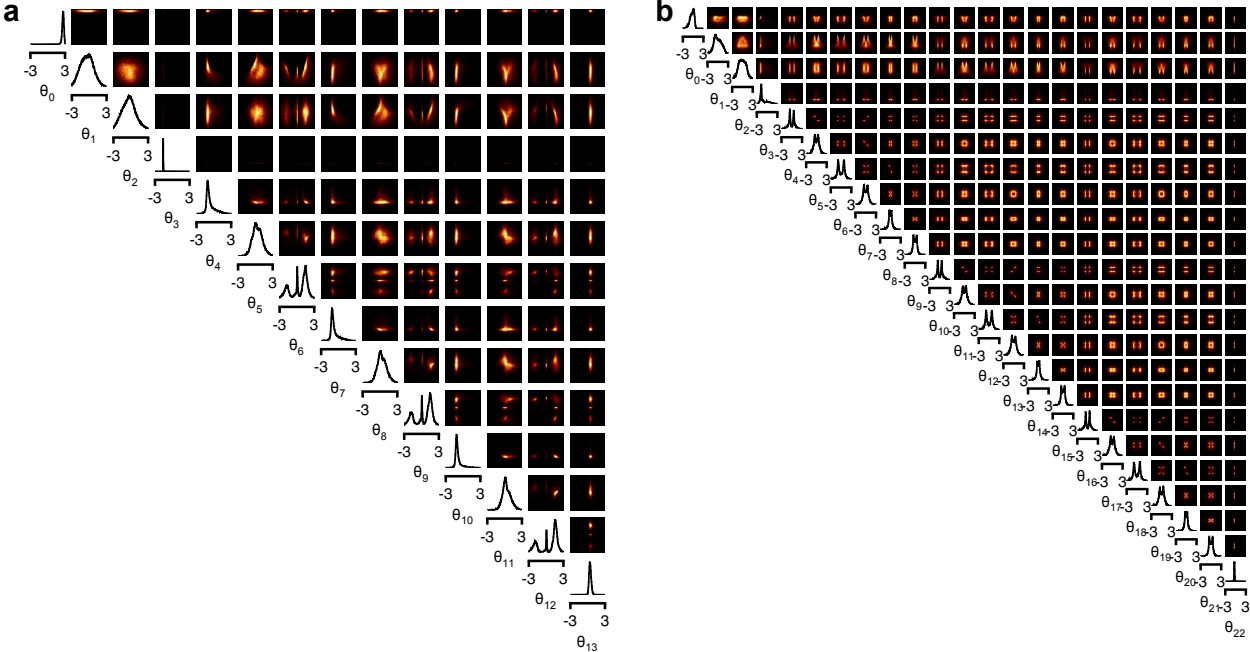

*Figure A.12.* **Parameter posterior for dMRI.** Parameter posterior for the two voxels shown in Fig. 6 as estimated by the BSZT-space model. **(a)** Posterior for the B3S model (different from previous due to unshared diffusivity and uninformative fraction prior). **(b)** Posterior for the B3T model showing large degeneracy due to symmetries and nonidentifiability on current datasets.

*Table 7.* **Parameter inference performance metrics.** Median and 10th/90th quantiles of performance metrics on synthetic and real data for the effective sample size (ESS), the Kernel Stein discrepancy (KSD) and the predictive rooted mean square error (RMSE) for the BSZT + conv model. Synthetic is over the whole model space, where real is restricted to specific models.

| | Synthetic | | Real (UKB) | | |
|---|---|---|---|---|---|
| Metric | UKB-like | HCP-like | $BS_{\mathcal{W}}S_{\mathcal{B}}$ | $No_{\mathcal{B}}$ | $2TS_{\mathcal{W}}S_{\mathcal{B}}No_{\mathcal{W}}$ |
| ESS | 0.073 [0.025, 0.453] | 0.083 [0.021, 0.433] | 0.109 [0.028, 0.464] | 0.053 [0.020, 0.335] | 0.135 [0.043, 0.410] |
| KSD | 3.873 [0.671, 22.23] | 6.597 [0.922, 35.49] | 0.001 [0.000, 0.016] | 1.000 [0.089, 4.306] | 0.000 [0.000, 0.000] |
| RMSE | 0.021 [0.013, 0.072] | 0.019 [0.012, 0.073] | 0.022 [0.013, 0.049] | 0.028 [0.014, 0.065] | 0.018 [0.012, 0.031] |

inspecting the directions of the Stick compartments together with their associated fractions, but an analogous procedure is not directly comparable across other model classes.

To enable a consistent analysis across models, we instead map each posterior sample $(\theta_{\mathcal{M}}, \mathcal{M})$ to a universal representation: the Orientation Distribution Function (ODF). The ODF is a spherical distribution that provides a common summary of the orientation content implied by any model in our family. We detail this conversion in Sec. B.2. This representation allows us to visualize and compare joint (or model-averaged) posterior distributions in a single object that captures both the predicted orientation structure and its associated uncertainty.

For a slice of HCP data, we compare the fODF predicted by the baseline method Rumba (Canales-Rodríguez et al., 2015) (maximum-likelihood spherical deconvolution with anatomical priors) against posterior-predictive fODFs obtained from B3S, B3T, and the BSZT-space model average. Overall, we observe substantial qualitative agreement in the dominant diffusion directions across methods on this dataset, while the associated uncertainty differs markedly between approaches. In particular, PRISM yields uncertainty estimates that vary spatially and across configurations, reflecting both measurement noise and model ambiguity in a principled way.

A key limitation is that determining the "most faithful" fiber orientation in vivo is itself an open problem, and a definitive evaluation of orientation accuracy is beyond the scope of this work. Consequently, agreement or disagreement in fODF uncertainty should not be over-interpreted as direct evidence of greater anatomical correctness. What we can assess reliably, however, is statistical accuracy relative to the observed dMRI signal. Using this criterion, we find that posterior samples produced by PRISM are better aligned with the observed data than those from the baseline methods, indicating that the inferred models are better supported by the measurements (Fig. 8).

*Table 8.* Classification metrics across BSZT spaces (macro-averaged).

| **Metric** | **B3S space** | **BSZT space** | **BSZT + conv space** |
|---|---|---|---|
| Accuracy (Top-1, Macro) | 0.351 | 0.428 | 0.422 |
| Accuracy (Top-5, Macro) | 0.933 | 0.940 | 0.980 |
| Macro precision | 0.362 | 0.404 | 0.420 |
| Macro recall | 0.351 | 0.428 | 0.422 |
| Macro F1 | 0.337 | 0.403 | 0.408 |

### A.4.4. MAP ESTIMATION: BEST-OF-$N$, TEMPERED SAMPLING, AND BEAM SEARCH

Recovering the discrete MAP $\mathcal{M}^\star = \arg\max_{\mathcal{M}} q(\mathcal{M} \mid x, \lambda)$ over combinatorial model spaces is non-trivial and is not the main contribution of this work; the primary output of PRISM is the joint posterior $q(\mathcal{M}, \theta \mid x, \lambda)$ and Bayesian model averaging derived from it. Nevertheless, MAP-style decoding is a useful complement for downstream model selection, and we therefore compare standard decoding strategies for the autoregressive model posterior: (i) best-of-$N$ sampling (BoN, $N = 128$), (ii) BoN with tempered autoregressive sampling at temperatures $\{1.0, 0.5, 0.1\}$, and (iii) beam search with beam widths $\{16, 32\}$. Note that this is distinct from posterior sampling: the goal here is to recover the most likely model, not to draw representative samples from the posterior.

We evaluate these decoders on the BSZT model space ($> 4 \cdot 10^3$ models) and the BSZT+ conv model space ($> 2 \cdot 10^6$ models), measuring the gap to a brute-force MAP reference. For each method we report $|\log q(\hat{\mathcal{M}}) - \log q(\mathcal{M}^\star_{\text{ref}})|$ and the success rate (fraction of cases with gap below $10^{-3}$, Fig. A.14). Tempered BoN consistently improves over plain BoN and is competitive across both spaces. Beam search becomes increasingly useful in the larger BSZT+ conv space where naive sampling rarely lands on the modal model; tempered BoN with low temperature performs nearly as well. Overall, this

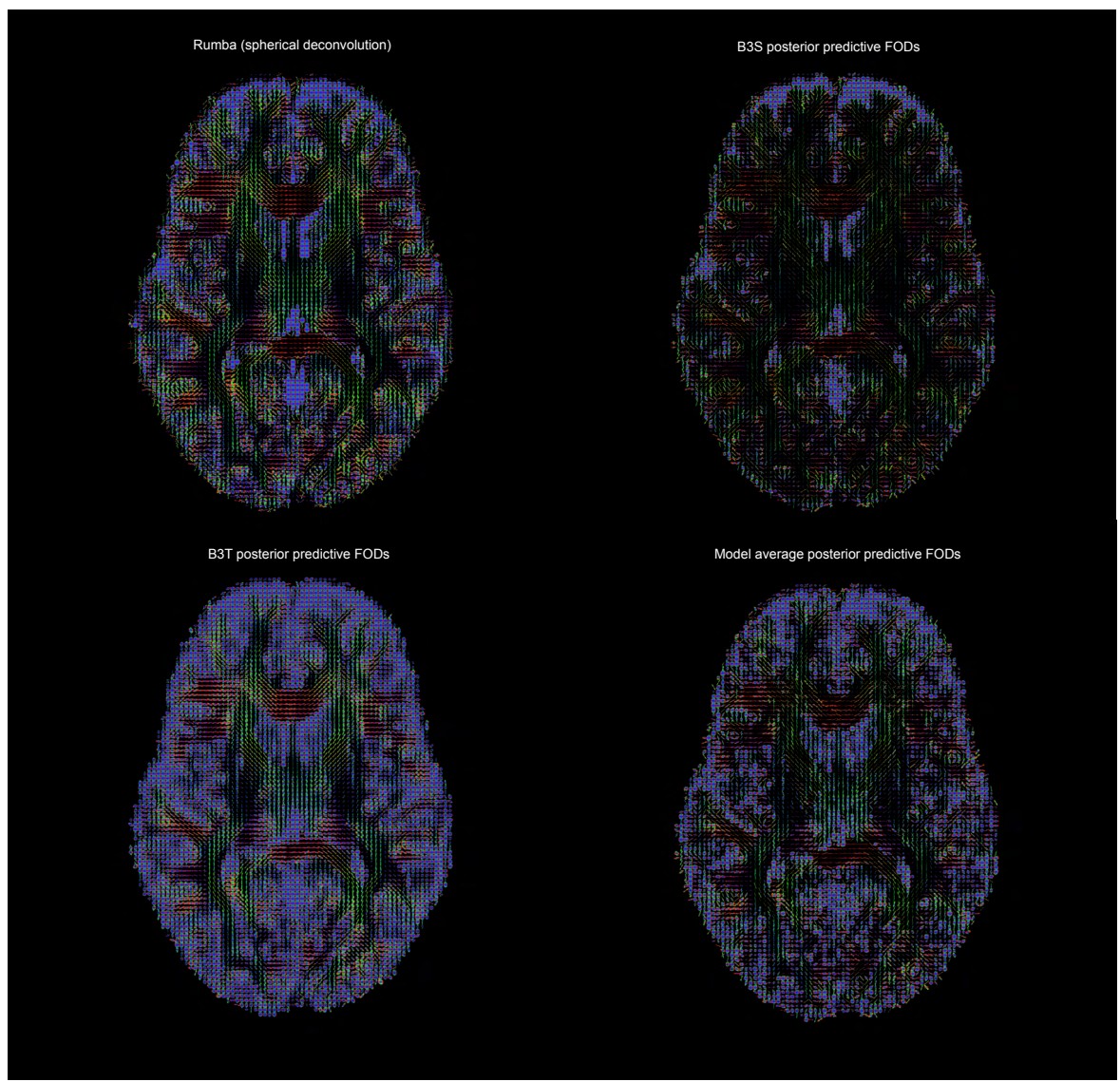

*Figure A.13.* Posterior predictive fiber orientation distributions for a classical spherical-deconvolution-based method (Rumba, (Canales-Rodríguez et al., 2015), as implemented in DIPY (Garyfallidis et al., 2014)), under the B3S model, under B3T, or considering the **global** posterior predictive FOD averaged over all possible tensor-based models.

confirms that MAP recovery is harder than posterior sampling, but tractable with standard decoding heuristics.

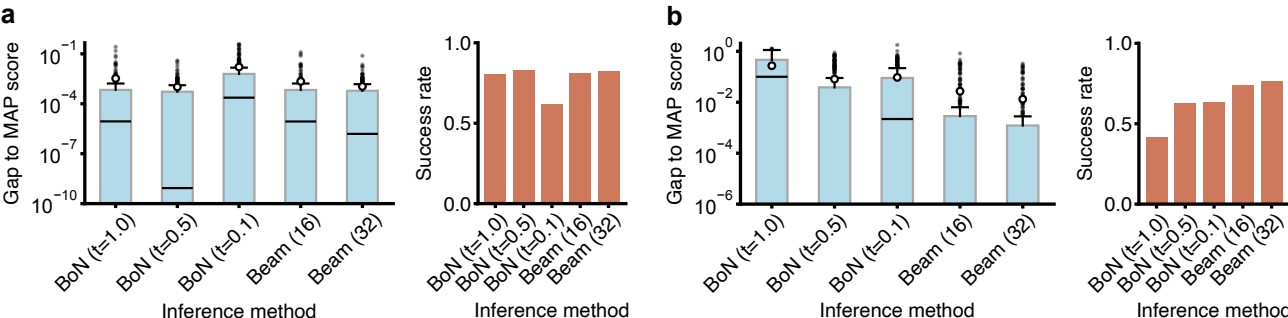

*Figure A.14.* **Comparison of MAP estimation methods.** Best-of-$N$ sampling (BoN, $N = 128$) with tempered autoregressive sampling $(1/t)$, and beam search with beam widths 16 and 32. **(a)** Model selection in the BSZT model space ($> 4{\cdot}10^3$ models). **(b)** Model selection in the BSZT+ conv model space ($> 2{\cdot}10^6$ models). *Left:* Gap between estimated MAP $\hat{\mathcal{M}}$ and the brute-force reference $\mathcal{M}^\star_{\text{ref}}$, $|\log q(\hat{\mathcal{M}}) - \log q(\mathcal{M}^\star_{\text{ref}})|$. The reference MAP is obtained by brute-force enumeration. *Right:* Success rate, i.e. the fraction of cases for which the gap is below $10^{-3}$.

### A.4.5. ADDITIONAL MODEL SELECTION RESULTS

Model selection is performed on a (prior-weighted) evidential basis, i.e. by choosing the model with the largest marginal likelihood (Bayesian evidence). In practice, we consider two complementary procedures:

(i) *Model selection (within a user-specified subspace).* Given a user-selected subspace $\mathcal{M}_{\text{sub}} \subset \mathcal{M}$, we select

$$\mathcal{M}^* \;=\; \arg\max_{\mathcal{M}\in\mathcal{M}_{\text{sub}}} q(\mathcal{M} \mid \boldsymbol{x}_o). \tag{9}$$

For a uniform prior ($\lambda = 1.0$), this is equivalent to selecting the model with maximal Bayesian evidence. We solve this problem exactly by evaluating $q(\mathcal{M} \mid \boldsymbol{x}_o)$ for every $\mathcal{M} \in \mathcal{M}_{\text{sub}}$ and returning the maximizer.

(ii) *Model discovery followed by model selection.* We first solve the global optimization problem

$$\mathcal{M}^* \;=\; \arg\max_{\mathcal{M}\in\mathcal{M}} q(\mathcal{M} \mid \boldsymbol{x}_o). \tag{10}$$

Since this is a discrete optimization over a large model space, we approximate it by drawing 100 samples from the model posterior and selecting the model with the highest posterior probability among the sampled set. This "discovery" step can yield many distinct models across the brain; to obtain a compact and interpretable set, we restrict attention to the 10 most frequently selected models, denoted $\mathcal{M}_{\text{top10}}$. We then perform voxel-wise selection within $\mathcal{M}_{\text{top10}}$ using procedure (i).

As in the symbolic regression task, the model-complexity prior strongly influences selection. When restricting model comparison to the B(1/2/3)(S/Z/T) subset, more complex models become less probable as $\lambda$ decreases. However, the rate of this decline depends on the acquisition: for HCP data, complex models remain competitive over a wider range of $\lambda$ (Fig. A.15**a**).

Beyond such constrained comparisons, the learned model posterior enables exploratory analysis over the full combinatorial space and highlights high-probability component combinations. Notably, when selecting over the full BSZT space, combinations such as BZT, BT, or BZ consistently rank highest across all $\lambda$. This suggests that models that dominate within the restricted B(1/2/3)(S/Z/T) subset (e.g., B3T) can often be replaced by simpler multi-component alternatives once the search space is expanded (Fig. A.15**b**). Expanding further to the BSZT + conv space shifts the posterior mass again: Watson- or Bingham-stick variants combined with Balls or Tensor components (and NODDI-style formulations) become substantially more dominant (Fig. A.15**b**). These shifts underscore that model-selection outcomes are conditional on the candidate set: there may always exist a better model outside the chosen search space, so claims about the "best" model should be framed relative to the models considered.

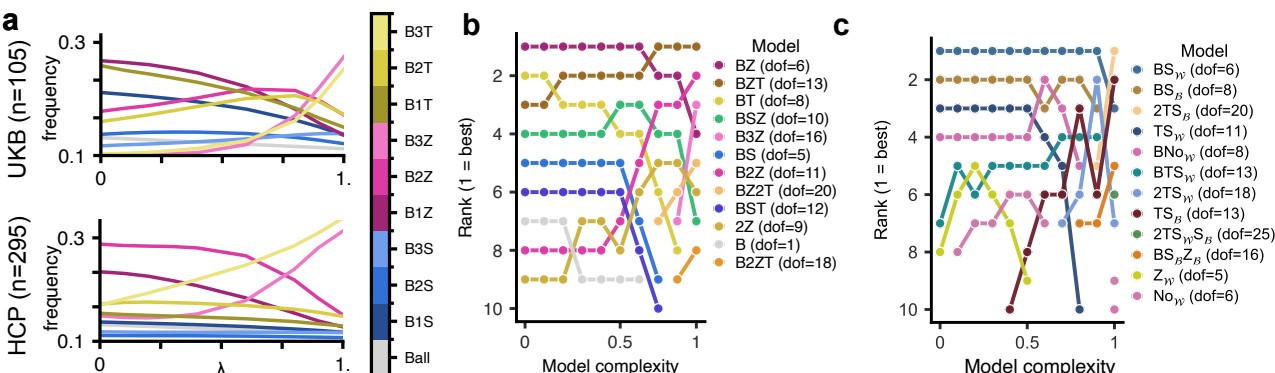

*Figure A.15.* **Model selection results on extended model spaces. (a)** Model-selection frequencies if constrained to a set of B(1/2/3)(S/Z/T) models for the UKB and HCP datasets. **(b)** Top-10 ranking of the whole BSZT space across $\lambda$ for the UKB dataset. **(c)** Top-10 ranking of the BSZT+conv space across $\lambda$ for the UKB dataset.

Overall, the posterior concentrates on multi-compartment combinations that better explain the diffusion MRI signal, consistent with prior cross-validation studies (Panagiotaki et al., 2012; Ferizi et al., 2014; 2015; 2017). However, a detailed analysis remains for future work. While the selected models are consistent with the observed data, to answer domain-specific questions it is better to constrain the spaces/priors so as to compare "appropriate" models for the question at hand.

# B. Problem specifications

## B.1. Symbolic regression problem specifications

We list all components in Tab. 10 together with their naming. For $K = 15$, we used the exact setting reported in Schröder & Macke (2024) with noise models 'NoiseObserver', 'NoiseIncreasing', 'NoiseDecreasing', 'NoiseQuadratic', and 'NoiseQuadraticDecreasing'; otherwise, we used all noise models.

Note that we allow for repeated components. For all settings investigated, the symbolic base functions are given by Tab. 9.

*Table 9.* Explicit listing of base functions (see Tab. 10 for definitions) in order, i.e., mask index $i$ corresponds to the $i$-th base function listed.

| Setting | Basis functions |
|---|---|
| $K$=15 | Linear, Linear, Quadratic, ShiftedSquare, Cubic, Sinusoidal, Cosinusoidal, ConstantWide, ConstantPositive, TanhRight, TanhLeft, GaussianBump, GaussianWide, RampUp, RampDown |
| $K$=30 | Linear, Quadratic, ShiftedSquare, Cubic, Sinusoidal, Cosinusoidal, ConstantWide, ConstantPositive, TanhRight, TanhLeft, TanhCentered, GaussianBump, GaussianWide, RampUp, RampDown, QuarticScaled, QuinticScaled, SinusoidalPhase, CosinusoidalPhase, ExponentialDecay, SaturatingExponential, Logarithmic, SquareRoot, Reciprocal, AbsoluteValue, InverseQuadratic, Lorentzian, Sigmoid, DampedSinusoidal, DampedCosinusoidal |
| $K$=50 | Linear, Quadratic, ShiftedSquare, Cubic, Sinusoidal, Cosinusoidal, ConstantWide, ConstantPositive, TanhRight, TanhLeft, TanhCentered, GaussianBump, GaussianWide, RampUp, RampDown, QuarticScaled, QuinticScaled, SinusoidalPhase, CosinusoidalPhase, ExponentialDecay, SaturatingExponential, Logarithmic, SquareRoot, Reciprocal, AbsoluteValue, InverseQuadratic, Lorentzian, Sigmoid, DampedSinusoidal, DampedCosinusoidal, ExponentialGrowth, PowerLawDecay, ArctangentStep, HyperbolicSecant, SincDecay, AbsoluteSinusoidal, RectifiedLinear, Softplus, Gompertz, LinearFractional, SineSquared, TriangularBump, Linear, Quadratic, ShiftedSquare, Cubic, Sinusoidal, Cosinusoidal, ConstantWide, ConstantPositive |
| $K$=80 | Linear, Quadratic, ShiftedSquare, Cubic, Sinusoidal, Cosinusoidal, ConstantWide, ConstantPositive, TanhRight, TanhLeft, TanhCentered, GaussianBump, GaussianWide, RampUp, RampDown, QuarticScaled, QuinticScaled, SinusoidalPhase, CosinusoidalPhase, ExponentialDecay, SaturatingExponential, Logarithmic, SquareRoot, Reciprocal, AbsoluteValue, InverseQuadratic, Lorentzian, Sigmoid, DampedSinusoidal, DampedCosinusoidal, ExponentialGrowth, PowerLawDecay, ArctangentStep, HyperbolicSecant, SincDecay, AbsoluteSinusoidal, RectifiedLinear, Softplus, Gompertz, LinearFractional, SineSquared, TriangularBump, Linear, Quadratic, ShiftedSquare, Cubic, Sinusoidal, Cosinusoidal, ConstantWide, ConstantPositive, TanhRight, TanhLeft, TanhCentered, GaussianBump, GaussianWide, RampUp, RampDown, QuarticScaled, QuinticScaled, SinusoidalPhase, CosinusoidalPhase, ExponentialDecay, SaturatingExponential, Logarithmic, SquareRoot, Reciprocal, AbsoluteValue, InverseQuadratic, Lorentzian, Sigmoid, DampedSinusoidal, DampedCosinusoidal, ExponentialGrowth, PowerLawDecay, ArctangentStep, HyperbolicSecant, SincDecay, AbsoluteSinusoidal, RectifiedLinear, Softplus |

| Setting | Basis functions |
|---------|-----------------|
| $K=100$ | Linear, Quadratic, ShiftedSquare, Cubic, Sinusoidal, Cosinusoidal, ConstantWide, ConstantPositive, TanhRight, TanhLeft, TanhCentered, GaussianBump, GaussianWide, RampUp, RampDown, QuarticScaled, QuinticScaled, SinusoidalPhase, CosinusoidalPhase, ExponentialDecay, SaturatingExponential, Logarithmic, SquareRoot, Reciprocal, AbsoluteValue, InverseQuadratic, Lorentzian, Sigmoid, DampedSinusoidal, DampedCosinusoidal, ExponentialGrowth, PowerLawDecay, ArctangentStep, HyperbolicSecant, SincDecay, AbsoluteSinusoidal, RectifiedLinear, Softplus, Gompertz, LinearFractional, SineSquared, TriangularBump, Linear, Quadratic, ShiftedSquare, Cubic, Sinusoidal, Cosinusoidal, ConstantWide, ConstantPositive, TanhRight, TanhLeft, TanhCentered, GaussianBump, GaussianWide, RampUp, RampDown, QuarticScaled, QuinticScaled, SinusoidalPhase, CosinusoidalPhase, ExponentialDecay, SaturatingExponential, Logarithmic, SquareRoot, Reciprocal, AbsoluteValue, InverseQuadratic, Lorentzian, Sigmoid, DampedSinusoidal, DampedCosinusoidal, ExponentialGrowth, PowerLawDecay, ArctangentStep, HyperbolicSecant, SincDecay, AbsoluteSinusoidal, RectifiedLinear, Softplus, Gompertz, LinearFractional, SineSquared, TriangularBump, Linear, Quadratic, ShiftedSquare, Cubic, Sinusoidal, Cosinusoidal, ConstantWide, ConstantPositive, TanhRight, TanhLeft, TanhCentered, GaussianBump, GaussianWide, RampUp, RampDown, QuarticScaled |

## B.2. dMRI problem specifications

### B.2.1. BACKGROUND

Diffusion MRI (dMRI) modelling is an *inverse problem*: given diffusion-encoding settings and noisy magnitude measurements, we seek latent tissue parameters that determine the expected signal attenuation. In the pulsed-gradient spin-echo (PGSE) experiment, the encoding is conveniently summarized by the $b$-tensor (or, in the common single-direction case, by a scalar $b$ and a unit direction $\mathbf{g} \in \mathbb{S}^2$) (Stejskal & Tanner, 1965; Mattiello et al., 1997).

A useful starting point is to model water motion as Brownian diffusion. If spin displacements over the effective diffusion time satisfy a Gaussian law (as for free Brownian motion, or as an approximation for *hindered* diffusion in tissue at the voxel scale), then the displacement $\Delta\mathbf{x}$ has density

$$p(\Delta\mathbf{x}) = \mathcal{N}\big(\mathbf{0},\, 2\Delta\,\mathbf{D}\big),$$

with diffusion time $\Delta$ and a positive definite diffusion tensor $\mathbf{D} \succ 0$. In PGSE, the normalized signal can be written (under narrow-pulse and Gaussian phase approximations) as the characteristic function of the displacement distribution evaluated at the encoding wavevector,

$$\frac{S}{S_0} \approx \mathbb{E}\Big[e^{i\,\mathbf{q}^\top \Delta\mathbf{x}}\Big].$$

For Gaussian $\Delta\mathbf{x}$ this expectation is available in closed form and yields a mono-exponential attenuation in the encoding strength. In the single-direction case one obtains

$$\frac{S(b, \mathbf{g})}{S_0} \approx \exp\big(-b\,D_{\mathrm{app}}(\mathbf{g})\big), \qquad D_{\mathrm{app}}(\mathbf{g}) = \mathbf{g}^\top \mathbf{D}\,\mathbf{g}.$$

This second-order parameterization defines diffusion tensor imaging (DTI) (Basser et al., 1994),

$$\frac{S(b, \mathbf{g})}{S_0} = \exp\big(-b\,\mathbf{g}^\top \mathbf{D}\mathbf{g}\big), \qquad \mathbf{D} \succ 0.$$

Mathematically, DTI assumes that within each voxel the ensemble displacement is adequately summarized by a *single* multivariate Gaussian (equivalently, by its covariance), so that direction-dependent signal decay is fully captured by the quadratic form $\mathbf{g}^\top \mathbf{D}\mathbf{g}$. DTI provides a low-dimensional description of anisotropy and a dominant local orientation, but it is not expressive enough when multiple micro-environments and/or multiple fiber orientations contribute within the same voxel, in which case the voxel-scale displacement distribution is better described as non-Gaussian or multi-compartment (Alexander et al., 2019; Novikov, 2021).

**Orientation distributions and spherical convolution.** Many forward models describe a *single-fiber* response conditional on an orientation $\mathbf{n} \in \mathbb{S}^2$. In the same spirit as DTI—where hindered diffusion within a voxel is approximated by a single Gaussian displacement (equivalently, a single diffusion tensor)—a single-fiber kernel can be seen as a locally oriented response. If a voxel contains dispersion and/or multiple fiber populations, a single Gaussian (single tensor) is no longer adequate, and the signal is better modeled as an average over orientations. This is achieved by integrating the single-fiber kernel against an orientation distribution function (ODF). When the kernel depends on $\mathbf{n}$ only through the inner product $\mathbf{g}^\top \mathbf{n}$, this average becomes a spherical convolution:

$$g(b, \mathbf{g}) = \int_{\mathbb{S}^2} g_{\mathrm{sf}}(b, \mathbf{g}^\top \mathbf{n} \mid \theta) \, p(\mathbf{n} \mid \phi) \, d\mathbf{n},$$

with a parametric ODF $p(\mathbf{n} \mid \phi)$ (e.g. Watson or Bingham) or a nonparametric ODF as in spherical deconvolution (Tournier et al., 2007). Dispersion-aware models can improve orientation estimates, but add degrees of freedom and can increase sensitivity to acquisition design and regularization (Behrens et al., 2007; Jbabdi et al., 2012).

**Why model choice matters.** In practice, model choice should be guided by anatomy and acquisition. The compartment family and priors should reflect plausible voxel-scale structure (e.g. single bundle vs. crossings; dispersed vs. tightly aligned orientations; presence/absence of free-water-like signal) and be supported by the available $b$-values, SNR, and gradient sampling. Without such constraints, flexible mixtures may fit the signal while implying implausible tissue configurations, which can propagate to downstream fiber reconstruction and tractography (Thomas et al., 2014; Ferizi et al., 2014). Accordingly, it is often useful to constrain inference via explicit priors, structured model selection, and sensitivity checks across plausible compartment families (Panagiotaki et al., 2012; Alexander et al., 2019; Jelescu & Budde, 2017). Toolkits such as DMIPY expose these kernels as composable modules, enabling rapid specification of multi-compartment mixtures, dispersed kernels, and acquisition-dependent forward operators (Fick et al., 2019). Other libraries such as DIPY (Garyfallidis et al., 2014) provide a collection of established, easy-to-use pipelines (with limited but practical tuning, e.g. tissue-specific options for white vs. grey matter). Widely used packages such as FSL provide end-to-end workflows for diffusion processing and modelling. In particular, BedpostX performs Bayesian estimation via MCMC and uses shrinkage priors (ARD) to discourage unsupported fibre populations, helping stabilize inference under typical acquisitions.

B.2.2. IMPLEMENTATION DETAILS (TABLE 11).

To connect the components in Table 11 within a single generative model, we use a *multi-compartment* formulation in which the normalized diffusion signal is a convex mixture of $K+1$ candidate signal compartments and one noise model selected from a small family:

$$\frac{S(b, \vec{\mathbf{b}} \mid \boldsymbol{\Theta})}{S_0} = \sum_{k=0}^{K} M_k \, f_k \, g_k\left(b, \vec{\mathbf{b}} \mid \boldsymbol{\theta}_k\right) \qquad S_o \sim p\left(\frac{S(b, \vec{\mathbf{b}} \mid \boldsymbol{\Theta})}{S_0} \mid \widetilde{M}_m, \boldsymbol{\theta}_m\right) \tag{11}$$

Each $g_k(b, \vec{\mathbf{b}} \mid \boldsymbol{\theta}_k)$ is a compartment kernel from the library in Table 11 (e.g., Ball, Stick, Zeppelin, Tensor, and dispersed/convolutional variants such as Watson/Bingham sticks and zeppelins), with compartment-specific parameters $\boldsymbol{\theta}_k$ (diffusivities, orientations, dispersion parameters, etc.). The weights $\mathbf{f} = (f_0, \ldots, f_K)$ are *fractions* constrained to the simplex $f_k \geq 0$, $\sum_{k=0}^{K} f_k M_k = 1$, which we encode using a Dirichlet prior, $\mathbf{f} \sim \mathrm{Dirichlet}(\boldsymbol{\alpha})$.

The binary indicators $M_k \in \{0, 1\}$ implement *structural model selection* by turning candidate compartments on/off, while the noise indicators $\widetilde{M}_m \in \{0, 1\}$ select among candidate noise models $\epsilon_m$ (e.g., Gaussian or Rician magnitude noise; Table 11). Noise models are treated as *mutually exclusive*; we enforce $\sum_{m=0}^{M-1} \widetilde{M}_m = 1$ so that exactly one noise model is active. Collecting all unknowns yields $\boldsymbol{\Theta} = \left(\mathbf{f}, \{M_k, \boldsymbol{\theta}_k\}_{k=0}^{K}, \{\widetilde{M}_m, \boldsymbol{\theta}_m^{(\epsilon)}\}_{m=0}^{M-1}\right)$, and the priors over $\boldsymbol{\theta}_k$ and $\boldsymbol{\theta}_m^{(\epsilon)}$ are taken directly from Table 11.

**Parametric spherical convolution.** We implement two parametric representations of orientation distribution functions (ODFs) on $\mathbb{S}^2$ (Watson, 1965; Bingham, 1974). The Watson distribution (axially symmetric, antipodally symmetric) is

$$f_{\mathcal{W}}(\mathbf{n} \mid \boldsymbol{\mu}, \kappa) = C_{\mathcal{W}}(\kappa) \exp\left(\kappa(\boldsymbol{\mu}^\top \mathbf{n})^2\right), \qquad \mathbf{n} \in \mathbb{S}^2, \qquad C_{\mathcal{W}}(\kappa) = \frac{1}{4\pi \, _1F_1\left(\frac{1}{2}; \frac{3}{2}; \kappa\right)}. \tag{12}$$

It is parameterized by $\boldsymbol{\mu} \in \mathbb{S}^2$ and $\kappa \in \mathbb{R}$ (3 degrees of freedom: 2 for $\boldsymbol{\mu}$ and 1 for $\kappa$). The Bingham distribution (elliptical, antipodally symmetric), as implemented in Fick et al. (2019), is

$$f_{\mathcal{B}}(\mathbf{n} \mid \mathbf{A}) = C_{\mathcal{B}}(\mathbf{A}) \exp(\mathbf{n}^\top \mathbf{A} \mathbf{n}), \qquad \mathbf{n} \in \mathbb{S}^2, \qquad \mathbf{A} = \kappa \, \boldsymbol{\mu} \boldsymbol{\mu}^\top + \beta \, \boldsymbol{\mu}_\beta \boldsymbol{\mu}_\beta^\top, \tag{13}$$

with axes $\boldsymbol{\mu}, \boldsymbol{\mu}_\beta \in \mathbb{S}^2$ and concentrations $\kappa, \beta \in \mathbb{R}$. The normalizer $C_{\mathcal{B}}(\mathbf{A})$ has no closed form and is approximated via numerical integration.

To implement spherical convolutions with a fixed single-fiber response kernel—as required by dispersion/convolutional components (e.g. NODDI-style formulations)—we compute convolutions in the spherical-harmonics (SH) basis (Garyfallidis et al., 2014; Fick et al., 2019). We approximate an ODF by a truncated SH expansion $f(\mathbf{n}) \approx \sum_{\ell=0}^L \sum_{m=-\ell}^\ell c_{\ell m} Y_{\ell m}(\mathbf{n})$ and represent an axially symmetric response kernel $R(\mathbf{n})$ by its zonal SH coefficients $\{r_\ell\}_\ell$ (equivalently, a Legendre series). In this representation, spherical convolution reduces to per-band multiplication, $\tilde{c}_{\ell m} = r_\ell \, c_{\ell m}$, which is more efficient and numerically stable than direct quadrature over $\mathbb{S}^2$ (Garyfallidis et al., 2014; Fick et al., 2019). In practice, we (i) project the Watson/Bingham ODF onto SH coefficients (truncated to $L = 13$), (ii) apply the coefficient-wise multiplication by $\{r_\ell\}$, and (iii) reconstruct the convolved angular profile on the required gradient directions.

**Conversion to ODF.** To compare fiber-orientation uncertainty across heterogeneous model configurations, we map each fitted model to an orientation distribution function (ODF), denoted by $\mathrm{ODF}(\mathbf{n})$, whenever possible. For models already defined via an ODF (or an ODF convolved with a response kernel), this mapping is immediate. For tensor-based compartments, we use the analytical single-tensor ODF of Aganj et al. (2010). Given a diffusion tensor $\mathbf{D} = \mathbf{R} \, \mathrm{diag}(\boldsymbol{\lambda}) \, \mathbf{R}^\top$ with eigenvalues $\boldsymbol{\lambda} = (\lambda_1, \lambda_2, \lambda_3)$ and eigenvectors $\mathbf{R} \in \mathbb{R}^{3\times3}$ (columns), the induced ODF on directions $\mathbf{n} \in \mathbb{S}^2$ is

$$\mathrm{ODF}_T(\mathbf{n} \mid \mathbf{D}) = \frac{1}{4\pi \sqrt{\det(\mathbf{D})} \left(\mathbf{n}^\top \mathbf{D}^{-1} \mathbf{n}\right)^{3/2}} = \frac{1}{4\pi \sqrt{\lambda_1 \lambda_2 \lambda_3} \left(\mathbf{n}^\top \mathbf{D}^{-1} \mathbf{n}\right)^{3/2}}. \tag{14}$$

This includes the axially symmetric *Zeppelin* tensor by setting $\boldsymbol{\lambda} = (\lambda_\parallel, \lambda_\perp, \lambda_\perp)$ and choosing $\mathbf{R}$ such that its principal axis aligns with $\boldsymbol{\mu}$:

$$\mathrm{ODF}_Z(\mathbf{n} \mid \boldsymbol{\mu}, \lambda_\parallel, \lambda_\perp) = \frac{1}{4\pi \sqrt{\lambda_\parallel \lambda_\perp^2} \left(\mathbf{n}^\top \mathbf{D}_{\mathcal{Z}}^{-1} \mathbf{n}\right)^{3/2}}, \qquad \mathbf{D}_{\mathcal{Z}} = \mathbf{R}(\boldsymbol{\mu}) \, \mathrm{diag}(\lambda_\parallel, \lambda_\perp, \lambda_\perp) \, \mathbf{R}(\boldsymbol{\mu})^\top. \tag{15}$$

The *Stick* arises as the limiting case $\lambda_\perp \to 0$ (rank-1), yielding an ODF concentrated on a single axis (in the distributional sense),

$$\mathrm{ODF}_S(\mathbf{n} \mid \boldsymbol{\mu}) \propto \delta(\mathbf{n}; \boldsymbol{\mu}) + \delta(\mathbf{n}; -\boldsymbol{\mu}).$$

Any model configuration can thus be represented in ODF-space as a mixture, with mixture weights given by the fractions $\mathbf{f}$: $\mathrm{ODF}(\mathbf{n} \mid \mathcal{M}, \boldsymbol{\Theta}) = \sum_{k=0}^K f_k M_k \, \mathrm{ODF}_{\mathcal{M}_k}(\mathbf{n} \mid \boldsymbol{\theta}_k)$. Given a posterior over $(\mathcal{M}, \boldsymbol{\Theta})$, we define the posterior predictive ODF in local (model-conditional) and global (model-averaged) form as $\mathrm{ODF}_{\mathrm{local}}(\mathbf{n} \mid \mathcal{M}_k, \mathbf{x}_o) = \mathbb{E}_{p(\boldsymbol{\theta}_k \mid \mathcal{M}_k, \mathbf{x}_o)}[\mathrm{ODF}_{\mathcal{M}_k}(\mathbf{n} \mid \boldsymbol{\theta}_k)]$ and $\mathrm{ODF}_{\mathrm{global}}(\mathbf{n} \mid \mathbf{x}_o) = \mathbb{E}_{p(\mathcal{M}, \boldsymbol{\Theta} \mid \mathbf{x}_o)}[\mathrm{ODF}(\mathbf{n} \mid \mathcal{M}, \boldsymbol{\Theta})]$.

**Model prior.** Because candidate components differ substantially in parameter count, we quantify model complexity by the number of parameters rather than by the number of components. We use a dimension-penalized prior over model indicators

$$p(\mathcal{M} \mid \lambda) = \prod_k \mathrm{Ber}\big(M_k, p_k(\lambda, \dim(\boldsymbol{\theta}_{\mathcal{M}_k}))\big),$$

with $\eta_k = \mathrm{logit}(p_0) - (1 - \lambda) \cdot \lambda_{\max} \cdot \dim(\boldsymbol{\theta}_{\mathcal{M}_k})$ and $p_k = \sigma(\eta_k)$. Therefore $p_k$ decreases with increasing parameter dimension of $\boldsymbol{\theta}_{\mathcal{M}_k}$. We draw $\lambda \sim \mathrm{Unif}([0, 1])$ and set $\lambda_{\max} = 4$ and $p_0 = 0.5$.

### B.2.3. RANDOM ACQUISITION PROTOCOL GENERATION

We generated random `bvals` and `bvecs` as follows:

(i) **UKB-like**: We fixed $n = 105$ acquisitions. For each acquisition, we sampled the $b$-value from a mixture of (a) a fixed "typical" UKB-like set $\{b_i^{\mathrm{typ}}\}_{i=1}^n$ and (b) a uniform draw $b_i^{\mathrm{rand}} \sim \mathrm{Unif}(0, 6000)$, with mixture weights $(0.5, 0.5)$. We

then applied a small dropout mask $m_i \sim \text{Bernoulli}(0.99)$ (i.e., $1\%$ zeros) and set $b_i \leftarrow m_i \, b_i$. For directions, we drew random unit vectors $\mathbf{g}_i^{\text{rand}} \sim \text{Unif}(\mathbb{S}^2)$ by sampling $\tilde{\mathbf{g}}_i \sim \mathcal{N}(\mathbf{0}, \mathbf{I}_3)$ and normalizing, and then sampled `bvecs` from a mixture of the typical directions $\{\mathbf{g}_i^{\text{typ}}\}_{i=1}^n$ (i.e., an even spherical grid) and the random directions, again with weights $(0.5, 0.5)$.

(ii) **HCP-like**: We used the same procedure with $n = 295$ acquisitions.

### B.2.4. B3S TASK CONFIGURATION

The B3S task includes one Ball, up to three Sticks, and a Gaussian noise model (Tab. 11). To remain comparable to Manzano-Patron et al. (2025), we impose additional structure on this family. In particular, we tie the diffusivity across Ball and Stick compartments via a shared parameter $d$, i.e., $d_{\mathcal{B}} = d_{\mathcal{S}_1} = d_{\mathcal{S}_2} = d_{\mathcal{S}_3} = d$. Moreover, only *multi-shell* models were considered, and we therefore adapted to this case by replacing the mono-exponential nonlinearity with the appropriate multi-shell extension (Jbabdi et al., 2012). Moreover, to mimic the strong sparsity bias used in FSL BedpostX, which employs an improper prior that penalizes non-zero fractions and thereby encourages automatic model selection (Behrens et al., 2007), we use an informative prior over fractions, $p(\mathbf{f}) = \text{Dirichlet}(\mathbf{f} \mid \boldsymbol{\alpha})$ with $\boldsymbol{\alpha} = [3.5, 1.0, 0.3, 0.1]$.

Unlike Manzano-Patron et al. (2025), we do not enforce hard angular constraints between Stick orientations. Such constraints are acquisition-specific and can become misaligned when amortizing across heterogeneous protocols; our training targets acquisition-agnostic inference (e.g., improved angular resolution with increased numbers of acquisitions), and we therefore avoid constraint sets that implicitly encode a fixed sampling scheme.

### B.2.5. BSZT AND BSZT + CONV TASK CONFIGURATION

For the BSZT space, we include one Ball, three Sticks, three Zeppelins, and three Tensors, together with Gaussian and Rician noise models (Tab. 11). We do not impose parameter sharing across compartments and use a uniform Dirichlet prior over fractions, $p(\mathbf{f}) = \text{Dirichlet}(\mathbf{f} \mid \boldsymbol{\alpha})$ with $\boldsymbol{\alpha} = \mathbf{1}$.

For the BSZT+conv space, we further extend BSZT by adding all remaining dispersed and convolutional compartments from Table 11, while keeping the same uniform fraction prior $\boldsymbol{\alpha} = \mathbf{1}$.

*Table 10.* Extended component and noise catalogue (as implemented in code).

| ID | Component | Token | Expression (implementation string) | Parameter prior |
|----|-----------|-------|-----------------------------------|-----------------|
| 0 | Linear | lin | `c_1*x` | $c_1 \sim \mathcal{U}(-2, 2)$ |
| 1 | Quadratic | quad | `c_1*x*x` | $c_1 \sim \mathcal{U}(-0.5, 0.5)$ |
| 2 | ShiftedSquare | shift2 | `(c_1+x)*(c_1+x)` | $c_1 \sim \mathcal{U}(-5, 0)$ |
| 3 | Cubic | cub | `c_1*x*x*x` | $c_1 \sim \mathcal{U}(-0.1, 0.1)$ |
| 4 | Sinusoidal | sin | `c_1*sin(c_2*x)` | $c_1 \sim \mathcal{U}(0, 5)$ 
 $c_2 \sim \mathcal{U}(0.5, 5)$ |
| 5 | Cosinusoidal | cos | `c_1*cos(c_2*x)` | $c_1 \sim \mathcal{U}(0, 5)$ 
 $c_2 \sim \mathcal{U}(0.5, 5)$ |
| 6 | ConstantWide | const_w | `c_1` | $c_1 \sim \mathcal{U}(-5, 5)$ |
| 7 | ConstantPositive | const_p | `c_1` | $c_1 \sim \mathcal{U}(0, 10)$ |
| 8 | TanhRight | tanh_r | `c_1*tanh(x-c_2)` | $c_1 \sim \mathcal{U}(1, 10)$ 
 $c_2 \sim \mathcal{U}(2, 8)$ |
| 9 | TanhLeft | tanh_l | `c_1*tanh(-x+c_2)` | $c_1 \sim \mathcal{U}(1, 10)$ 
 $c_2 \sim \mathcal{U}(2, 8)$ |
| 10 | GaussianBump | gauss_b | `c_1*exp(-(x-c_2)*(x-c_2))` | $c_1 \sim \mathcal{U}(1, 10)$ 
 $c_2 \sim \mathcal{U}(2, 8)$ |
| 11 | GaussianWide | gauss_w | `c_1*exp(-(x-c_2)*(x-c_2)/8)` | $c_1 \sim \mathcal{U}(1, 10)$ 
 $c_2 \sim \mathcal{U}(2, 8)$ |
| 12 | RampUp | ramp_u | `c_1*Piecewise((0.0,x<c_2),(x,` 
 `x>=c_2))` | $c_1 \sim \mathcal{U}(1, 5)$ 
 $c_2 \sim \mathcal{U}(2, 8)$ |
| 13 | RampDown | ramp_d | `c_1*Piecewise((0.0,x>c_2),(-x+c_2,` 
 `x<=c_2))` | $c_1 \sim \mathcal{U}(1, 5)$ 
 $c_2 \sim \mathcal{U}(2, 8)$ |
| 14 | QuarticScaled | quart4 | `c_1*(x/10)**4` | $c_1 \sim \mathcal{U}(-5, 5)$ |
| 15 | QuinticScaled | quint5 | `c_1*(x/10)**5` | $c_1 \sim \mathcal{U}(-5, 5)$ |
| 16 | SinusoidalPhase | sin_ph | `c_1*sin(c_2*x + c_3)` | $c_1 \sim \mathcal{U}(0, 5)$ 
 $c_2 \sim \mathcal{U}(0.5, 5)$ 
 $c_3 \sim \mathcal{U}(-\pi, \pi)$ |
| 17 | CosinusoidalPhase | cos_ph | `c_1*cos(c_2*x + c_3)` | $c_1 \sim \mathcal{U}(0, 5)$ 
 $c_2 \sim \mathcal{U}(0.5, 5)$ 
 $c_3 \sim \mathcal{U}(-\pi, \pi)$ |
| 18 | ExponentialDecay | exp_dec | `c_1*exp(-c_2*x)` | $c_1 \sim \mathcal{U}(0, 10)$ 
 $c_2 \sim \mathcal{U}(0.1, 2)$ |
| 19 | SaturatingExponential | exp_sat | `c_1*(1-exp(-c_2*x))` | $c_1 \sim \mathcal{U}(0, 10)$ 
 $c_2 \sim \mathcal{U}(0.1, 2)$ |
| 20 | Logarithmic | log | `c_1*log(x+c_2)` | $c_1 \sim \mathcal{U}(-5, 5)$ 
 $c_2 \sim \mathcal{U}(0.1, 2)$ |
| 21 | SquareRoot | sqrt | `c_1*sqrt(x+c_2)` | $c_1 \sim \mathcal{U}(-5, 5)$ 
 $c_2 \sim \mathcal{U}(0, 2)$ |
| 22 | Reciprocal | recip | `c_1/(x+c_2)` | $c_1 \sim \mathcal{U}(-10, 10)$ 
 $c_2 \sim \mathcal{U}(0.5, 3)$ |
| 23 | AbsoluteValue | abs | `c_1*Abs(x-c_2)` | $c_1 \sim \mathcal{U}(-5, 5)$ 
 $c_2 \sim \mathcal{U}(0, 10)$ |
| 24 | InverseQuadratic | invquad | `c_1/(1 + c_2*(x-c_3)*(x-c_3))` | $c_1 \sim \mathcal{U}(0, 10)$ 
 $c_2 \sim \mathcal{U}(0.1, 2)$ 
 $c_3 \sim \mathcal{U}(0, 10)$ |
| 25 | Lorentzian | lorentz | `c_1/(1 + ((x-c_2)/c_3)**2)` | $c_1 \sim \mathcal{U}(0, 10)$ 
 $c_2 \sim \mathcal{U}(0, 10)$ 
 $c_3 \sim \mathcal{U}(0.5, 5)$ |

| ID | Component | Token | Expression (implementation string) | Parameter prior |
|---|---|---|---|---|
| 26 | Sigmoid | sig | `c_1/(1+exp(-c_2*(x-c_3)))` | $c_1 \sim \mathcal{U}(0, 10)$
$c_2 \sim \mathcal{U}(0.1, 5)$
$c_3 \sim \mathcal{U}(0, 10)$ |
| 27 | DampedSinusoidal | d_sin | `c_1*exp(-c_2*x)*sin(c_3*x)` | $c_1 \sim \mathcal{U}(0, 5)$
$c_2 \sim \mathcal{U}(0.05, 1)$
$c_3 \sim \mathcal{U}(0.5, 8)$ |
| 28 | DampedCosinusoidal | d_cos | `c_1*exp(-c_2*x)*cos(c_3*x)` | $c_1 \sim \mathcal{U}(0, 5)$
$c_2 \sim \mathcal{U}(0.05, 1)$
$c_3 \sim \mathcal{U}(0.5, 8)$ |
| 29 | TanhCentered | tanh_c | `c_1*tanh(c_2*(x-c_3))` | $c_1 \sim \mathcal{U}(-10, 10)$
$c_2 \sim \mathcal{U}(0.1, 2)$
$c_3 \sim \mathcal{U}(2, 8)$ |
| 30 | ExponentialGrowth | exp_grow | `c_1*exp(c_2*x)` | $c_1 \sim \mathcal{U}(0, 10)$
$c_2 \sim \mathcal{U}(0.05, 0.8)$ |
| 31 | PowerLawDecay | pow_dec | `c_1/(x + c_2)**c_3` | $c_1 \sim \mathcal{U}(0, 10)$
$c_2 \sim \mathcal{U}(0.5, 5)$
$c_3 \sim \mathcal{U}(0.5, 3)$ |
| 32 | ArctangentStep | atan | `c_1*atan(c_2*(x-c_3))` | $c_1 \sim \mathcal{U}(0, 10)$
$c_2 \sim \mathcal{U}(0.1, 2)$
$c_3 \sim \mathcal{U}(0, 10)$ |
| 33 | HyperbolicSecant | sech | `c_1*sech(c_2*(x-c_3))` | $c_1 \sim \mathcal{U}(0, 8)$
$c_2 \sim \mathcal{U}(0.1, 2)$
$c_3 \sim \mathcal{U}(0, 10)$ |
| 34 | SincDecay | sinc | `c_1*sin(c_2*x)/(x+c_3)` | $c_1 \sim \mathcal{U}(0, 5)$
$c_2 \sim \mathcal{U}(0.5, 5)$
$c_3 \sim \mathcal{U}(0.5, 5)$ |
| 35 | AbsoluteSinusoidal | abs_sin | `c_1*Abs(sin(c_2*x + c_3))` | $c_1 \sim \mathcal{U}(0, 5)$
$c_2 \sim \mathcal{U}(0.5, 5)$
$c_3 \sim \mathcal{U}(-\pi, \pi)$ |
| 36 | RectifiedLinear | relu | `c_1*Piecewise((0, x < c_2), (x - c_2, True))` | $c_1 \sim \mathcal{U}(0, 5)$
$c_2 \sim \mathcal{U}(0, 8)$ |
| 37 | Softplus | softplus | `c_1*log(1+exp(c_2*(x-c_3)))/c_2` | $c_1 \sim \mathcal{U}(0, 5)$
$c_2 \sim \mathcal{U}(0.1, 5)$
$c_3 \sim \mathcal{U}(0, 10)$ |
| 38 | Gompertz | gomp | `c_1*exp(-c_2*exp(-c_3*x))` | $c_1 \sim \mathcal{U}(0, 10)$
$c_2 \sim \mathcal{U}(0.1, 3)$
$c_3 \sim \mathcal{U}(0.05, 1.5)$ |
| 39 | LinearFractional | linfrac | `c_1*x/(1 + c_2*x)` | $c_1 \sim \mathcal{U}(-5, 5)$
$c_2 \sim \mathcal{U}(0.1, 2)$ |
| 40 | SineSquared | sin2 | `c_1*sin(c_2*x)**2` | $c_1 \sim \mathcal{U}(0, 5)$
$c_2 \sim \mathcal{U}(0.5, 5)$ |
| 41 | TriangularBump | tri | `c_1*Piecewise((0, Abs((x-c_2)/c_3) >= 1), (1 - Abs((x-c_2)/c_3), True))` | $c_1 \sim \mathcal{U}(0, 5)$
$c_2 \sim \mathcal{U}(0.5, 5)$
$c_3 \sim \mathcal{U}(0.5, 5)$ |
| 42 | NoiseObserver | n_obs | `normal(c_1)` | $c_1 \sim \mathcal{U}(0.1, 2)$ |
| 43 | NoiseIncreasing | n_inc | `normal(c_1) * (x+1)` | $c_1 \sim \mathcal{U}(0.5, 2)$ |
| 44 | NoiseDecreasing | n_dec | `normal(c_1) * (11 - x)` | $c_1 \sim \mathcal{U}(0.5, 2)$ |
| 45 | NoiseQuadratic | n_quad | `normal(c_1) *(x**2 + 1)` | $c_1 \sim \mathcal{U}(0.2, 1)$ |
| 46 | NoiseQuadraticDecreasing | n_qdec | `normal(c_1) * (11 - x**2)` | $c_1 \sim \mathcal{U}(0.2, 1)$ |

*Continued on next page*

| ID | Component | Token | Expression (implementation string) | Parameter prior |
|----|-----------|-------|-----------------------------------|-----------------|
| 47 | NoiseExponential | n_exp | `normal(c_1) * exp(c_2 * x)` | $c_1 \sim \mathcal{U}(0, 5)$ 
 $c_2 \sim \mathcal{U}(0.05, 0.5)$ |
| 48 | NoiseSigmoid | n_sig | `normal(c_1)/(1+exp(-c_2*(x-c_3)))` | $c_1 \sim \mathcal{U}(0, 5)$ 
 $c_2 \sim \mathcal{U}(0.1, 1)$ 
 $c_3 \sim \mathcal{U}(0, 10)$ |
| 49 | NoisePeaked | n_peak | `normal(c_1) * exp(-((x-c_2)**2)/(c_3 + 1e-3))` | $c_1 \sim \mathcal{U}(0, 5)$ 
 $c_2 \sim \mathcal{U}(0, 10)$ 
 $c_3 \sim \mathcal{U}(0.5, 5)$ |

*Table 11.* Diffusion MRI signal models (`DMIPY`-style) with per-parameter priors.

| ID | Name | Symbol | Signal model | $\theta$ | Prior |
|---|---|---|---|---|---|
| 0 | Ball | B | $S_B(b) = \exp\big(-b\,d_{\mathrm{iso}}\big)$ | $d_{\mathrm{iso}}$ | $d_{\mathrm{iso}} \sim \mathcal{U}(0, 0.01)$ |
| 1 | Stick | S | $S_S(b, \vec{\mathbf{b}}) = \exp\Big(-b\,d_\parallel(\vec{\mathbf{b}}\cdot\mathbf{n})^2\Big)$ | $d_\parallel$ $\mathbf{n}$ | $d_\parallel \sim \mathcal{U}(0,\ 0.01)$ $\mathbf{n} \sim \mathcal{U}(\mathbb{S}^2_+)$ |
| 2 | Zeppelin | Z | $S_Z(b, \vec{\mathbf{b}}) =$ $\exp\Big(-b\big[d_\parallel(\vec{\mathbf{b}}\cdot\mathbf{n})^2 + d_\perp\big(1-(\vec{\mathbf{b}}\cdot\mathbf{n})^2\big)\big]\Big)$ | $d_\parallel$ $d_\perp$ $\mathbf{n}$ | $d_\parallel \sim \mathcal{U}(0,\ 0.01)$ $d_\perp \sim \mathcal{U}(0, 0.01)$ $\mathbf{n} \sim \mathcal{U}(\mathbb{S}^2_+)$ |
| 3 | Tensor (DTI) | T | $S_T(b, \vec{\mathbf{b}}) = \exp\Big(-b\,\vec{\mathbf{b}}^\top \mathbf{D}\,\vec{\mathbf{b}}\Big), \quad \mathbf{D} \succ 0$ | $\mathbf{D}$ | $\theta \sim \mathcal{N}(\mathbf{0}, \mathbf{I}_6)$ $\mathbf{L} = \mathrm{tril}(\theta) \in \mathbb{R}^{3\times3}$ $\tilde{\mathbf{D}} = \mathbf{L}\mathbf{L}^\top$ $\mathbf{D} = D_{\mathrm{scale}}\tilde{\mathbf{D}} + \lambda_{\min}\mathbf{I}_3$ $D_{\mathrm{scale}} = 10^{-3}, \lambda_{\min} = 10^{-4}$ |
| 4 | Watson Stick | $\mathrm{S}_\mathcal{W}$ | $S_{WS}(b, \vec{\mathbf{b}}) = \int_{\mathbb{S}^2} S_S(b, \vec{\mathbf{b}} \mid \mathbf{n}, d_\parallel)\, f_\mathcal{W}(\mathbf{n} \mid \boldsymbol{\mu}, \kappa)\, d\mathbf{n}$ | $d_\parallel$ $\boldsymbol{\mu}$ ODI | $d_\parallel \sim \mathcal{U}(0,\ 0.01)$ $\boldsymbol{\mu} \sim \mathrm{Unif}(\mathbb{S}^2_+)$ $\mathrm{ODI} \sim \mathrm{Unif}(0,1)$ $\kappa = \dfrac{1}{\tan\big(\mathrm{ODI}\,\frac{\pi}{2}\big)}$ |
| 5 | Bingham Stick | $\mathrm{S}_\mathcal{B}$ | $S_{BS}(b, \vec{\mathbf{b}}) = \int_{\mathbb{S}^2} S_S(b, \vec{\mathbf{b}} \mid \mathbf{n})\, f_B(\mathbf{n} \mid \boldsymbol{\mu}, \kappa, \beta, \psi)\, d\mathbf{n}$ | $d_\parallel$ $\boldsymbol{\mu}$ ODI $\beta_{\mathrm{frac}}$ $\psi$ | $d_\parallel \sim \mathcal{U}(0,\ 0.01)$ $\boldsymbol{\mu} \sim \mathcal{U}(\mathbb{S}^2_+)$ $\mathrm{ODI} \sim \mathcal{U}(0,1)$ $\beta_{\mathrm{frac}} \sim \mathcal{U}(0,1)$ $\psi \sim \mathcal{U}(0,\pi)$ $\kappa = \dfrac{1}{\tan\big(\mathrm{ODI}\,\frac{\pi}{2}\big)}$ $\beta = \beta_{\mathrm{frac}}\kappa$ |
| 6 | Watson Zeppelin | $\mathrm{Z}_\mathcal{W}$ | $S_{WZ}(b, \vec{\mathbf{b}}) = \int_{\mathbb{S}^2} S_Z(b, \vec{\mathbf{b}} \mid \mathbf{n})\, f_\mathcal{W}(\mathbf{n} \mid \boldsymbol{\mu}, \kappa)\, d\mathbf{n}$ | $d_\parallel$ $d_\perp$ $\boldsymbol{\mu}$ ODI | $d_\parallel \sim \mathcal{U}(0,\ 3 \times 10^{-9})$ $d_\perp \sim \mathcal{U}(0,\ d_\parallel)$ $\boldsymbol{\mu} \sim \mathcal{U}(\mathbb{S}^2_+)$ $\mathrm{ODI} \sim \mathcal{U}(0., 1.)$ $\kappa = \dfrac{1}{\tan\big(\mathrm{ODI}\,\frac{\pi}{2}\big)}$ |
| 7 | Bingham Zeppelin | $\mathrm{Z}_\mathcal{B}$ | $S_{BZ}(b, \vec{\mathbf{b}}) = \int_{\mathbb{S}^2} S_Z(b, \vec{\mathbf{b}} \mid \mathbf{n})\, f_\mathcal{B}(\mathbf{n} \mid \boldsymbol{\mu}, \kappa, \beta, \psi)\, d\mathbf{n}$ | $d_\parallel$ $d_\perp$ $\boldsymbol{\mu}$ ODI $\beta_{\mathrm{frac}}$ $\psi$ | $d_\parallel \sim \mathcal{U}(0,\ 0.01)$ $d_\perp \sim \mathcal{U}(0,\ 0.01)$ $\boldsymbol{\mu} \sim \mathcal{U}(\mathbb{S}^2_+)$ $\mathrm{ODI} \sim \mathcal{U}(0., 1.)$ $\beta_{\mathrm{frac}} \sim \mathcal{U}(0,\ 1)$ $\psi \sim \mathcal{U}(0., \pi)$ $\kappa = \dfrac{1}{\tan\big(\mathrm{ODI}\,\frac{\pi}{2}\big)}$ $\beta = \beta_{\mathrm{frac}}\,\kappa$ |
| 9 | NODDI (Watson) | $\mathrm{No}_\mathcal{W}$ | $S_{\mathrm{NODDI\text{-}}\mathcal{W}}(b, \vec{\mathbf{b}}) = f\, S_{WS}(b, \vec{\mathbf{b}}; \boldsymbol{\mu}, \mathrm{ODI}, \lambda_\parallel) + (1-f)\, S_{WZ}(b, \vec{\mathbf{b}}; \boldsymbol{\mu}, \mathrm{ODI}, \lambda_\perp, \lambda_\parallel)$ | $f$ $\lambda_\perp$ $\lambda_\parallel$ $\boldsymbol{\mu}$ ODI | $f \sim \mathcal{U}(0., 1.)$ $\lambda_\perp \sim \mathcal{U}(0., 0.01)$ $\lambda_\parallel \sim \mathcal{U}(0., 0.01)$ $\boldsymbol{\mu} \sim \mathcal{U}(\mathbb{S}^2_+)$ $\mathrm{ODI} \sim \mathcal{U}(0., 1.)$ |

| ID | Name | Symbol | Signal model | $\theta$ | Prior |
|---|---|---|---|---|---|
| | | | | | $\kappa = \dfrac{1}{\tan\left(\text{ODI}\frac{\pi}{2}\right)}$ |
| 10 | NODDI (Bingham) | $\text{No}_{\mathcal{B}}$ | $S_{\text{NODDI}\_\mathcal{B}}(b,\vec{\mathbf{b}}) =$ $f\,S_{BS}(b,\vec{\mathbf{b}};\boldsymbol{\mu},\text{ODI},\beta_{\text{frac}},\psi,\lambda_{\parallel}) + (1- f)\,S_{BZ}(b,\vec{\mathbf{b}};\boldsymbol{\mu},\text{ODI},\beta_{\text{frac}},\psi,\lambda_{\perp},\lambda_{\parallel})$ | $f$ $\lambda_{\perp}$ $\lambda_{\parallel}$ $\boldsymbol{\mu}$ ODI $\beta_{\text{frac}}$ $\psi$ | $f \sim \mathcal{U}(0.,\ 1.)$ $\lambda_{\perp} \sim \mathcal{U}(0.,\ 0.01)$ $\lambda_{\parallel} \sim \mathcal{U}(0.,\ 0.01)$ $\boldsymbol{\mu} \sim \mathcal{U}(\mathbb{S}^2_+)$ $\text{ODI} \sim \mathcal{U}(0.,\ 1.)$ $\beta_{\text{frac}} \sim \mathcal{U}(0.,\ 1.)$ $\psi \sim \mathcal{U}(0.,\ \pi)$ $\kappa = \dfrac{1}{\tan\left(\text{ODI}\frac{\pi}{2}\right)},\quad \beta = \beta_{\text{frac}}\kappa$ |
| 9 | SANDI (Watson) | $\text{Sa}_{\mathcal{W}}$ | $S_{\text{SANDI}\_\mathcal{W}}(b,\vec{\mathbf{b}}) =$ $(1-f_{\text{ec}})\Big[f_{\text{in}}\,S_{WS}(b,\vec{\mathbf{b}};\boldsymbol{\mu},\text{ODI},\lambda_{\parallel}^{\text{in}}) + (1- f_{\text{in}})\cdot 1\Big] + f_{\text{ec}}\,S_{WZ}(b,\vec{\mathbf{b}};\boldsymbol{\mu},\text{ODI},\lambda_{\perp}^{\text{ex}},\lambda_{\parallel}^{\text{ex}})$ | $f_{\text{in}}$ $f_{\text{ec}}$ $\lambda_{\parallel}^{\text{in}}$ $\lambda_{\perp}^{\text{ex}}$ $\lambda_{\parallel}^{\text{ex}}$ $\boldsymbol{\mu}$ ODI | $f_{\text{in}} \sim \mathcal{U}(0.,\ 1.)$ $f_{\text{ec}} \sim \mathcal{U}(0.,\ 1.)$ $\lambda_{\parallel}^{\text{in}} \sim \mathcal{U}(0.,\ 0.01)$ $\lambda_{\perp}^{\text{ex}} \sim \mathcal{U}(0.,\ 0.01)$ $\lambda_{\parallel}^{\text{ex}} \sim \mathcal{U}(0.,\ 0.01)$ $\boldsymbol{\mu} \sim \mathcal{U}(\mathbb{S}^2_+)$ $\text{ODI} \sim \mathcal{U}(0.,\ 1.)$ $\kappa = \dfrac{1}{\tan\left(\text{ODI}\frac{\pi}{2}\right)}$ |
| 10 | SANDI (Bingham) | $\text{Sa}_{\mathcal{B}}$ | $S_{\text{SANDI}\_\mathcal{B}}(b,\vec{\mathbf{b}}) =$ $(1-f_{\text{ec}})\Big[f_{\text{in}}\,S_{BS}(b,\vec{\mathbf{b}};\boldsymbol{\mu},\text{ODI},\beta_{\text{frac}},\psi,\lambda_{\parallel}^{\text{in}}) +$ $(1-f_{\text{in}})\cdot 1\Big] +$ $f_{\text{ec}}\,S_{BZ}(b,\vec{\mathbf{b}};\boldsymbol{\mu},\text{ODI},\beta_{\text{frac}},\psi,\lambda_{\perp}^{\text{ex}},\lambda_{\parallel}^{\text{ex}})$ | $f_{\text{in}}$ $f_{\text{ec}}$ $\lambda_{\parallel}^{\text{in}}$ $\lambda_{\perp}^{\text{ex}}$ $\lambda_{\parallel}^{\text{ex}}$ $\boldsymbol{\mu}$ ODI $\beta_{\text{frac}}$ $\psi$ | $f_{\text{in}} \sim \mathcal{U}(0.,\ 1.)$ $f_{\text{ec}} \sim \mathcal{U}(0.,\ 1.)$ $\lambda_{\parallel}^{\text{in}} \sim \mathcal{U}(0.,\ 0.01)$ $\lambda_{\perp}^{\text{ex}} \sim \mathcal{U}(0.,\ 0.01)$ $\lambda_{\parallel}^{\text{ex}} \sim \mathcal{U}(0.,\ 0.01)$ $\boldsymbol{\mu} \sim \mathcal{U}(\mathbb{S}^2_+)$ $\text{ODI} \sim \mathcal{U}(0.,\ 1.)$ $\beta_{\text{frac}} \sim \mathcal{U}(0.,\ 1.)$ $\psi \sim \mathcal{U}(0.,\ \pi)$ $\kappa = \dfrac{1}{\tan\left(\text{ODI}\frac{\pi}{2}\right)}$ $\beta = \beta_{\text{frac}}\kappa$ |
| 10 | Gaussian noise | G | $y = S_0\,S(\cdot) + \epsilon,\quad \epsilon \sim \mathcal{N}(0,\sigma^2)$ | $\sigma$ | $\text{SNR} \sim \mathcal{U}(3,\ 80)$ $\sigma = \dfrac{1}{\text{SNR}}$ |
| 11 | Rician noise | R | $y \sim \text{Rice}(\nu = S_0 S(\cdot),\sigma),\ p(y\mid\nu,\sigma) =$ $\dfrac{y}{\sigma^2}\exp\left(-\dfrac{y^2+\nu^2}{2\sigma^2}\right)I_0\left(\dfrac{y\nu}{\sigma^2}\right)$ | $\sigma$ | $\text{SNR} \sim \mathcal{U}(3,\ 80)$ $\sigma = \dfrac{1}{\text{SNR}}$ |

