# OpenReview forum: "Scalable Simulation-Based Model Inference with Test-Time Complexity Control"
_ICML.cc/2026/Conference — ICML 2026 regular_

### Official Review · Reviewer_SKBZ · 2026-03-11

**Soundness:** 3
**Presentation:** 4
**Significance:** 3
**Originality:** 3
**Overall Recommendation:** 5
**Confidence:** 3

**Summary:**

The paper introduces PRISM, an amortized simulation-based inference framework for joint inference over discrete model structure and continuous parameters, while allowing test-time control of model complexity through a tunable prior parameter lambda. The method is evaluated on a large symbolic regression benchmark and on diffusion MRI biophysical model selection, where it aims to scale Bayesian model comparison to large combinatorial model families without retraining for different prior preferences.

**Compliance With Llm Reviewing Policy:**

Affirmed.

**Final Justification:**

Recommend for accept.
Well-structured paper with clear presentation. The core idea about Bayesian model selection over large families of candidate simulators, enabling a test-time complexity control is quite interesting and practical. The evaluations across symbolic regression and dMRI demonstrate scaling and utility. The rebuttal addressed my initial concerns about scaling, parameter settings and computational cost.

**Key Questions For Authors:**

1.How does the independence assumption in p(M|λ) = ∏ Ber(Mk, λ) affect inference when model components have structural dependencies, such as mutually exclusive or hierarchical relationships?
2. How does the joint training objective balance model posterior inference and parameter posterior inference in practice? Are there diagnostics showing that the model decoder does not collapse toward overconfident model choices or ignore model uncertainty?
3. What guidance do the authors have for choosing lambda in practice?
4. Are there ablations on key design choices, such as autoregressive versus independent model decoding, or alternative tokenization strategies?
5. For both symbolic regression and dMRI, could the authors provide more details on training data requirements and computational cost? This would help readers assess the practical training burden of the method.

**Limitations:**

yes

**Strengths And Weaknesses:**

Strength
1. Well-structured paper with clear presentation. The related work is quite thorough.
2. The core idea about Bayesian model selection over large families of candidate simulators, enabling a test-time complexity control is quite interesting and practical.
3. The combination of (i) autoregressive Bernoulli decoder for model posterior and (ii) diffusion v-prediction decoder for parameter posterior, both conditioned on lambda via adaptive layer normalization, is technically sound and well-motivated.
4. The evalution across symbolic regression, and scale to computationally cost diffusion MRI condition in real scientific application, allowing for prior-flexible setting.

Weakness
1. The paper shows promising scaling behavior, but also explicitly notes that performance degrades when large parts of model space remain insufficiently covered during training. This makes the practical scaling limit somewhat unclear.
2. It is still somewhat unclear how reliably the method captures the joint interaction between model posterior and parameter posterior across different lambda settings, especially in examples such as Fig. 4a where the true posterior remains challenging.
3. The HCP-like and UKB-like results are not consistently close. This weakens the claim of robust cross-protocol amortization, and suggests that performance may depend substantially on acquisition protocol and training coverage of model space.
4. The dMRI results are promising, but practical readiness is still limited.

---

> ### Author Rebuttal · Authors · 2026-03-31
>
> We thank the reviewer for the thoughtful review, and for highlighting the paper’s “clear presentation,” the “interesting and practical” core idea of Bayesian model selection with test-time complexity control, and that the symbolic regression benchmark and diffusion MRI evaluation address meaningful settings.
>
> We address the main concerns below:
> - **Practical scaling limit**: We thank the reviewer for noting that “the paper shows promising scaling behavior,” as this is what we aimed to demonstrate (Fig. 4c and Fig. A.2). Even in this regime, PRISM shows encouraging extrapolation to larger model spaces. That said, this is not a full scaling-law analysis; such a study would be valuable but is beyond the scope of this manuscript.
> - **Interaction of model and parameter posterior across λ**: For the dMRI tasks we already provide a detailed analysis across λ in Appendix A.4.1 (Fig. A5, A7, A8, A12), including calibration, comparison to MC, and confusion matrices. We now extend this analysis to symbolic regression with a similar addition in A.3 (see [1], Fig. 2a). These results show consistent performance across this axis.
> - **UKB/HCP results**: We do not agree with the reviewer on this point. There are differences, but they are small. While in Fig. 6b right (and Fig. 8a right) calibration may appear slightly different for the two acquisition schemes, a calibration error around 0.01 still indicates very good calibration. A full comparison across acquisition schemes on synthetic data is shown in Tab. A2 (B3S), Tab. A3 (BSZT), and Tab. A4 (BSZT + conv.), with similar performance across metrics and data schemes for all model classes. Therefore, the claim of robust cross-protocol amortization holds.
> - **Practical readiness**: We fully acknowledge that further work is needed for practical readiness in dMRI. However, we neither claim nor aim in this work to provide a “production-ready” tool, but rather a method that can tackle dMRI-relevant problems on both small and larger scales.
>
> >Q1. Independence assumption of the prior.
>
> A1. We want to clarify that the independence is within the *user-chosen* model prior, not within the learned model posterior, which will contain structural dependencies/correlations. We chose a simple, interpretable prior for simplicity, but more complex model priors, e.g. graph-based ones, can be used without difficulty. The only requirement is a sampling procedure. See also A2 to R. YsWR.
>
> >Q2. Joint training objective.
>
> A2. We focus on an “online” training setting, i.e. streaming data from the simulator, resulting in effectively “unlimited” training data. As both objectives—learning the model and parameter posterior—are non-conflicting and mass-covering (Wildberger et al., 2023), the risk of posterior overfitting is low. We provide exemplary training loss curves in A4 [1], Fig. 3. On small datasets, overfitting can still be a risk, and we observe that convergence of the parameter posterior is often slower than that of the model posterior. More generally, SBI is in principle an infinite-data regime, though simulation cost can limit the number of training examples in practice. PRISM is not specifically designed for low-data settings, but standard strategies such as early stopping or freezing parts of the model, e.g. the model posterior, can mitigate overfitting.
>
> The presented diagnostics, specifically SBC calibration errors, also check for overconfidence in model predictions (see Reviewer GcKH A3 for an intuitive description; also Fig. 6b, 8a, and Fig. A2c, A5, A8).
>
> >Q3. Choice of lambda
>
> A3. Choosing lambda is somewhat analogous to choosing the “right” regularization strength in standard optimization. This is often difficult to do a priori (and is frequently done via cross-validation), since it can strongly affect generalization (e.g. Fig. 2/A1). PRISM amortizes over different lambda values to allow trying and selecting the right trade-off.
>
> >Q4. Key design choices
>
> A4. We want to highlight that there are many tokenization strategies, and they are task dependent. While we did not investigate them systematically, we built on established encoding schemes. We also refer to R. GckH, A2, for a discussion of the architectural choices. For the model posterior, there are clear correlation structures, partly even induced by the prior ([1] Fig. 2b left; e.g. small lambda encourages posterior correlations). We also conducted an ablation of independent vs. autoregressive model posterior as suggested by the reviewer ([1] Fig. 2b right), which clearly shows that the independent posterior fails SBC calibration.
>
> >Q5. Computational costs
>
> A5. We admit that this information is only partially available in the Appendix. We now summarize all relevant information in a dedicated Appendix section; in particular, we compiled a comprehensive table with computational costs across tasks (see hMMd response and [1], Table 2/3).
>
> [1] Supplementary PDF: https://www.swisstransfer.com/d/ead46821-0d93-4190-8535-d281bad7c606

---

> > ### Author Rebuttal · Reviewer_SKBZ · 2026-04-02
> >
> > I appreciate the authors' responses. I will keep my score.

---

### Official Review · Reviewer_hMMd · 2026-03-12

**Soundness:** 3
**Presentation:** 3
**Significance:** 3
**Originality:** 2
**Overall Recommendation:** 4
**Confidence:** 4

**Summary:**

This paper introduces PRISM, a simulation-based model inference framework that uses a transformer encoder-decoder architecture to jointly infer discrete model structures and continuous parameters. The model conditions amortized inference on a test-time complexity control variable, allowing users to navigate the trade-off between parsimony and goodness-of-fit without retraining. The method is evaluated on both a symbolic regression benchmark with combinatorially large model spaces and a biophysical modeling task, and the reported results inform a possible scalability and competitive inference quality.

**Compliance With Llm Reviewing Policy:**

Affirmed.

**Key Questions For Authors:**

1. The architecture appears to omit standard skip/residual connections. Can the authors provide ablation results or empirical justification for this choice, given the known optimization benefits of residual pathways [1]?
2. How does PRISM compare against strong density-estimation baselines such as normalizing flows, diffusion-only approaches, or Gaussian mixture posterior models in both quality and computational cost?
3. Given that neural networks are universal approximators, what motivates this relatively large encoder-decoder design over smaller alternatives (for example, an MLP with residual connections)? Please discuss the performance-versus-compute trade-off.
4. For the dMRI task, can the authors quantify problem complexity and explain how PRISM affects memory usage and inference/runtime cost relative to alternatives?
5. Please report the model size (number of parameters), training wall-clock time, and hardware setup for each experiment.
6. How does PRISM compare with the method in [2], which also uses deep generative modeling for simulation-based inference?
7. Could the authors include a more detailed compute breakdown (training and inference) for PRISM and all baselines to support practical deployment claims?

**Limitations:**

Limitations/Weaknesses
The paper would benefit from a more explicit discussion of optimization stability and convergence behavior. Large neural architectures can suffer from overfitting and training instability, and these issues can directly affect posterior quality. The authors should provide either a stronger theoretical motivation for the objective/training procedure or additional empirical evidence of convergence across complexity settings and evaluation metrics. In addition, although the benchmarking is generally strong, comparisons to classical and widely used posterior estimators (for example, normalizing flows, Gaussian mixtures, and alternative diffusion configurations) could be expanded. Finally, the computational burden of PRISM training should be presented more clearly and transparently and compared to the cumulative cost of competing amortized inference approaches.

**Strengths And Weaknesses:**

Strengths
1. The paper addresses an important and challenging problem: performing simulation-based inference over very large combinatorial model spaces while still enabling post-training control over model complexity.
2. The related work section is comprehensive and demonstrates strong awareness of prior work in simulation-based and amortized Bayesian inference.
3. The proposed architecture is technically interesting, especially the combination of a multivariate Bernoulli decoder for structure inference and a diffusion-based decoder for continuous parameter inference.
4. The experiments include both a controlled synthetic setting and a realistic scientific application, which strengthens the empirical support for the proposed method.

Soundness
1. The approach is grounded in established Bayesian inference principles, and the formulation appears methodologically sound.
2. Using diffusion models for multimodal continuous posterior approximation is a reasonable and appropriate design choice.
3. The empirical study includes meaningful baselines and quantitative metrics for posterior quality, which supports the paper’s claims.

Presentation
The paper is generally well organized and readable. The motivation is clear, and the technical pipeline is understandable. However, a few architectural choices and compute-related details could be explained more explicitly to improve clarity and reproducibility.

Significance
The contribution is potentially significant for scientific domains that rely on expensive simulators and large candidate model families, where retraining for each prior preference is impractical.
Originality
The overall paradigm of deep amortized simulation-based inference is not new, but conditioning inference on a test-time complexity parameter in this specific way is a useful and relatively novel contribution.

---

> ### Author Rebuttal · Authors · 2026-03-31
>
> We thank the reviewer for the constructive review, and for highlighting the “important and challenging problem,” the “technically interesting” architecture, the “comprehensive” related work, and the evaluation on both synthetic and scientific applications.
>
> We appreciate the reviewer’s suggestions for strengthening the paper, which we will implement:
> - **Baseline coverage**. Beyond the existing comparison to SBMI (using Gaussian mixture models), we now additionally compare to a neural spline flow baseline in the fixed-model B3S setting (see Q2). This shows that PRISM’s parameter posterior is on par with a specialized fixed-model approach, while also estimating the model posterior and amortizing over associated parameter posteriors.
> - **Optimization:** We provide training loss curves in A4, Fig. 3, which we will add to Appendix. Overall training is stable, with minor loss spikes (e.g. Fig. 3c [1]). See also R. SKBZ, Q2.
> - **Compute transparency**. While some of this information was already in the Appendix, it was scattered and should be collected centrally. We therefore consolidated compute details into a dedicated Appendix section with tables covering model size, training time, hardware, and key runtimes for PRISM and baselines (see Q4/5/7).
>
> > Q1. “The architecture appears to omit standard skip/residual connections. Can the authors provide ablation results or empirical justification for this choice?”
>
> A1. We do use skip connections; we only omitted them in Fig. 2 for stylistic reasons. We will clarify in the caption.
>
> > Q2. Comparison to density-estimation baselines
>
> A2. While we already compare against SBMI (Schröder & Macke, 2024), which uses Gaussian mixture models for the parameter posterior, we are not aware of any other method tackling the same joint model and parameter inference problem. Training a normalizing flow or diffusion network in the standard setting only allows parameter inference for one fixed model. While this may be feasible in small model classes such as B3S, it becomes prohibitively expensive as the model space grows exponentially with the number of components.
>
> Nonetheless, we added a baseline for the fixed-model B3S task: a comparably large normalizing flow (neural spline flow, NSF, 8 bijective transforms), using a transformer-based embedding network to amortize over multiple acquisition schemes, as PRISM does. Since NSF requires fixed-size summary statistics, we increased the embedding net size and used 4 CLS tokens to pool conditioning data. The final statistic concatenates the CLS-token outputs, yielding a 512-dimensional summary statistic. We aimed to keep model size (~7M parameters) and training process (best after 72h) as similar as possible.
>
> We then applied the applicable metrics (ESS, KSD, RMSE) to the resulting parameter posterior (see A1, Table 1). As expected, this model captures the parameter posterior and is comparable to PRISM on simulated data. On real data, it is slightly worse. While we do not want to claim that PRISM generally outperforms a specialized model, this demonstrates that PRISM’s parameter posterior converged to a good solution, comparable to other model-specific approximators, despite additionally amortizing over other models and the model posterior.
>
> >Q3. Encoder-decoder design
>
> A3. We compare a “smaller alternative” (e.g. Schröder & Macke, 2024) against PRISM, but observe that PRISM outperforms it on larger problems. Although PRISM addresses these tasks using higher-capacity networks, it also offers greater flexibility, as transformers can easily accommodate a wide range of data and parameter formats. Extending PRISM to other domains mainly requires adapting the encoder and tokenizer. See also R. GckH, Q2.
>
> >Q4/5/7. Computational costs and hardware setup
>
> A4/5/7. We agree that this information is valuable and was scattered across the manuscript. We will add an Appendix section where relevant information is gathered. We already created tables collecting this information (A4, Tables 2 and 3).
>
> Parameter counts varied between 1-6 million in the symbolic tasks and 6-9 million in the dMRI task. We provide detailed hardware configurations in the table captions; the main device was a single H100 GPU. Training was run with a fixed time limit: 24h for symbolic tasks and 72h for dMRI.
>
> Training/inference runtimes for all settings are listed in the new supplementary tables A4, Tables 2-3. In summary, we distinguish between: i) sample model, 1-6 ms on dMRI; ii) sample parameter, ~30 ms; iii) forward pass, 10-30 ms (batch size 2048); iv) backward pass, 30-100 ms (batch size 2048). Compared to the NSF baseline (Q2), sampling is slightly slower, log_prob evaluation is much slower (8x, as expected for diffusion), but the backward pass is faster (4x).
>
> > Q6. “How does PRISM compare with the method in [2], …?”
>
> A6. Sorry, can the reviewer clarify what [2] refers to?
>
>
> [1] Supplementary PDF: [here](https://www.swisstransfer.com/d/ead46821-0d93-4190-8535-d281bad7c606)

---

### Official Review · Reviewer_YsWR · 2026-03-13

**Soundness:** 3
**Presentation:** 3
**Significance:** 3
**Originality:** 3
**Overall Recommendation:** 4
**Confidence:** 4

**Summary:**

This paper introduces PRISM, an amortized simulation-based inference framework for joint Bayesian inference over a combination of discrete model structure and continuous model parameters across large combinatorial model families. Models are represented as binary component masks $M$ with the model-specific parameters $\theta_M$. A key feature is test-time complexity control: PRISM is trained while conditioning on a hyperparameter $\lambda$ that parameterizes a hierarchical model prior. This allows users to tune the sparsity level after observing data without the need for retraining. The authors evaluates PRISM on a synthetic additive symbolic regression task with large component libraries and enormous induced model spaces and a scientific case study in diffusion MRI, performing model selection among multi-compartment biophysical models under varying acquisition protocols. Both experiments demosntrate the improvements over prior SBI baselines

**Compliance With Llm Reviewing Policy:**

Affirmed.

**Final Justification:**

The author did a good rebuttal, and I'm happy to maintain my score.

**Key Questions For Authors:**

- How robust is “model discovery” in huge spaces to the decoding/search procedure?
- How does performance change if the $\lambda$-parameterization of the model prior is modified (e.g., different sparsity priors, structured priors, etc.)?

**Limitations:**

The technical limitations are discussed reasonably.

**Strengths And Weaknesses:**

Strengths:

- The paper has a good motivation: for mechanistic scientific simulators, the bottleneck is often model family exploration rather than single-model parameter inference. A method that amortizes across a large model family while producing model uncertainty (not just point selection) is practically valuable.
- The paper in general is easy to read, with clear presentation.
- Regarding methodology, I think the factorization goal is clear: we need to estimate a joint posterior with a user-controlled prior hyperparamers, and PRISM implements this via two amortized conditional models. The training objectives are standard for the chosen parameterizations: BCE / AR NLL for discrete masks and diffusion v-prediction for continuous parameters.
- The symbolic regression experiments explicitly target huge combinatorial spaces, and the paper is transparent that training only covers a tiny fraction of the space; the performance is therefore about generalization across related models, not enumeration.
- In MRI, amortizing across acquisition protocols (b-values, gradient directions, number of measurements) is a meaningful step beyond many SBI pipelines that assume fixed acquisition schemes.

Weaknesses:

- I have a minor concern about novelty of the paper. In particular, the combination of AR discrete structure + diffusion continuous parameters + shared tokenization + conditioning on a prior hyperparameter is interesting, but this feels more like a combination of existing components in the literature. The strongest novelty claim is the test-time complexity control via $\lambda$-conditioning in a joint model+parameter SBI system that works in combinatorial spaces. I suggest the authors should elaborate more clearly the distinction between their method and SBI workflows that enable post-hoc prior adjustment via learned likelihood/evidence surrogates, and broader “conditional amortization” ideas where hyperparameters are treated as context variable.
- The method uses an autoregressive mask model, and since evaluation of the full posterior or true MAP over M in huges spaces is not feasible, the paper uses sampling to pick the best among them. This is reasonable, but it makes the practical optimization quality heavily dependent on the randomness of the sampling method and posterior concentration. A comparison to beam search / structured decoding / SMC over masks could clarify how robust this is.
- The training appears extremely expensive in the large-scale symbolic settings. Even if the simulator is cheap, that’s a major barrier to adoption.
- Attention-based components imply at least quadratic scaling in number of components $C$. This limits extensibility to libraries with thousands of components.

---

> ### Author Rebuttal · Authors · 2026-03-31
>
> We thank the reviewer for the thoughtful review, and especially for highlighting the paper’s “good motivation,” “clear presentation”, the “practical value” for amortizing across large model families, and that amortizing across MRI acquisition protocols is “a meaningful step beyond many SBI pipelines.” We also appreciate the reviewer’s constructive comments on novelty, model discovery in large spaces, and scalability.
>
> Below address the reviewers main concerns :
> - **Novelty:** Viewed component-by-component, PRISM combines established ingredients to a powerful and new setup rather than introducing fundamentally new components. Our contributions are therefore new SBI capabilities their combination enables: joint amortized Bayesian inference over discrete model structure and model-specific continuous parameters across large combinatorial model families, while amortizing over priors. We agree that PRISM is very related to the “broader” conditional amortization strategies, i.e. lambda is indeed very much treated as an additional “context variable”. However, the model identities we amortize over are conceptually different.
> In the context of likelihood surrogates,  the prior can change similarly to our setting, however, this still requires MC estimates of the evidence (which effectively requires posterior estimates for each model). We therefore view the contribution not as a technical combination for its own sake, but as enabling a practically important new SBI regime: direct amortized inference of p(θ,M∣x,λ) for large combinatorial model spaces, with test-time complexity control and without retraining.
> - **MAP estimation:** Discrete optimization within autoregressive models is not a main contribution within our work. However, we acknowledge that MAP estimation is important and we will add a discussion on this topic. More specifically, we now compare to more involved methods as suggested by the reviewer (see Q1).
> - **Computational cost:**  We agree that large-scale amortization is computationally demanding at training time *in general* (e.g. see Reuter et. al, 2025). This is the price of learning a reusable inference procedure across many related models rather than fitting a method for a single instance. In return, the trained model can be deployed flexibly in an amortized fashion across large model families and varying settings without retraining, which is precisely the regime we target. We will make this trade-off clearer in the discussion. We additionally added a table with a detailed break down of the computational cost (see response to R. hMMd, Q7)
>
> >Q1. Robustness of ‘model discovery’
>
> A1. We agree that finding the MAP in discrete space (even given the approximate model posterior) is non-trivial. While in most experiments we are interested in samples from the model posterior (for which methods like Beam Search do not apply/would introduce bias), for model selection we do rely on discrete optimization. This is nontrivial in large spaces, as pointed out by the reviewer. We use a best-of-n (BoN) samples approach (with or without tempered autoregressive decoding). We now investigated this in more detail: we tested on the larger dMRI spaces where model selection was used (e.g. Fig. A12). We compared BoN (with N=128, temp=1,0.5,0.1) vs Beam Search (beam width 16/32). Beam search specifically becomes more useful on the larger spaces, whereas BoN with moderate temperature did also perform well (and can increase performance by increasing N at higher cost). We will add this analysis in A.4.4. to guide the reader on the MAP recovery as suggested by the reviewer. See [PDF](https://www.swisstransfer.com/d/ead46821-0d93-4190-8535-d281bad7c606) Fig. 2.
>
> > Q2. Influence of model prior
>
> A2. Within our work we limited the prior to rather simple families, although more complex structured priors can be incorporated (with the constraint that we can obtain samples from it). In our design, lambda controls the complexity of the model, which will to a large degree be task dependent and can be defined in different ways. For example, in the symbolic regression task a more sophisticated prior would incorporate a measure for symbolic complexity of the basis functions instead of the raw count. In principle, however, we amortize Bayesian inference over a family of priors. While ‘complexity’ is a useful hyperparameter, domain scientists may instead wish to amortize over other domain-specific model priors.
>
> The model prior directly influences the model posterior. However, on a performance level, it will also impact the parameter posterior: The model prior effectively defines an “importance weight” on the model masks i.e. if a model is very unlikely under the prior then making errors will only minorly affect the loss. Thus, models that are rarely sampled under the prior can yield worse parameter posteriors. This is intentional, however, since there is little reason to spend capacity for models that the model prior effectively rules out.

---

> > ### Author Rebuttal · Reviewer_YsWR · 2026-04-02
> >
> > I thank the authors for the thorough response. I still have concerns regarding the novelty of the paper (as the authors also restated as "established ingredients to a powerful and new setup rather than introducing fundamentally new components", therefore I decide to keep my original score (4: weak accept).

---

### Official Review · Reviewer_GckH · 2026-03-13

**Soundness:** 3
**Presentation:** 3
**Significance:** 3
**Originality:** 3
**Overall Recommendation:** 5
**Confidence:** 3

**Summary:**

This paper addresses the problem of model selection over large model families. The proposed method, PRISM, is a simulation-based method that infers the joint posterior over the possible models and their parameters using a transformer-based encoder-decoder architecture. PRSIM also allows controlling for the model complexity at test-time. Experiments on diffusion MRI models demonstrate the real-world applicability of the proposed method.

**Compliance With Llm Reviewing Policy:**

Affirmed.

**Final Justification:**

I initially scored this paper 5, and with the proposed improvements to the manuscript during the rebuttal phase, I am confident that this will be a valuable contribution to the community. I stand by my recommendation of acceptance.

**Key Questions For Authors:**

* How exactly is $\lambda$ controlled at test-time? Do I understand correctly that PRISM amortizes over $\lambda$ as well, and so at test-time we would need to select a value of $\lambda$ to get the joint posterior?
* The choice of model and parameter posterior decoder could be discussed a bit. What is the rationale behind the specific choices of decoders? What could have been other choices?
* On page 6, the authors mention "...it is neither over- nor underconfident...(Fig 6b)" when discussing the SBC results. I am wondering how they came to that conclusion. If this was a coverage plot then I could understand this conclusion looking at Fig 6b, but not sure you can make the same conclusion from an SBC plot.
* How is the true posterior obtained in Section 4.2.1?
* What are the x-axis labels in Fig 8a and how do they differ from that of Fig 8b?
* Minor typo in page 8, section 6: "this trade-off needs not be fixed".

**Limitations:**

Yes

**Strengths And Weaknesses:**

This is a strong paper that makes model selection of scientific simulators possible for a large class of models, which was previously not possible using existing methods, extending the kinds of problems tackled by amortized simulation-based inference (SBI). The paper is polished and generally very well-written, with comprehensive literature review. The experiments on real-world data and simulators are convincing. The chosen performance metrics are sensible. Overall, this is a worthwhile contribution to the SBI literature and I am happy to recommend acceptance.

The only weakness I can point to is that some of the experimental details and results are difficult to interpret for me as I am not familiar with the specific application. I would urge the authors to combine the application level details with some bigger picture overview of what the experiment is trying to test and how the results validate the hypothesis. Similarly, some of the plots are difficult to interpret for me, such as the calibration error plot. More caption details on what we observe in each plot would help the readers follow the conclusions of experiments.

---

> ### Author Rebuttal · Authors · 2026-03-31
>
> We thank the reviewer for the thoughtful review, and especially for highlighting the paper as a “strong paper,” “polished and well-written,” and “convincing” real-world experiments. The main comment concerns the experimental details and interpretation of the results for readers less familiar with the application. We will revise the manuscript as follows:
> - **Introduction of metrics**: We will add an intuitive introduction of the main metrics at the beginning of Sec. 4, complementing the appendix description (see Q3).
> - **Bigger picture**: We agree this can be improved, specifically in Sec. 4 (Results). The previous point should already improve interpretability. In addition, we will provide clearer context by introducing the objective at the beginning of each section and adding brief remarks at the end (e.g. Sec. 4.1, Sec. 4.2.1). For example, Sec. 4.1 now begins with: “To compare PRISM with SBMI, ... . This task is extensible to substantially larger model spaces, allowing us to study the scaling behavior in large model regimes … .” rather than bluntly defining the task.
>
> > Q1. Implications of lambda.
>
> A1. Yes, PRISM amortizes over lambda. The information is injected via adaptive layer normalization to the model posterior network. At test time, the user is free to choose their preferred lambda, which should match their prior assumptions on model complexity. PRISM then infers the joint posterior $p(M, theta | x, \lambda)$. More explicitly, our method amortizes over different priors, parametrized by lambda, to compute the Bayesian posterior. In practice, practitioners often adjust the parsimony trade-off after seeing the data, and PRISM allows this without retraining.
>
> > Q2. Architectural choices.
>
> A2. Many architectural choices are possible, but we chose a transformer-based encoder-decoder because PRISM requires flexibility in both the inputs and the inference task. The encoder must handle variable-length inputs and potentially different modalities, while the decoder must support model-conditioned parameter inference. For the model posterior, we use an autoregressive decoder because autoregressive models often outperform alternatives such as discrete diffusion methods on structured discrete data (e.g. Sahoo et al., “Simple and Effective Masked Diffusion Language Models”). For the parameter posterior, we use a diffusion-based decoder because diffusion models are well suited for flexible multimodal continuous densities. Autoregressive parameter decoders are also possible, but would scale with parameter dimension $O(d_\theta^2)$, whereas our component-level design scales more favorably with the number of components $O(C^2)$. This can be a good alternative for lower parameter dimensions, we will discuss this in the limitation/future work section.
>
> > Q3. Interpretation of the SBC results
>
> A3. Intuitively, SBC assesses the approximate posterior’s calibration over many simulated datasets by checking whether the “true” simulated parameter value looks like a typical draw from the approximate posterior. If the approximate posterior is too narrow, it will place the true value too often in its tails (overconfidence), and if it is too wide, it will place it too often near its center (underconfidence). Calibration curves (nominal vs empirical ranks, i.e. Fig. 6b left) visualize this miscalibration as being “under” or “above” the diagonal (whereas the “Calibration Error” measures the absolute deviation from the diagonal). We will extend our discussion of Fig. 6 to better explain the interpretation of SBC.
>
> > Q4. “How is the true posterior obtained in Section 4.2.1?”
>
> A4. Within Section 4.2.1 we consider two “references” in the B3S setting: Firstly, reference model posteriors (Fig.6a), which we obtain for a subset of models via an unbiased evidence estimator (See A.4.1, Eq. 6). Notably this leverages the tractable likelihood and *PRISM*’s parameter posterior as an importance sampling proposal. Using the prior as a proposal instead leads to a large-variance estimate even with billions of samples. Secondly, MCMC references (FSL bedpostX, Fig. 7): This is the current “gold-standard” MCMC-based alternative for the inference task, not a ground truth. This was used in Manzano-Patron et al. (2025) as reference and revealed some limitations with standard sbi approaches that are important to domain scientists (Fig. 7b). We will clarify these differences in the manuscript.
>
> > Q5. “What are the x-axis labels in Fig 8a and how do they differ from that of Fig 8b?”
>
> A5. We thank the reviewer for pointing out this ambiguity. The x-tick labels in Fig. 8a denote random subset of models such as BS, BZ, 2S, or B2S, corresponding to combinations of Ball, Stick, Zeppelin, or Tensor components, repeated according to the number, analogous to the B3S notation. While this nomenclature was partly explained in the appendix (A.4.2, B.2.5), we will already introduce it in the main text (Sec. 4.2.2). We will also fixed the missing y-tick label at 0.01.

---

> > ### Author Rebuttal · Reviewer_GckH · 2026-04-01
> >
> > Thank you for your response. I urge the authors to revise their manuscript as per the comments from all the reviewers. I believe this would substantially improve the paper's clarity. I am happy to recommend acceptance of this paper.

---

### Decision · Program_Chairs · 2026-04-30

**Decision:**

Accept (regular)

**Comment:**

The reviewers were quite positive on this work as it enables "performing simulation-based inference over very large combinatorial model spaces" that is necessary in large scientific simulators.  Strengths included:
1) Clear presentation with strong motivation and literature review.
2) The experiments encompass synthetic scenarios and realistic scientific applications on large combinatorial spaces.
3) The proposed architecture, while a combination of many existing ideas, is well-motivated and potentially of practical value.

Many reviewers espically liked the real-world scenario.  The reviewers also identified some potential weaknesses/places where the authors could improve the presentation.
1) The computational cost at training time is significant.  Though this is not atypical/unexpected for the task being considered.
2) Some experimental details could be improved/clarified, e.g., reducing the randomness by eliminating sampling, further clarifying training issues of practical value such as convergence, overfitting, etc., and practical discussions regarding scalability.

Overall, while there are some potential questions regarding how practical the approach is, it appears to represent a significant contribtuion to the simulation-based model inference community.